



# TephraNZ: a major and trace element reference dataset for prominent Quaternary rhyolitic tephras in New Zealand and implications for correlation

Jenni L. Hopkins[1], Janine E. Bidmead[1], David J. Lowe[2], Richard J. Wysoczanski[3], Bradley J. Pillans[4], Luisa Ashworth[1], Andrew B.H. Rees[1], Fiona Tuckett[1]

[1]School of Geography Environment and Earth Science, Victoria University of Wellington, Wellington, PO Box 600, New Zealand

[2]School of Science (Earth Sciences), University of Waikato, Hamilton, Private Bag 3105, New Zealand 3240
[3]National Institute of Water and Atmospheric Research, Wellington, Private Bag 14901, New Zealand
[4]Research School of Earth Science, Australian National University,

*Correspondence to*: Jenni L. Hopkins (jenni.hopkins@vuw.ac.nz)



## Abstract

Although analyses of tephra-derived glass shards have been undertaken in New Zealand for nearly four decades (pioneered by Paul Froggatt), our study is the first to systematically develop a formal, comprehensive, open access, reference dataset of glass-shard compositions for New Zealand tephras. These data will provide an important reference tool for future studies to identify and correlate tephra deposits and for associated petrological and magma-related studies within New Zealand and
beyond.

Here we present the foundation dataset for "TephraNZ", an open access reference dataset for selected tephra deposits in New Zealand. Prominent, rhyolitic, tephra deposits from the Quaternary were identified, with sample collection targeting original type sites or reference locations where the tephra's identification is unequivocally known based on independent dating or mineralogical
techniques. Glass shards were extracted from the tephra deposits and major and trace element geochemical compositions were determined. We discuss in detail the data reduction process used to obtain the results and propose that future studies follow a similar protocol in order to gain comparable data. The dataset contains analyses of twenty-three proximal and twenty-seven distal tephra samples characterising 45 eruptive episodes ranging from Kaharoa ($636 \pm 12$ cal. yrs BP) to the Hikuroa Pumice
member ($2.0 \pm 0.6$ Ma) from six or more caldera sources, most from the central Taupō Volcanic Zone. We report 1385 major element analyses obtained by electron microprobe (EMPA), and 590 trace element analyses obtained by laser ablation (LA)-ICP-MS, on individual glass shards.

Using PCA, Euclidean similarity coefficients, and geochemical investigation, we show that chemical compositions of glass shards from individual eruptions are commonly distinguished by major
elements, especially $CaO$, $TiO_2$, $K_2O$, $FeO_t$ ($Na_2O + K_2O$ and $SiO_2/K_2O$), but not always. For those tephras with similar glass major-element signatures, some can be distinguished using trace elements (e.g. HFSEs: Zr, Hf, Nb; LILE: Ba, Rb; REE: Eu, Tm, Dy, Y, Tb, Gd, Er, Ho, Yb, Sm), and trace element ratios (e.g. LILE/HFSE: Ba/Th, Ba/Zr, Rb/Zr; HFSE/HREE: Zr/Y, Zr/Yb, Hf/Y; LREE/HREE: La/Yb, Ce/Yb).

Geochemistry alone cannot be used to distinguish between glass shards from the following tephra groups: Taupō (Unit Y in the post-Ōruanui eruption sequence of Taupō volcano) and Waimihia (Unit S); Poronui (Unit C) and Karapiti (Unit B); Rotorua and Rerewhakaaitu; and Kawakawa/Ōruanui, Okaia, and Unit L (of the Mangaone subgroup eruption sequence). Other characteristics can be used to separate and distinguish all of these otherwise-similar eruptives except Poronui and Karapiti.
Bimodality caused by $K_2O$ variability is newly identified in Poihipi and Tahuna tephras. Using glass shard compositions, tephra sourced from Taupō Volcanic Centre (TVC) and Mangakino Volcanic Centre (MgVC) can be separated using bivariate plots of $SiO_2/K_2O$ vs. $Na_2O + K_2O$. Glass shards from tephras derived from Kapenga Volcanic Centre, Rotorua Volcanic Centre, and Whakamaru Volcanic Centre have similar major- and trace-element chemical compositions to those from the MgVC, but can
overlap with glass analyses from tephras from Taupō and Okataina volcanic centres. Specific trace elements and trace element ratios have lower variability than the heterogeneous major element and bimodal signatures, making them easier to geochemically fingerprint.



## 1. Introduction

Tephrochronology is the principle by which volcanic ash (tephra) deposits are used as stratigraphic isochronous marker horizons (isochrons) for correlating, dating, and synchronising deposits and events in geologic, palaeoenvironmental, and archaeological records (Sarna-Wojcicki, 2000; Shane, 2000, Dugmore et al., 2004; Lowe, 2011; Alloway et al., 2013). In regions where rates of volcanism are high, and eruptive products are widespread, tephrochronology is an essential tool in many
aspects of geoscience and associated research. Geochemical fingerprinting of the glass shards within the tephra deposits is one of the most common ways in which tephra are correlated. Traditionally, major elements were used for correlations (e.g. Westgate and Gorton, 1981; Froggatt, 1983, 1992), but more recent studies have included minor and trace element compositions as well (e.g. Westgate et al., 1994; Pearce et al., 2002, 2004, 2007; Pearce, 2014; Knott et al., 2007; Allen et al., 2008; Denton and Pearce,
2008; Turney et al., 2008; Westgate et al., 2008;  Kuehn et al., 2009; Hopkins et al., 2017; Lowe et al. 2017).

      Trace elements are more strongly partitioned by fractional crystallisation processes that occur during the formation of melts, and therefore have the potential to be unique for discrete eruption episodes (e.g. Pearce et al., 2004). Specifically, a number of key trace elements have been identified as
important for the correlation of rhyolitic tephras, including the high field strength elements (HFSEs) Zr and Nb; the large ion lithophile elements (LILE) Rb, Sr, and Ba; the heavy rare earth elements (HREE) Gd, Yb, Sc, and Y; and the light rare earth elements (LREE) La and Nd. Also identified are important trace element ratios, including: (1) HFSE/HREE – for example Zr/Y, Nb/Y, Hf/Y; (2) LILE/HFSE – for example Ba/Th; (3) LREE/HFSE – for example Ce/Th, La/Nb; (4) LREE/HREE – for example La/Yb,
Ce/Yb; and (5) HFSE/HFSE – for example Zr/Nb, Zr/Th. Some studies have shown that trace elements and trace element ratios can distinguish between tephra beds that have indistinguishable glass-shard major element signatures and thus are a more robust way of providing accurate correlations (e.g. Westgate et al., 1994; Pearce et al., 1996; 2002; 2004; Allan et al., 2008; Hopkins et al., 2017).

      Tephra correlation is also increasingly being quantified through statistical approaches on
geochemical data (Lowe et al., 2017), but many of these approaches (supervised learning) often require a robust (comprehensive) set of "known" reference data against which to test the analyses of "unknown" samples. Statistics can also scale data to make them comparable, but they cannot account or correct for inter-laboratory or historical variance in analyses. Therefore, incomplete datasets, or datasets constructed from a range of data sources, will limit the ability to provide holistic statistical correlations
with accurate outputs. Therefore the formation of reference datasets that are run in one analytical session, in one lab, with a consistent methodology are highly valuable but difficult to obtain.  The production of tephra databases is thus being recognised as an exceptionally useful tool internationally (e.g. Lowe et al., 2017), made more obtainable with open access journals and online, effectively limitless storage, leading to easier publication and maintenance of large data repositories. Ideally, a



global tephra database would exist, but at present this is beyond the scope and remit of any individual researchers or institutes. Therefore regional databases for volcanically active and other regions are becoming increasingly popular, such as TephraKam – Kamchatka (Portnyagin et al., 2019); Tephrabase – Europe (Newton, 1996); AntT tephra database – Antarctic ice cores (Kurbatov et al., 2014); Alaska Tephra Database (Wallace, 2018); Klondyke Goldfields, Yukon (Preece et al., 2011); VOLCORE –

DSDP, ODP, and IODP marine tephra deposits (Mahony et al., 2020).

### 1.1. Geologic Setting

The volcanically active nature of New Zealand (Mortimer and Scott, 2020), and the longevity and consistency of large-scale rhyolitic eruptions (Howorth, 1975; Froggatt and Lowe, 1990; Houghton

et al., 1995; Wilson et al., 1995a, 2009; Jurado-Chichay and Walker, 2000; Carter et al., 2003; Briggs et al., 2005; Wilson and Rowland, 2016; Barker et al., 2021) mean the landscape currently has a very long, detailed, and complex rhyolitic tephrostratigraphic framework that is used for a wide range of applications. However, at present New Zealand tephra studies are lacking a comprehensive reference dataset resource that has been developed in a systematic way.

The first large rhyolite-producing eruptions in the Quaternary in New Zealand were sourced from the Coromandel Volcanic Zone (CVZ) (Carter et al., 2003; Briggs et al., 2005). At or after ~2 Ma, volcanism moved into the Taupō Volcanic Zone (TVZ), currently the most active rhyolitic system on Earth (Wilson et al., 1995a, 2009; Wilson and Rowland, 2016). Nine calderas are recognised within the TVZ : Mangakino (1.6–1.53 Ma and 1.2–0.9 Ma); Kapenga (0.9–0.7 Ma, 0.3–0.2 Ma, and ~0.06 Ma);

Whakamaru (0.35–0.32 Ma); Reporoa (~0.23 Ma); Rotorua (~0.22 Ma); Ohakuri (~0.22 Ma); Maroa (0.32–0.013 Ma); Taupō (0.32–0.0018 Ma); and Okataina (~0.6–0 Ma) (**Fig. 1B**; Houghton et al., 1995; Wilson et al., 1995a, 2009; Gravely et al., 2006, 2007). The TVZ is further subdivided into the "old TVZ", which is defined as being active from inception to the Whakamaru eruptives (~0.34 Ma), and the "young TVZ", which is defined as being active from the Whakamaru eruptives to the present. "Modern

TVZ" is also used to describe the activity since the Rotoiti eruption (which includes the Rotoehu Ash and Matahi Scoria members) ~45-47 ka (Danišík et al., 2012; Flude and Storey et al., 2016) to the present (Wilson et al., 1995a, 2009). In addition to these rhyolitic caldera sources in the TVZ and CVZ, the peralkaline rhyolitic Tuhua/Mayor Island (MI) volcano (**Fig. 1**), forming the Tuhua Volcanic Centre (TuVC) (Froggatt and Lowe, 1990), is responsible for (amongst at least six other dispersed MI-derived

tephras; Shane et al., 2006) the Tuhua tephra (7637 ± 100 cal. yr BP; Lowe et al., 2019), a well-recognised mid-Holocene rhyolitic marker horizon within the New Zealand geologic record due to its distinctive peralkaline geochemistry and mineralogy (Buck et al., 1981; Hogg and McCraw 1983; Froggatt and Lowe 1990; Wilson et al., 1995b; Lowe et al., 1999; Shane et al., 2006).





New Zealand sits in the path of predominantly westerly to southern-westerly winds, and
therefore the majority of tephra plumes are dispersed to the east of the volcanic zones (Barker et al.,
2019). However, tephra deposits from these rhyolitic eruptions are found in a range of different
environments, including:
(1) Marine (e.g. Nelson et al., 1985; Carter et al., 1995; Alloway et al., 2005; Allan et al., 2008; Lowe,
2014; Hopkins et al., 2020)
(2) Lacustrine (e.g. Lowe, 1988; Shane and Hoverd, 2002; Molloy et al., 2009; Shane et al., 2013;
Hopkins et al., 2015, 2017; Peti et al., 2020)
(3) Bog settings (e.g. Lowe, 1988; Newnham et al., 1995, 2007, 2019; Lowe et al., 1999, 2013; Gehrels
et al., 2006), or
(4) within terrestrially exposed (commonly marine or riverine) sediments, for example in the
140            - Whanganui Basin (e.g. Seward, 1976, Naish et al., 1996; Pillans et al., 2005; Rees et al., 2019),
- Wairarapa region (e.g. Shane and Froggatt, 1991; Shane et al., 1995; Nicol et al., 2002), or
- Hawke's Bay region (e.g. Erdman and Kelsey, 1992; Bland et al., 2007; Orpin et al., 2010;
Hopkins and Seward, 2019) (**Fig. 1**).
Because of their pervasive nature, high repose period, and high preservation potential, tephra
deposits are a common chronological aid in many studies in New Zealand. For example, the eruption of
Kaharoa ($636 \pm 12$ cal. yr BP, Hogg et al., 2003) from Mt Tarawera in the Okataina Volcanic Centre
(OVC) has been used to date the arrival of Polynesians in northern New Zealand and map their
expansion and impact across the country (Newnham et al., 1998; Lowe and Newnham, 2004).  The
Rerewhakaaitu eruption ($17,496 \pm 462$ cal. yr BP; Lowe et al., 2013), sourced from OVC, is used as a
marker horizon for the transition between the last glacial and present interglacial (Newnham et al.,
2003), and several other widespread late Quaternary tephra deposits form boundaries or key
stratigraphic markers in the New Zealand Climate Event Stratigraphy developed by the NZ-INTIMATE
community (e.g. Kawakawa/Oruanui tephra; Barrell et al., 2013; Lowe et al., 2013). Compositions of
glass and mineral components from rhyolitic tephra deposits have also been used to reconstruct changes
in magmatic systems, and give insight into the complexity of caldera-related eruption episodes (e.g.
Smith et al., 2002, 2005; Cooper et al., 2012; Barker et al., 2016, 2021; Wilson and Rowland, 2016).
Many of the commonly found rhyolitic tephra horizons in New Zealand are well studied, dated,
and geochemically and mineralogically characterised. However, often these studies have been eruption-,
source-, or depocentre-specific, and thus only provide a small, effectively piecemeal catalogue of tephra
geochemistry that is not necessarily comparable to those of other studies. In addition, data are not
usually published in their entirety, or not at all, meaning future studies cannot access nor use the data
for correlation techniques. Furthermore, Lowe et al. (1999) identified that differing procedural methods
employed at different institutes around New Zealand before and after 1995 produced variable elemental
concentrations for the same tephra (post-1995 $SiO_2$ values were lower by 0.5−1.0 wt.%, and all other



elements had slightly higher values). Therefore, it is likely that some of the older tephra compositions that have been relied upon in the past for correlative purposes are no longer appropriate.

It is therefore timely for a comprehensive, systematic, and accessible New Zealand tephra database to be initiated and developed. In this study we present "TephraNZ" as a foundation reference dataset of internally consistent, open access data for major and trace element compositions of glass

shards from a selection of the most pervasive Quaternary tephra deposits in New Zealand (**Table 1**). This is by far the most complete dataset of New Zealand tephra-derived glass-shard compositions published to date. We discuss in detail the sample preparation, methods of analysis and data reduction processes used to obtain and interrogate the data, thereby providing a template for future studies to produce comparable datasets. Using the glass-shard data obtained, we present an overview of the

geochemical variability for a range of rhyolitic tephras of the TVZ, we suggest key geochemical parameters that can be used to identify the individual tephra layers, and apply common statistical principles to explore the data. Finally, we propose some future avenues of study, utilising these data, which would aid in the progression of a formal, holistic New Zealand tephrostratigraphical framework.

## 2. Methods

### 2.1. Sample collation and collection

Known tephra samples in personal collections were collated, prepared, and reanalysed for this study. Where samples were lacking for key tephra deposits, their type localities were found and samples obtained through field work (**Fig. 1**). **Table 1** provides full details of all the sample locations including their status as either proximal (0-10s km from source) or distal (10-100s km from source) and GPS co-

ordinates for their exact sampling location. We note here that we have not attempted to sample multiple tephra beds from a single eruptive episode in proximal sequences, nor deposits of the same tephra at different azimuths, as has been undertaken in some more localised or petrologically-focussed studies (e.g. Shane et al., 2005, 2008). We recognise this limitation but instead have concentrated on analysing a wide range of pervasive rhyolitic tephras, both proximal and distal, in a systematic and well-

documented way so that future tephrostratigraphic studies will have a foundation of new, high-quality glass-shard compositional data for facilitating robust correlations and applications. Where we have both, we compare proximal and distal analyses of the same tephra and comment on similarities or differences. In addition, we have used statistical methods to demonstrate the integrity of our new datasets (and show how such methods can enable unknown tephras to be classified).




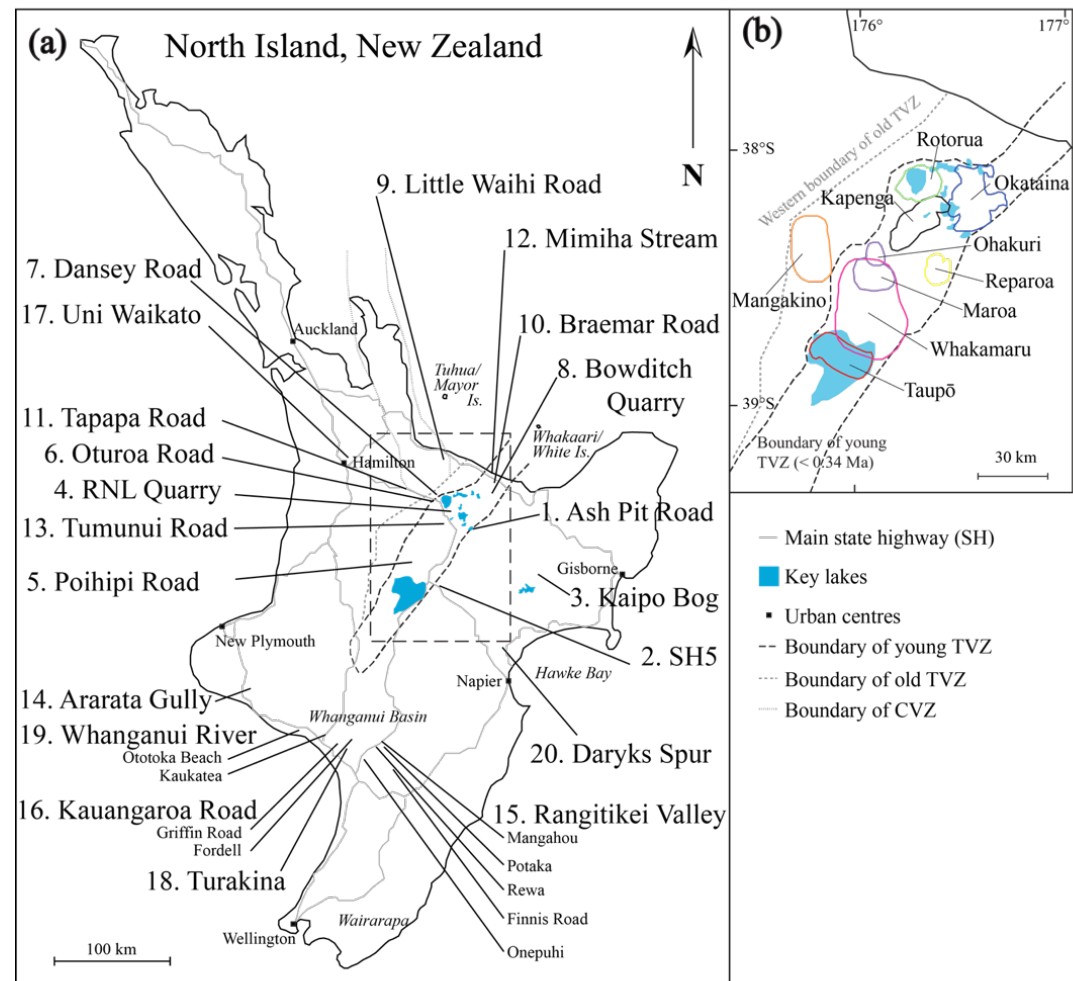

**Figure 1. (a) Map of the North Island, New Zealand, detailing the samples sites where the reference tephra deposits for the TephraNZ database were collected. Outlines of CVZ (Coromandel Volcanic Zone) and TVZ (Taupō Volcanic Zone) are shown by dashed lines. Exact co-ordinates for all sample sites are detailed in Table 1. (b) Inset, outline shown in (a), the calderas of TVZ (details from Houghton et al., 1995; Wilson et al., 1995a ,2009; Gravely et al., 2006, 2007); outline colours of the calderas are used throughout this article in graphs to link the tephra data with their source caldera, if known.**




### 2.2. Sample preparation

205        Bulk tephra samples were disaggregated in water for 1-5 min in an ultra-sonic water bath. Clays and ultra-fines (< 5 µm) were rinsed off and samples were then wet sieved using disposable sieve cloths to 125–250 µm shard size or, where necessary, 60–125 µm. Samples were then dried for 12–24 hr at 50º C before mounting in epoxy resin. Seven samples were mounted into individual drill holes (4-mm diameter) in 25 mm epoxy round blocks (a 4:1 ratio of EpoTek 301 resin [A]: hardener [B]). Individual

drill holes were then backfilled using the same epoxy mix (see Lowe, 2011, p. 124, for a schematic illustration). Sample blocks were polished using the following sequence: ~3 min in a figure of eight pattern on 800 grit paper with water lubricant to remove the epoxy and break through to the glass shards, ~1 min on 1200 grit paper with water lubricant to remove any large scratches, and ~1 min on 2500 grit paper with water lubricant to begin to reveal the outline of the shards. Blocks were then

moved on to the diamond laps with their appropriate lubricant, all at 280 revolutions min$^{-1}$ rotating the block 90 degrees every 30 s followed by 2 min of ultrasonic bathing at < 24 ºC between each lap stage to remove any loose material on the surface of the blocks: ~ 3 min on 6 µm, ~ 1 min at 3 µm, and ~ 1 min at 1 µm.  Blocks were then carbon coated before loading in the electron microprobe system for analysis (EMPA).


### 2.3. EPMA method and data reduction

       Major element analysis of glass shards was undertaken at Victoria University of Wellington (VUW) by wavelength dispersive X-ray spectroscopy (WDS) on a JEOL JXA8230 Superprobe electron probe microanalyser (EPMA). Broadly the method follows that espoused by Kuehn et al. (2011).

Backscatter electron images of each sample were taken and used as block maps to allow the location of EPMA analyses to be replicated for trace element analysis. A defocused circle beam 10 µm in diameter was used at 8 nA to analyse all major elements as oxides ($SiO_2$, $TiO_2$, $Al_2O_3$, $FeO_t$, MnO, MgO, CaO, $Na_2O$, and $K_2O$). During standardisation, $Na_2O$ was run twice, the second time skipping the peak search to reduce the volatilisation of the element, with the second standardisation value then used. **Table 2**

shows the EPMA set up and run times. During the analysis, VG-568 was run as a calibration standard, and VG-A99 and ATHO-G were run as secondary standards, with two of each standard (calibration and secondary) analysed between ten sample analyses to monitor machine drift.

       Initial concentrations were determined using the ZAF correction method, with secondary offline data reduction undertaken to correct for variability in VG-568. Internal correction values were

calculated using the GeoREM reference values of VG-568 (**Eq. 1**) and applied to all the data (**Eq. 2**). Following this, samples were corrected for deviations from 100 wt.% total, this assumes any variation is



due mostly to magmatic water, with a very small amount of minor and trace elements (Froggatt, 1983; Lowe, 2011) that are not analysed by the EPMA (**Eq. 3**). The difference is reported as "$H_2O_D$" in all data tables to allow back calculation to original data values including totals. Accuracy and analytical
precision of the standards were calculated, where accuracy is the percentage offset from the reference value for the secondary standards (**Eq. 4**), and precision is the standard deviation of all of the measured secondary standards throughout a run, reported at 2 standard deviations (sd) to represent a 95% variability.

***Eq. (1)***            *Internal correction value = average($X_m^p$/$X_r^p$)*

where $X_m^p$ = measured concentration of element X of the calibration standard, and $X_r^p$ = reference concentration for element X of the calibration standard (reference values taken from GeoRem preferred values http://georem.mpch-mainz.gwdg.de/).

***Eq. (2)***            *corrected data = $X_m^i$/ internal correction value*

where $X_m^i$ = measured concentration for element X of any sample or standard.

***Eq. (3)***     *Secondary hydration corrected data = ((corrected data (Eq.2)/total for that sample) x 100)*

***Eq. (4)***            *% offset from standard (accuracy) = ((($X_r^s$ – average$X_m^s$)/ $X_r^s$) x 100)*

where $X_r^s$ = reference concentration for element X of the secondary standard (GeoRem preferred value), and average$X_m^s$ = average concentration measured for element X of all analyses)

**2.4. LA-ICP-MS method and data reduction**

*In situ* trace element analysis was undertaken at VUW using laser-ablation inductively-coupled-plasma mass-spectrometry (LA-ICP-MS) where a RESOlution S155-SE 193 nm ArF excimer laser system was coupled with an Agilent 7900 quadrupole ICP-MS. Data for 43 trace elements were acquired using a static spot method, with a 25 µm spot size, ablation time of 30 s, repetition rate of 5 Hz
power (see **Supplementary Material Table 1** for full LA-ICP-MS set up details). Synthetic glass standards NIST-612 and NIST-610 were used to tune the ICP-MS and obtain the P/A factors, at a range of spot sizes and laser powers. During the analysis a full range of standards were analysed to determine which produced the most accurate and precise results as a calibration standard, including NIST-612,



NIST-610, BHVO2-G, and ATHO-G. StHS6/80-G was analysed as a secondary standard throughout,
and all standards (calibration and secondary) were analysed twice every ten samples. All data were
reduced offline using Iolite v.3$^{TM}$ software (Paton et al., 2011), using $^{43}$Ca as the internal standard value
(index channel) and the "*Trace_Elements_IS*" data reduction scheme (DRS). The data were reduced
against ATHO-G as the calibration standard. No post-processing data reduction was necessary for the
trace elements data; precision and accuracy were calculated on STHS6/80-G as described above (**Eq.**
**4**).

### 2.5. Standardisation method

       Multiple calibration standards with different trace element concentrations were analysed to
determine which would be most suitable for trace element data reduction. Potential calibration standards
included NIST-612, NIST-610, BHVO2-G, and ATHO-G. These were each run twice every ten
samples, along with secondary standard STHS6/80-G. **Figure 2** shows the STHS6/80-G results of a
range of selected, commonly-used trace elements, including Zn (transition metal), Rb (LILE), Zr
(HFSE), La (LREE), Yb (HREE), normalised using each of the calibration standards. Overall, the
results show that for the lighter masses (e.g. Zn) there is a large variability in the measured STHS6/80-
G values across the different standards, but all except BHVO2-G sit within error (2 sd) of the reference
value (**Fig. 2**). For the heavier masses, the variation from the reference value observed within the
analysed values decreases, except for NIST-610, which remains highly variable in the middle masses
(Rb, Zr, **Fig. 2**), with variability reducing in the heavier masses (La, Yb, **Fig. 2**). The data show that the
use of ATHO-G as the calibration standard (for data reduction of rhyolites) produces the most accurate
and precise data for the secondary standard, for all except the elements with the heaviest masses, and
smallest concentrations (e.g. Yb).



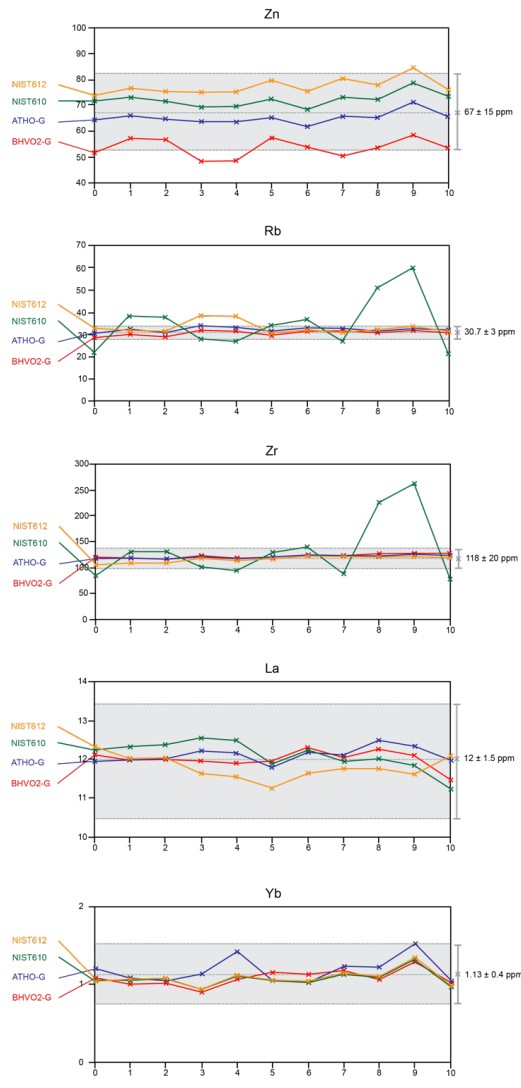

**Figure 2. Compilation of trace element standard data produced during the first run of glass shard analyses. These data show selected element concentrations of secondary standard STHs6/80 normalised using difference calibration standards (NIST-612 – orange; NIST-610 – green, ATHO-G – blue, and BHVO2-G – red). The grey shaded area shows the preferred GeoREM reference value (http://georem.mpch-mainz.gwdg.de/) error margin reported for each element for STHS6/80.**






### 2.6. Statistical methods

#### 2.6.1. Principal Component Analysis

To determine the elements that show the most variance with the reference dataset, and therefore the most appropriate (optimum) for compositional separation, we have used principal component analysis (PCA). PCA analysis was run in the coding platform R and RStudio using packages "factoextra", "ggbiplot", "vegan", "cowplot", "rioja", and "ggrepel". Data for Tuhua tephra were removed as these would unnecessarily skew the results due to their distinct geochemistry. Only element values were used (no ratios [e.g. $SiO_2/K_2O$] or sums [e.g. $Na_2O + K_2O$]). All element values were

centred (column mean subtracted from each value) and scaled (value divided by the standard deviation of the column) to allow their variability to be comparable, even when their absolute values are not. PCA was run using the "prcomp" function, and PCA contributions were calculated using "fiz_comp" function. A template of the coding script used can be found in **Supplementary Material 1**.

#### 2.6.2. Euclidean similarity coefficients

To identify the tephra samples that were most similar, and could therefore pose problems in unique fingerprinting, we ran euclidean similarity coefficients (ESC) analysis. ESC was run on the coding platform R and RStudio using the package "stats". Following the guidelines of Hunt et al. (1995) for ESC analysis, we used non-normalised, mean concentrations of the elements highlighted by the PCA to be the most indicative of variance in the dataset. These values were input as comparison values,

and the function "as.matrix.dist" was used to run the "euclidian" statistical method. This method calculates the similarity or samples based on an infinite number of comparison input values. A template of the coding script used can be found in **Supplementary Material 1**, the output table was manipulated post-production to provide the colour formatting shown in **Figure 14**.

### 3. Results

The averages and their standard deviations for all samples are reported in **Table 3**; the full reference dataset can be found in **Supplementary Material Table 2**. All reported values in the text and figures (unless stated otherwise) are recalculated (normalised) to 100% on a volatile-free basis (following Lowe et al., 2017) with the difference between the raw total and 100% being reported as "$H_2O_D$" (**Table 3**).  For best correlation results, we recommend that the full dataset is used in order to

see the trends in the geochemical data rather than just the means and standard deviations.



### 3.1. Major element results

All glass shards analysed are characterised as rhyolitic according to the classification of Le Maitre (1984) (**Fig. 3**), with $SiO_2$ concentrations ranging from 72.5 wt.% to 79.8 wt.% (with the majority 74-79 wt%), and $Na_2O + K_2O$ ranging from 5.2 wt.% to 9.8 wt.%. Three compositional regions
with high concentrations of samples are evident within **Figure 3**. These show a negative trend between $SiO_2$ and $Na_2O + K_2O$, with each region separated by differing $SiO_2$ values – for example, $SiO_2$ = 76-77 wt.%, 77-78 wt.%, and 78-79 wt.%. Glass samples from the peralkaline Tuhua tephra (Tuhua/Mayor Island, TuVC) are identifiable because of their unique (peralkaline) geochemistry, with much higher $Na_2O + K_2O$ ($\geq$ 9 wt.%) for equivalent $SiO_2$ (= 73.5-75 wt.%; Lowe, 1988) in comparison to those of
the rhyolitic TVZ-sourced deposits ($Na_2O + K_2O \leq$ 8.5 wt.%). Tuhua-tephra-derived glasses also have higher FeO ($\geq$ 5.6 wt.%) and $Na_2O$ ($\geq$ 4.7 wt.%), but lower CaO ($\leq$ 0.8), and $Al_2O_3$ ($\leq$10.1) in comparison to the analyses for the rest of the samples (FeO = 0.2-2.8 wt.%, $Na_2O$ = 2.6-5.1 wt.%, CaO = 0.5-2.6 wt.%, and $Al_2O_3$ = 11.8-15.2 wt.%; **Fig. 4**). For all other major elements, the compositional variation of the Tuhua tephra samples sits within the overall range for the other samples, with $TiO_2$ =
0.02-0.55 wt.%, MnO = 0.01-0.2 wt.%, MgO = 0.01- 0.63 wt.%, $K_2O$ = 1.8-6.0 wt.%, and Cl = 0.01-0.72 wt.% (**Fig. 4**).

Of the 45 tephra samples, 22 have a 'homogeneous signature', homogeneity being defined here where the standard deviation of the sample is equal to or less than analytical error (2sd of secondary standard, e.g. for FeO = $\pm$ 0.23 wt.%, CaO = $\pm$ 0.10 wt.%). The majority (~64%) of the samples that
have a homogeneous signature are from OVC (e.g. Whakatane, Mamaku, Rotoma) or from calderas older than OVC (~32%), for example: Upper Griffins Road tephra, a correlative of the Whakamaru eruptives, Whakamaru Volcanic Centre (WVC), and Mangapipi tephra, a correlative to deposits of Mangakino Volcanic Centre (MgVC; **Fig. 5a**). Ten samples show a heterogeneous signature (where standard deviations for both FeO and CaO are greater than analytical errors), with most from a proximal
source (~30%), or from tephras deposited in the Whanganui Basin area (40%), and with the remainder being from the Mangaone Subgroup eruptives from the OVC: Hauparu, Maketu, and Ngamotu (**Fig. 5b**).

Glass shards from four tephra samples show a bimodal signature in major and trace elements, where the populations split into two distinct groups. Tephras showing this phenomenon include Rotorua
(OVC), Rerewhakaaitu (OVC), Poihipi (TVC), and Tahuna (TVC). The bimodal signatures of Rerewhakaaitu and Rotorua are well documented (Shane et al., 2008), whereas those of Poihipi and Tahuna are newly identified here (**Fig. 6**). All four of these tephra horizons have their glass-shard bimodal signatures produced predominantly by $K_2O$ concentrations, into high ($\geq$ 3.8 wt.%) and low ($\leq$ 3.6 wt.%) populations (**Fig. 6**) linked to the crystallisation of biotite minerals.

For five of the tephras, we undertook analyses from both proximal and distal samples. These tephras included Whakatane, Rotoma, Waiohau, Rotorua, and Rerewhakaaitu, which are all derived





from OVC (**Table 1**). For Rotoma, Rerewhakaaitu, and Waiohau, the signatures of the proximal and distal deposits are indistinguishable, whereas, for Whakatane and Rotorua the proximal signature is highly variable, and the distal signature is homogeneous but overlapping with part of the extent of the

proximal signature (**Fig. 7).** Similar findings are reported and discussed in more detail for Whakatane tephra in Kobayshi et al. (2005) and Holt et al. (2011); and for Rotorua tephra in Shane et al. (2003a) and Kilgour and Smith (2008).

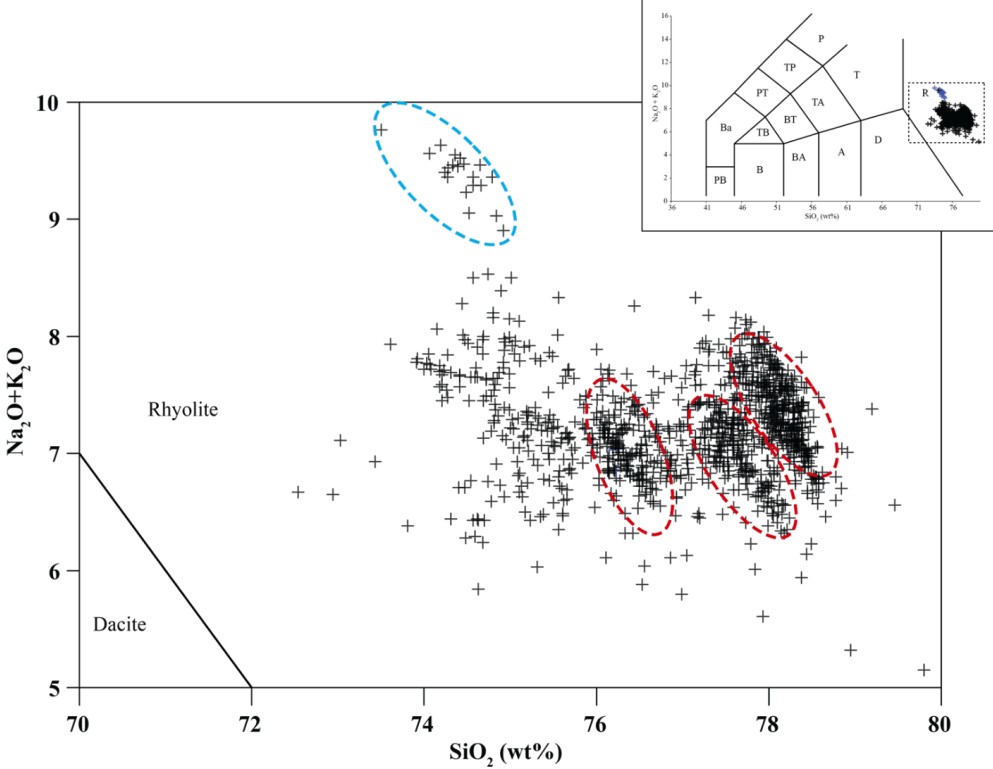

**Figure 3. Total alkali (Na$_2$O + K$_2$O) vs. SiO$_2$ (TAS) plot for glass compositions for all reference data (presented on a normalised basis). Identified and highlighted by blue dashed outline are the glass shard compositions for the Tuhua tephra (Mayor Island; MI), and highlighted by the red dashed outlines are the regions on the TAS diagram that show the highest density of samples. The inset shows a full TAS diagram (always on an anhydrous basis) to provide context for the enlarged figure. Regions of the TAS diagram follow the nomenclature of Le Maitre (1984) are: A – andesite, B – basalt, Ba – basanite, BA – basaltic andesite, BT – basaltic-**
**trachyte, D – dacite, P – phonolite, PB – picro-basalt, PT – phonotephrite, R – rhyolite, T – trachyte, TA – trachy-andesite, TB – trachy-basalt, TP- tephriphonolite.**

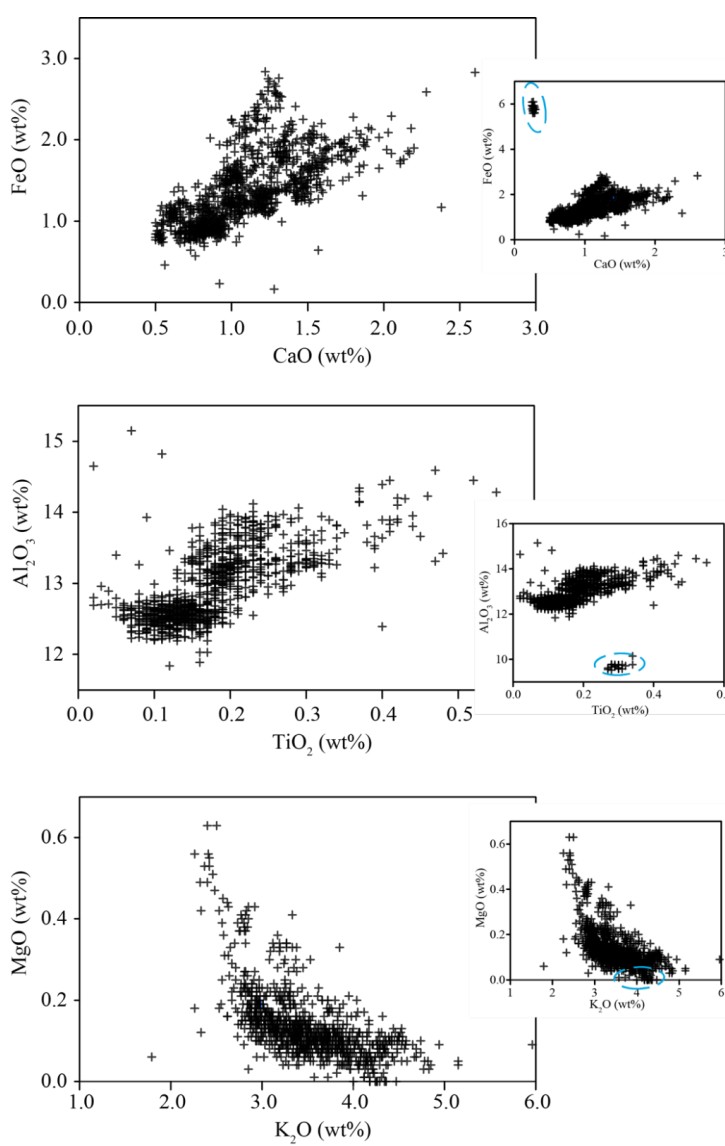

**Figure 4. Major element bivariate plots of glass shard compositions for all reference data (presented on a normalised basis). Highlighted in the insets by blue dashed lines are the Tuhua tephra samples. These are removed from the enlarged figure to allow the detail of the majority of the samples to be seen more clearly. Total iron expressed as FeO.**




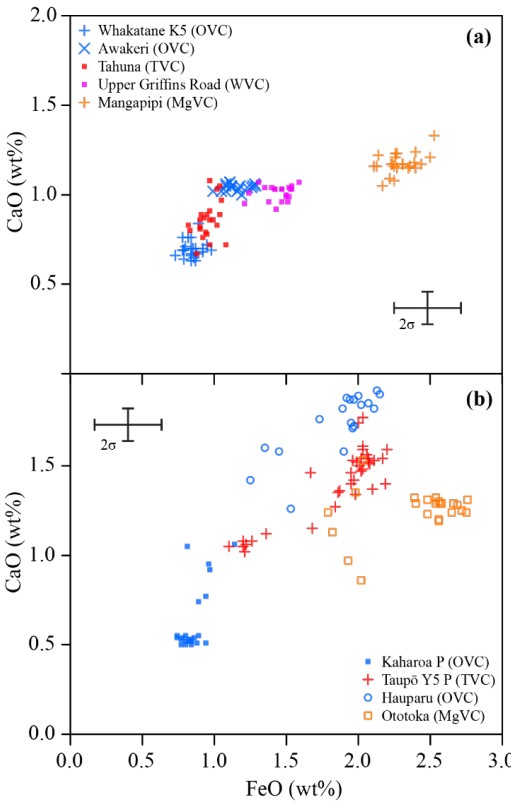

**Figure 5. Major element bivariate plots examples for glass shard analyses of tephras (presented on a normalised basis) which show (a) a homogeneous signatures, where the standard deviation of the analysis is less than the analytical error (shown as 2σ); and heterogeneous signatures, where the standard deviation of the analysis is greater than the analytical error. Different colours indicate the differing caldera sources (shown on Fig. 1) and different symbols show the different tephras. P = proximal sample (see Table 1). Total iron expressed as FeO.**





### 3.2. Trace element results

**Figure 8** shows a chondrite-normalised spider plot of all the trace element data for the glass
shards analysed. The majority of the data plots along a common pattern of variable concentrations of
HFSE, LILEs and LREEs, but they show more consistent concentrations of HREEs. Of note are peaks
in Ba, Nd, Pr, and a negative Sm anomaly. Sr shows the largest variability in concentrations from < 1 to
≤100 ppm, likely caused by a variability in feldspar crystallisation (Pearce et al., 2004). Several
different patterns are observable within this full data suite pertaining to individual samples. The
obviously different signature is that for glass from Tuhua tephra which shows a low concentration of Ba
(< 10 ppm) and Sr (<1 ppm) in comparison with values for the rest of the samples, and with high
concentrations of all other elements, especially the REEs (**Fig. 8**). Analyses of glass shards from the
Maketu tephra can also be identified by their very high Ba values (> 1000 ppm), mid-range Nb values
(between those of Tuhua and the general trend), and much higher concentrations of all other elements
(**Fig. 8**). We also note Er and Lu peaks, which pertain to glasses from the Te Rere tephra, that sit at the
higher concentration levels of the general trend (**Fig. 8; Table 3**), and samples from Ngamotu,
Rotoehu/Rotoiti and Earthquake Flat that sit at the lower overall trace element concentration levels of
the general trend, but with high Ba values (**Fig. 8**). For the tephras where both proximal and distal
samples of glass have been analysed for trace elements, the HFSEs (including Zr, Hf, Th, and Ti) and
LILEs (including Rb, Sr, and Cs) can be used to maintain the heterogeneity between the proximal and
distal samples, whereas the HREE and the LREE tend to have a lower variability (**Fig. 7**).



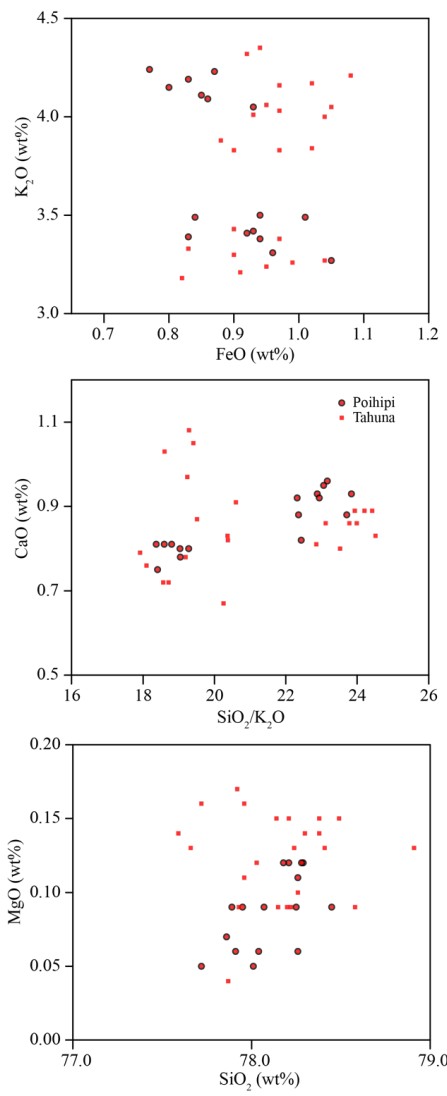


**Figure 6. Selected major element biplots of glass analyses (presented on a normalised basis) of samples from Poihipi and Tahuna tephras (both TVC sourced) that exhibit a bimodal signature. This bimodality is identified as being caused by K₂O concentration (e.g. see Lowe et al., 2008; Shane et al., 2008), and therefore plots with other elements (major or trace) do not show this bimodality. Total iron expressed as FeO.**


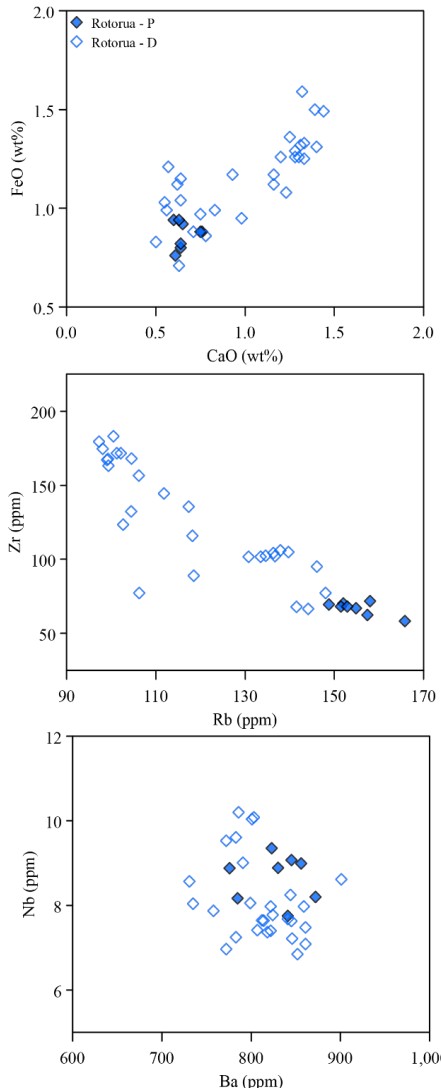

**Fig. 7. Major and trace element biplots showing the glass-shard-derived geochemical relationship of Rotorua (OVC) proximal (P) and distal (D) tephra deposits (presented on a normalised basis, total iron expressed as FeO). Distal deposits may have a signature with lower geochemical variability which overlaps within the spread of the heterogeneous proximal signatures. This variation can often be resolved by using trace element plots of selected elements − see text for discussion.**




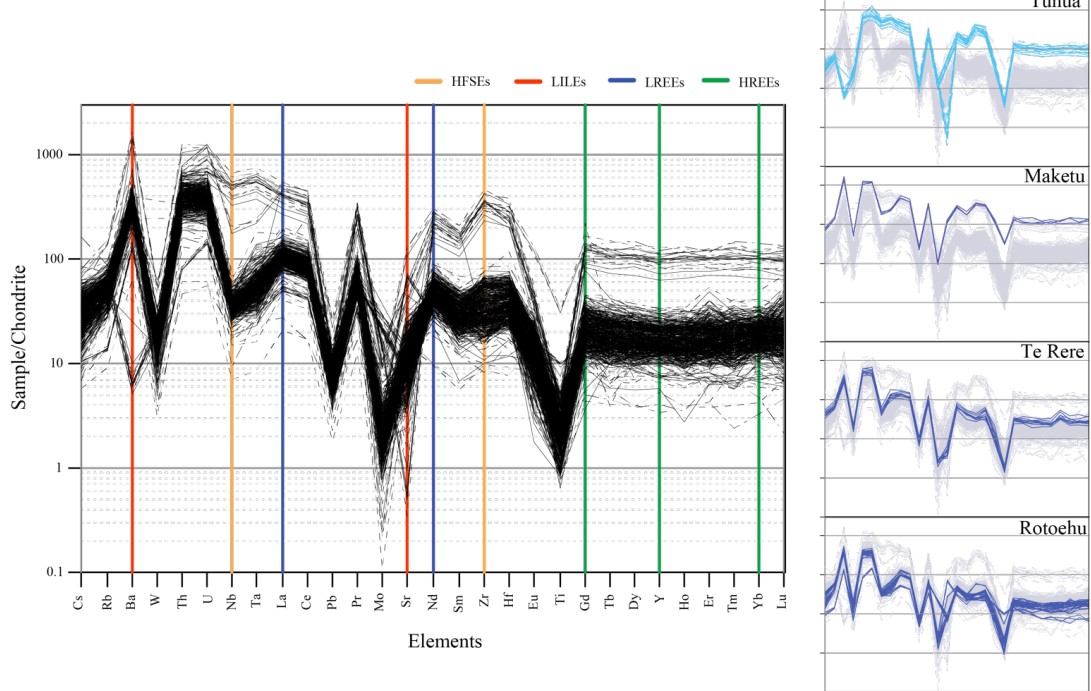

**Figure 8. Chondrite normalised (McDonough and Sun 1995) trace element spider plot for glass analyses for all reference samples. Highlighted are key elements discussed in the text coloured by their characteristics including HFSE, LILE , LREE, and HREE. The**
**full plot is presented to show the density of data with the dominant trend line plus the obvious deviations from this. The samples which correspond to these deviations are shown in the smaller plots at right, including analyses on glass from Tuhua (MI), Maketu, Te Rere, and Rotoehu (OVC) tephras. The Rotoehu Ash signature is also similar to that for the Rotoiti Ignimbrite (which are coeval deposits; Nairn, 1972), the Earthquake Flat tephra (Kapenga VC; Nairn and Kohn, 1973), and the Ngamotu tephra (OVC; Jurado-Chichay and Walker, 2000).**




## 4. Discussion

### 4.1. Distinguishing geochemical characteristics

#### 4.1.1. Major and trace elements in general

In many cases, the major element concentrations in glass are sufficient to allow different tephras to be distinguished, a result consistent with the findings from much previous work both in New Zealand and elsewhere (e.g. Lowe et al., 2017). PCA results for the glass-shard major elements (**Fig. 9**) show that PC1 and PC2 explain 82.7% of the variance within the data. When scaled, concentrations of $TiO_2$,
$Al_2O_3$, CaO, and $SiO_2$ for PC1 (**Fig. 9**), and MnO, $K_2O$, $FeO_t$, and $SiO_2$ for PC2 (**Fig. 9**) are shown to have the highest contribution to the variance and are therefore most appropriate for distinguishing between tephra deposits for the reference dataset as a whole (**Fig. 9**). These major elements, especially CaO, FeO, and $K_2O$, have long been recognised as being useful to distinguish many New Zealand late Quaternary tephras form one another (e.g. Lowe, 1988; Shane, 2000; Alloway et al., 2013), the presence
of $TiO_2$, $Al_2O_3$, and MnO are somewhat unusual. In a number of cases (discussed below) however, major element concentrations are shown to overlap for certain tephra horizons, and thus trace elements and trace element ratios are investigated to provide additional variables to use as discriminants. PCA was also applied to scaled trace elements with the results suggesting that PC1 and PC2 could explain 66.0 % of the variability in the trace element data with Tb, Ho, Dy, Y, Sm, Nd, Tm, Yb, Hf, and Gd the
ten highest contributors to PC1, and Rb, Th, Sr, U, Cs, Ta, Pb, La, Ce, and Pr highlighted as the ten highest contributors the variance of PC2 (**Fig. 10**). Therefore, statistically these trace element concentrations and ratios of these trace elements have the potential to be the most useful in distinguishing the individual tephra horizons when using their glass-shard compositions alone.




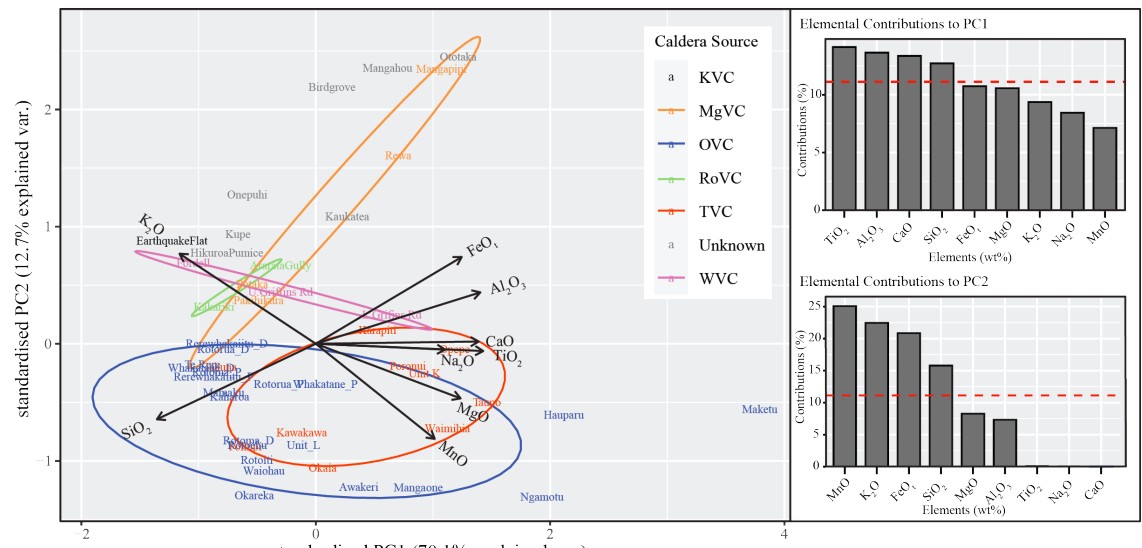

**Figure 9. Results of PCA analysis on all TephraNZ major element reference data for glass (normalised). Data are scaled to allow comparison within the PCA analysis. Tephra names are coloured as per their source centre. PCA analysis was performed in R (see SM1 for R script). Bar plots highlight the top elemental contributions for PC1 and PC2. The red dashed line on the elemental 470 contribution plots indicate the expected average contribution; if the contribution by each element was uniform, the expected value would be 1/no. variables (e.g. 1/9 = ~11%). Therefore, a variable with a contribution larger than this cut off line (~11%) is considered important in the contributing to the component.**



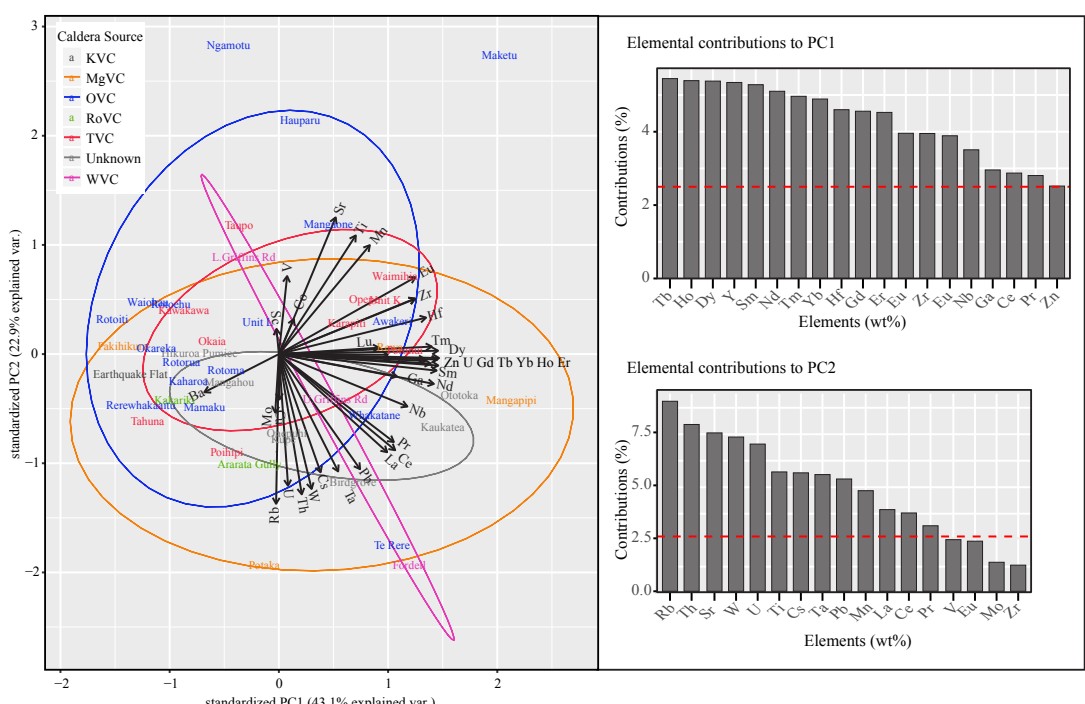

**Figure 10. Results of PCA analysis on all TephraNZ reference trace element data for glass. Data are scaled to allow comparison within the PCA analysis. Tephra names are coloured as per their source centre. PCA analysis was performed in R (see SM1 for R script). Bar plots highlight the top elemental contributions for PC1 and PC2. The red dashed line on the elemental contribution plots indicate the expected average contribution, a variable with a contribution larger than this cut off line is considered important in the contributing to the component.**




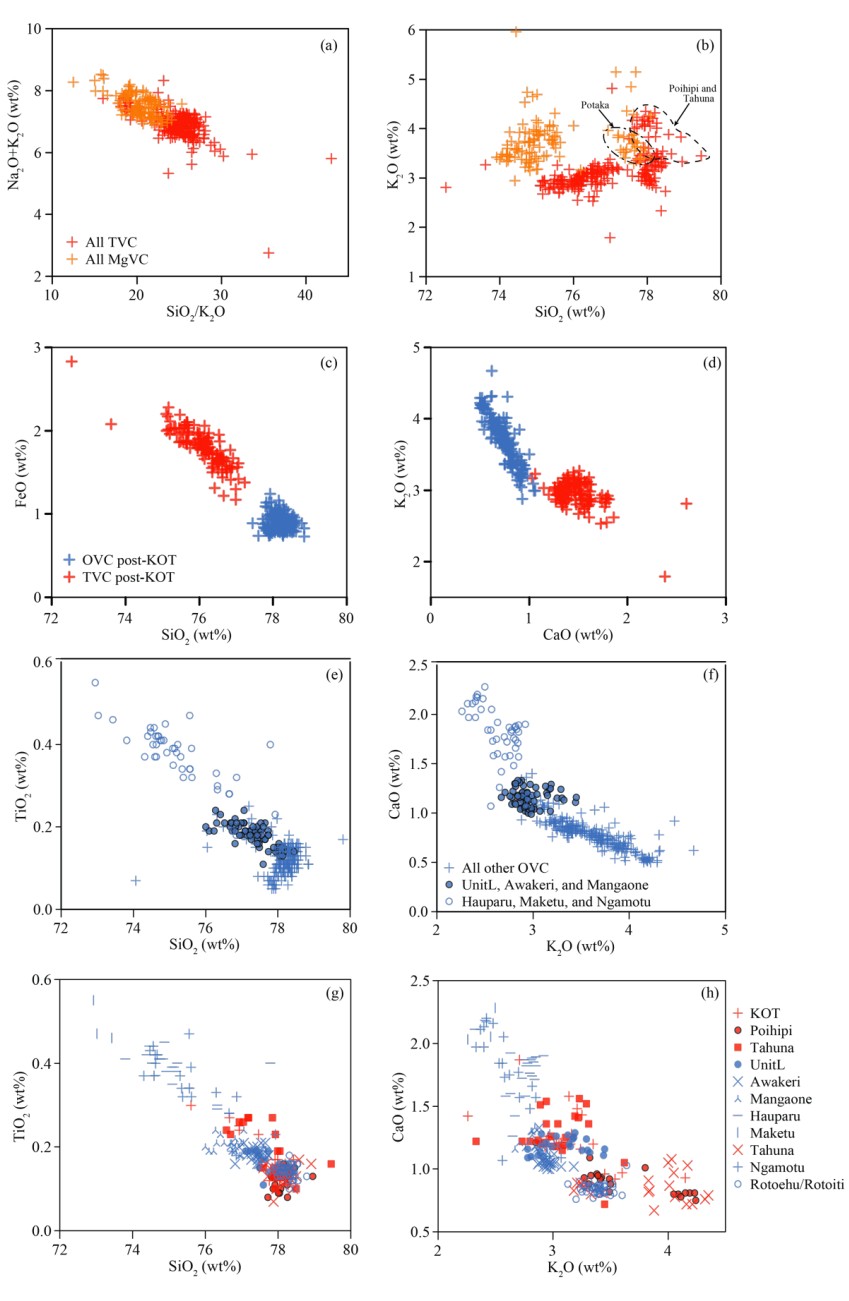



**Figure 11. Major element biplots to distinguish between caldera sources of tephras based on their glass major element compositions** (presented on a normalised basis; total iron is expressed as FeO). (a) & (b) show a comparison between all glass shard analyses for the TVC and MgVC sourced tephras. (c) & (d) Indicate the distinction in glass compositional signature for the eruptives from the OVC and TVC that post-date the Kawakawa/Oruanui (KOT) eruption. (e) & (f) These plots distinguish the glass analyses for tephra from the OVC into component eruptive time periods, with tephra from the Mangaone subgroup (Jurado-Chichay and Walker, 2000; Smith et al., 2002) distinguishable from all other tephra from the OVC. (g) & (h) Show the similarity in the geochemical compositions for the tephra from the OVC and TVC for the eruptions prior to, and including, the KOT eruption. Colours are consistent for each caldera source, symbols are representative of different groups of tephra defined in the keys for each set of plots.

### 4.1.2 Source-specific major and trace elements

The central TVZ contains nine recognised calderas, each with different eruption histories, but all having produced large magnitude/volume tephra-producing rhyolitic eruptives. Some of the calderas are attributed to single caldera collapse events (Rotorua, Reporoa, and Ohakuri), others to composite collapse events that overlap spatially but not temporally (Mangakino and Kapenga), but the majority to multiple collapse events over an extended period of time (Maroa, Okataina, Taupō, and Whakamaru) (**Fig. 1**; Wilson et al., 1995a, 2009; Barker et al., 2021). Although the calderas are mostly discrete in space, evidence from multiple eruptions has shown their plumbing systems may be linked tectonically (e.g. Wilson et al., 2009; Allan et al., 2012). Hence, the ability to trace a tephra deposit to a caldera source through glass-shard geochemistry alone could be challenging.

The results of the PCA analysis suggest that tephra sourced from the TVC can be distinguished from those of a proposed Mangakino source (MgVC) (**Fig. 9**). Using $SiO_2/K_2O$ vs. $Na_2O+K_2O$ ratios, the glass shards of the TVC tephras generally have higher $SiO_2/K_2O$ and lower $Na_2O+K_2O$ ratios in comparison to those of the equivalent oxides for MgVC-sourced tephra (**Fig. 11a**). This information is important, but because of the age differences for the calderas (TVC ~ 0.32 Ma to present, and MgVC ~1.6 Ma to 1.53 Ma and ~1.2 Ma to 0.95 Ma2 1995), the use of this distinction is likely more important for discussions on mantle source dynamics rather than for geochemical correlation of tephra deposits.

Previous studies have suggested that the geochemical characteristics of glass shards from TVC and OVC tephra deposits can be distinguished from after the eruption of the Kawakawa/Oruanui (KOT) to the present day using $fO_2$ of Fe-Ti oxides and minerals (Shane, 1998), pumice and lava compositions (Sutton et al., 2000), and glass chemistry (Froggatt and Lowe, 1990). Our results also show there is a bimodality in the TVC glass-shard data as a whole and that the post-KOT tephra deposits from the TVC and OVC are quite different whereas the pre-KOT tephra from OVC and TVC are similar (**Fig. 11c&d**). Most glass shards erupted after the KOT event from the TVC have low $SiO_2$ (≤ 77 wt.%) and less variable $K_2O$ (~3 wt.%), and higher values for all other major elements in comparison to those of the glass shards erupted from the OVC (**Fig. 11c&d**). In comparison, tephra erupted from the TVC and OVC prior to, and including the KOT, do show a large amount of overlap in their glass geochemical signatures. For OVC, there is a high density of samples that have their $SiO_2$ concentrations at ~78 wt.%;



however, there is a high variability in $SiO_2$ overall, with Maketu, Hauparu, and Ngamotu of the Mangaone subgroup plotting with $SiO_2$ concentrations ≤~76 wt.%, and the remaining Mangaone subgroup samples (Unit L, Awakeri, and Mangaone) clustering at $SiO_2$ = 76-77.5 wt.% (FeO =~1.2 wt.%, $K_2O$ =~2.8 wt.%, $Al_2O_3$ =~13 wt.%, CaO =~1.2 wt.%), a finding consistent with those of Smith

et al. (2005) who divided the Mangaone subgroup into 'old' and 'young' eruptives on the basis of low and high $SiO_2$, respectively, and also unlike the other OVC sourced samples that plot around $SiO_2$ =~77.5-79 wt.%, (FeO = ~0.8-0.9 wt.%, $K_2O$ = 2.75-4.5 wt.%, $Al_2O_3$ =~12-13 wt.%, CaO =~0.5-1.0 wt.%; **Fig. 11c&d**). Analyses from the Rotoehu/Rotoiti tephra deposits plot independently from those of other OVC eruptives for this time period. However, they overlap with those of some TVC-tephra-

derived glass compositions (Poihipi, Tahuna, Okaia, and KOT). The Rotoehu/Rotoiti tephra deposits have a markedly homogeneous geochemical signature, and are also much older than TVC eruptions (**Table 1**). Hence, coupled with the thickness of the deposits, it is likely that a tephra linked to the Rotoehu/Rotoiti eruption would be obvious to distinguish through stratigraphy and age combined with the geochemistry.

The TephraNZ dataset presented here also includes analyses of glass of samples from tephras erupted from the Kapenga Volcanic Centre (KVC; Earthquake Flat eruption), Rotorua Volcanic Centre (RoVC), and Whakamaru Volcanic Centre (WVC). In addition, some older tephra deposits have been recorded in the Whanganui Basin and elsewhere. These are well-known beds but their caldera sources are not yet defined (Alloway et al., 1993; Pillans et al., 1994, 2005; Shane et al., 1996; Rees et al., 2018;

2019). **Figure 12** shows a comparison plot for the data from KVC, RoVC, and WVC with those regions populated by glass data from samples from the OVC, TVC and MgVC sources. Overall, the samples plot with a lower $SiO_2/K_2O$ ratio (≤ ~25), similar to that of the MgVC-sourced tephra, which seems to be indicative of older sources in comparison to the OVC and TVC. The samples potentially linked to RoVC (Bussell, 1986; Bussell and Pillans, 1997) show different geochemical compositions. For

example, Kakariki-tephra-derived glass has slightly higher $SiO_2 \geq 78$ wt.%) in comparison to that of the Ararata Gully tephra ($SiO_2 \leq 77$wt %), suggesting that they are likely derived from different eruptions, but potentially the same source (Mamaku Ignimbrite reportedly has variable geochemical phases; Milner et al., 2003). Glass from the KVC sample (Earthquake Flat tephra) has a very homogeneous signature in the major elements, but a more variable signature in the trace elements, both of which

overlap with OVC- and TVC-source signatures. There is a very large spread for the data from the unknown samples, precluding the ability to specify their source based simply on major and trace elements alone. Nevertheless, their glass compositional signatures are clearly more similar to those of the older MgVC sourced tephra, in comparison to those of the younger TVC and OVC deposits, as would be expected based on their known age range (**Table 1**).



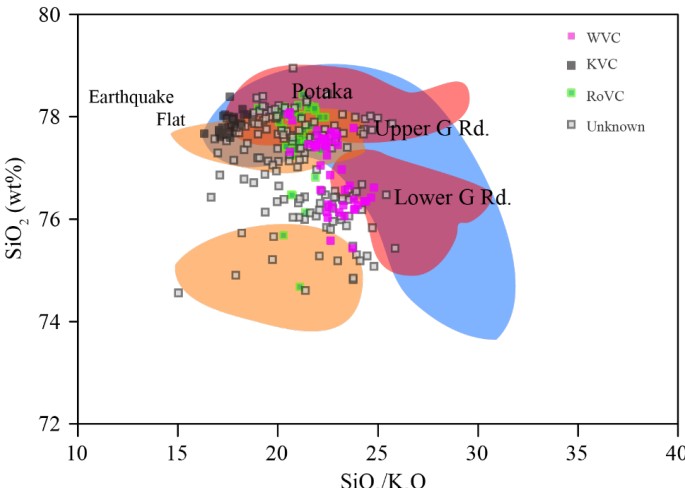

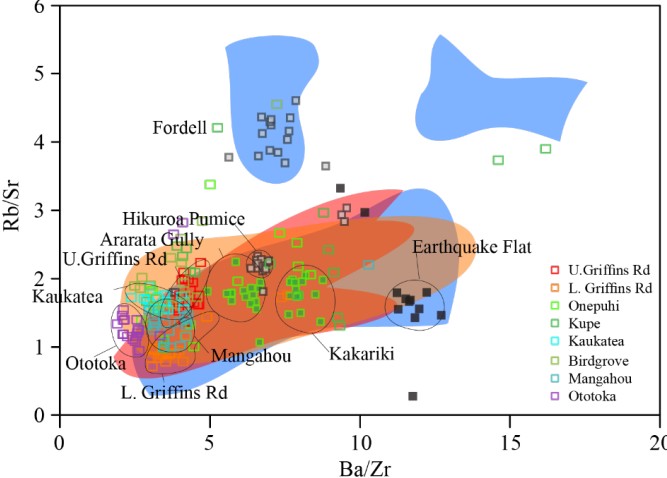


**Figure 12. Major and trace element biplots of indicative elements in glass to show the relationships between the tephras from Mangakino Volcanic Centre (MgVC – orange shaded regions), Taupō Volcanic Centre (TVC – red shaded regions), and Okataina Volcanic Centre (OVC- blue shaded regions), and the tephras from known and unknown sources within the TephraNZ data base.**






### 4.1.3 Homogeneous, heterogeneous, and bimodal samples

Fingerprinting of glass shards for correlation relies on the ability to distinguish between different deposits and therefore a homogeneous signature that is distinct from all other samples is the ideal 'fingerprint'. However, sometimes there is more complexity in the geochemical data and heterogeneity can develop in a tephra deposit through a number of mechanisms (Lowe, 2011):

(1) Variability in the magma body itself (e.g. Nairn, 1992; Nairn et al., 2004; Smith et al., 2004; Kobayashi et al., 2005; Shane et al., 2008; Charlier and Wilson, 2010; Klemetti et al., 2011; Cole et al., 2014);

(2) Proximal vs. distal complexity, linked to (1) (e.g. Manning, 1996; Shane et al., 2003a; Holt et al., 2011);

(3) Post- or syn- depositional reworking (e.g. Schneider et al., 2001)

For example, the heterogeneous signature identified for the Kaharoa tephra agrees with previous findings for this eruption. Nairn et al. (2004) and Sahetapy-Engel et al. (2014) reported that tephra compositional variability within the Kaharoa deposits shows sequential tapping of a stratified magma body coupled with syn-eruptive changes in dispersal patterns. In general, this is likely one of the reasons why some of the proximal tephra deposits analysed in this study have a more variable geochemical signature in comparison to those of their distal counterparts (**Fig. 7**). Although the proximal deposits record the detail in the eruption progression, the distal deposits tend to record the very largest phase of the eruption (e.g. Walker, 1980) but differences can be expected to occur according to the azimuths of wind direction during an eruption and the number and degree of interconnectedness of magma bodies involved in the eruption (e.g. Walker, 1981; Kilgour and Smith, 2008; Sahetapy-Engel et al., 2014; Storm et al. 2014; Rubin et al., 2016).

The tephrochronological principle is much more likely to utilise distal unknown deposits, and therefore we suggest that using the distal signature (or signatures) maybe more appropriate for correlation. In general, distal tephras are more chemically homogeneous – but with some notable and well-documented exceptions – and this attribute therefore allows them to be traced over large areas (Manning, 1996). Alternatively, the identification of heterogeneity or bimodality in distal tephras, once recognised, can be an additional useful characteristic for fingerprinting (e.g. Shane et al., 2003a, 2008; Lowe et al., 2017). These statements, however, rely on the tephra being identified as a primary deposit, and not reworked. Reworking is commonly seen in paleofluvial deposits, for example those in the Whanganui Basin, and in other environments prone to mixing such as in surficial soils. This reworking can mix tephra from multiple eruptions, and can cause highly variable glass chemistry within a single deposit (e.g. Shane et al., 2005, 2006). Fluvial reworking can be commonly identified by sedimentary structures within the deposit, for example, ripples or cross bedding indicative of fluvial transport and deposition (e.g. Shane, 1994; Schneider et al., 2001), over thickening of deposits (e.g. Vucetich and





Pullar, 1969; Lowe, 2011), or through shard morphology, for example anomalously large shards or rounding of shards (e.g. Leaphy, 1997).

Heterogeneous signatures (where the standard deviation of the analyses is greater than the analytical error) in major element compositions were identified for ten of the tephra deposits: Kaharoa, Taupō Y5 Proximal (P), Whakatane P, Hauparu, Maketu, Ngamotu, Fordell, Onepuhi, Birdgrove, and Ototoka. Our data show that for some samples, specific trace elements and trace element ratios have lower geochemical variability (**Fig. 13a**). The elements that work best to separate out the individual units within a deposit with a heterogeneous signature reflect the minerals that have formed during

fractional crystallisation of the melt. Because of this, different elements or element ratios work for different tephras. For example, for Kaharoa, Sr acts to effectively reduce the variability of the signature, whereas for Taupō Sr can be used to maintain the heterogeneity in the sample (**Fig. 13a)**.

Bimodality was identified for four of the tephra horizons analysed: Rotorua (OVC), Rerewhakaaitu (OVC), Poihipi (TVC), and Tahuna (TVC). For all four of these, $K_2O$ concentration

causes the bimodality, and therefore trace elements with similar chemical properties reinforce the bimodality (for example, LILEs Rb, Sr, and Cs; HFSEs Zr, or REE Eu), whereas most other trace elements do not show this bimodal signature (**Fig. 13b**).

### 4.2. Indistinguishable tephras

Euclidean similarity coefficient (ESC) analysis was used on all glass-shard reference data for

tephras from Rotoiti/Rotoehu to Kaharoa in addition to the PCA and geochemical investigation to determine those samples that have indistinguishable element concentrations at similar ages (**Fig. 14**). **Figure 14** shows similarities (similarity coefficient values (SC) are observed in the major element signatures between Waimihia and Unit K (SC=0.113); Mamaku and Rotoma-D (SC=0.170), Rotoiti/ Rotoehu and Rotoma-D (SC= 0.105 and 0.126, respectively); KOT and Okaia (SC=0.141); KOT and

Unit L (SC=0.137); and Poihipi and Tahuna (SC=0.120). When the key trace element are analysed, only Poihipi and Tahuna (SC = 15) and Mamaku and Rotoma-D (SC=3) come up with significantly low similarity coefficients, hence suggesting that these samples will be indistinguishable in both major and trace elements. When trace element ratios are run through the SC analysis, Waimihia and Unit K (SC=6), Waimihia and Poronui (SC=9), Unit K and Poronui (SC=5), and Karapiti and Waimihia

(SC=7) show significantly low similarity coefficients. In addition, when simple geochemical assessment is applied, similarities are observed between Taupō and Waimihia, and Waiohau, Rotorua, and Rerewhakaaitu (**Table 5**).

These results suggest that for Poihipi and Tahuna, and Mamaku and Rotoma tephras, trace element ratios in glasses could enable them to be distinguished. **Figure 15a** shows that for Poihipi and

Tahuna the best separation (although some overlap remains) is seen in the ratios La/Yb vs. Ba/Y; in addition, Tahuna also shows a bimodality in Ba/Th ratio which is not seen for Poihipi. For Rotoma and



Mamaku, the tephras can be separated (although some overlap remains) using Ba/Th vs. Rb/Sr and Rb/Zr vs. Rb/Sr (**Fig. 15b**). Rotoma and Rotoehu/Rotoiti are very similar in their glass-shard major elements, but can be distinguished using specific, but a wide range, of trace elements (**Fig. 15c**). They

are also very different in age, hence should not be too difficult to distinguish on the basis of stratigraphy or dating.

Waimihia and Unit K (Taupō Subgroup) tephras are very difficult to distinguish, and their relative similarity in age ($3382 \pm 50$ and $5088 \pm 73$ cal. yr BP, respectively; Lowe et al., 2013) and mineralogy could  see them misidentified if dates were unavailable or imprecise. Geochemical

investigation beyond the PCA and SC analyses of glass shows that Lu, Sc, Mn, and Co can be used to geochemically distinguish these two tephras (**Fig. 15d**), indicative of fractional crystallisation of differing amounts of clinopyroxene, plagioclase, and amphibole during the eruptive events. Although not identified by the SC analysis directly, Poronui (11,195- 51 cal yr BP) and Karapiti ($11,501 \pm 104$ cal. yr BP) tephras also have comparable age, geochemistry, and mineralogy; thus using major, trace,

and trace element ratios these two tephras remain indistinguishable. Glass shards from the three Holocene tephras, Waimihia, Poronui, and Karapiti, also have very similar trace element and trace-element ratios but, as for Waimihia and Unit K, they can be distinguished with Lu, Sc, Mn, and Co, where Waimihia has higher Sc, Lu, and Mn, but lower Co in comparison to those of the Poronui and Karapiti tephras. They can also be distinguished simply with a biplot of FeO vs. CaO, or $Na_2O+K_2O$ or

$SiO_2/K_2O$, or $SiO_2$, where the Waimihia samples in general have lower FeO, $Na_2O+K_2O$, $SiO_2/K_2O$, and higher CaO, and $SiO_2$ in comparison to the equivalent values for Poronui and Karapiti samples (**Fig. 15e, Table 5**).

Geochemical investigation and PCA analysis also highlights the similarity of the Waiohau, Rotorua, and Rerewhakaaitu tephras. There is added complexity with these samples as we have both

proximal and distal deposits to compare, where, as discussed previously, the proximal samples will likely be more heterogeneous. Glass analyses of the Waiohau tephra show it can be distinguished  from those for the Rotorua and Rerewhakaaitu tephras using a range of trace elements and trace-element ratios. In addition, the Rotorua and Rerewhakaaitu tephras are observed to be bimodal for some elements. The Waiohau also has different mineralogy from that of Rotorua and Rerewhakaaitu tephras

(Froggatt and Lowe, 1990; Lowe et al., 2008). Conversely, the Rotorua and Rerewhakaaitu tephras are usually indistinguishable in geochemistry and mineralogy, and therefore accurate dating and stratigraphic super-positioning would have to be relied upon to distinguish them(**Fig. 15f, Table 5**).

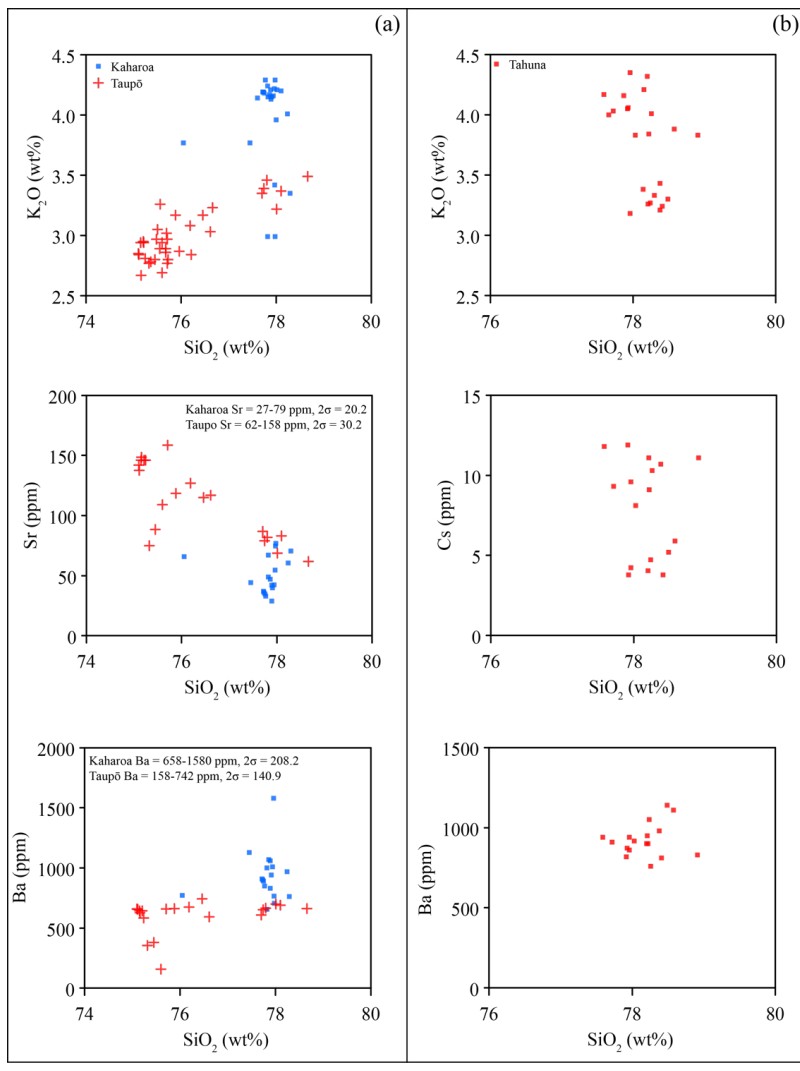

**Figure 13.** Biplots to show examples of how trace elements in glass enable manipulation of heterogeneous and bimodal geochemical data. Panel (a) shows analyses of glass from Kaharoa and Taupō tephras, both of which show a heterogeneous signature with most major elements (presented on a normalised basis). Sr has a low variability for Taupō, but does not for Kaharoa tephra, conversely, Ba has a low variability for Kaharoa, but does not for Taupō. Panel (b) shows the bimodal signature created for Tahuna tephra using $K_2O$ composition; this is also seen for Cs, but is not for Ba.




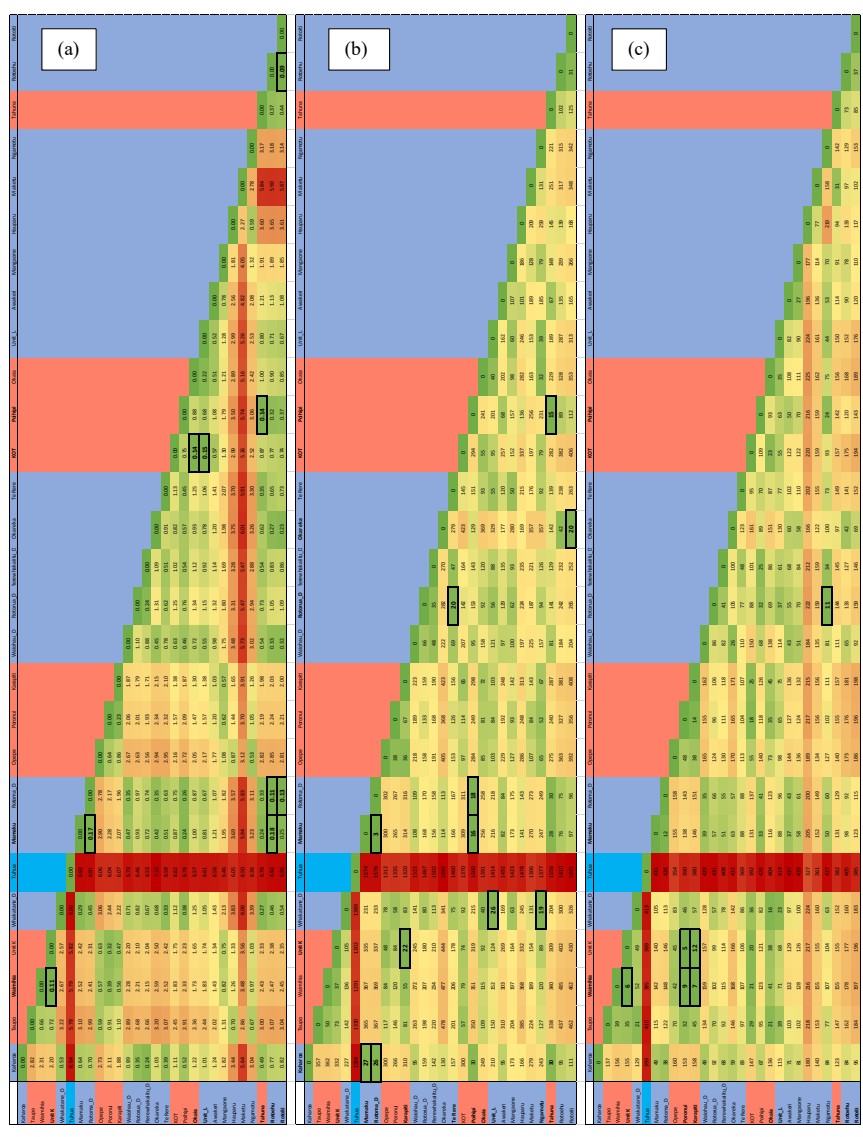

**Figure 14. Results of Euclidean similarity coefficient (SC) calculations for major (a), trace (b), and trace ratio (c) concentrations in glass from all the tephras analysed. See Supplementary Material for R code used for these calculations. Colour coding shows SC values: green shows the smaller the value (similar compositions); through to red showing the larger the values (different compositions). The lowest values, and hence the most similar tephras compositionally, based on all geochemical data, are highlighted with bold outlines and text.**

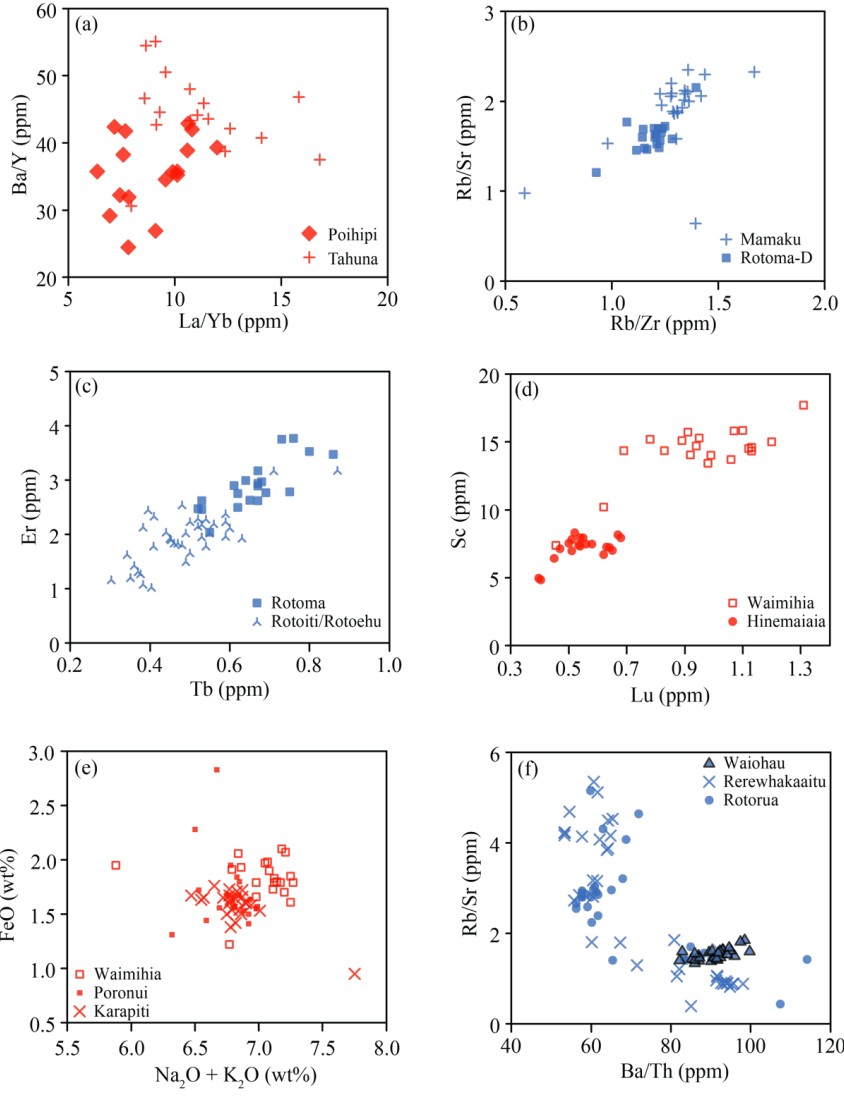

**Figure 15.** Biplots for glass analyses for specific tephras which have very similar compositions and similar ages (see text for discussion and Table 4 for alternative elements). Plots show examples of the elements that enable these tephras to be separated (a) Poihipi and Tahuna (from TVC); (b) Mamaku and Rotoma (from OVC); (c) Rotoma and Rotoiti/Rotoehu (from OVC); (d) Waimihia and Unit K (from TVC); (e) Waimihia, Poronui and Karapiti – note that Poronui and Karapiti are indistinguishable using glass-chemistry; (f) Waiohau, Rerewhakaiitu and Rotorua – note that Rerewhakaiitu and Rotorua are indistinguishable using glass chemistry. All major element data presented on a normalised basis, and total iron is expressed as FeO.



KOT, Okaia, and Unit L (Mangaone Subgroup) show indistinguishable major elements in their constituent glass shards, and very similar trace elements. The TephraNZ samples have been compared to existing published data and are complementary in major elements (e.g. Sandiford et al., 2002; Shane et al., 2002; Smith et al., 2002; 2005; Lowe et al., 2008; Allan et al., 2008; Molloy, 2008; 20). This is the first time trace element glass data have been published for Unit L and Okaia tephras. Our results show that Unit L glass shows bimodality in Rb/Zr, Ba/Th, Ce/Th and Y/Th and in this way it can be distinguished from the KOT and Okaia tephras (**Table 5**).

### 4.3. Proposed future research

This foundation dataset, derived in a formalised way, is unique in New Zealand and provides researchers with new avenues of research. It is our hope that the foundation dataset can be improved and expanded with analyses of other known deposits, and that a subsidiary catalogue of accurately correlated geochemical samples can be added to bolster the dataset. As noted earlier, it is beyond the scope of this paper to dive too deeply into the detail of the data but we feel that it will provide the basis for countless projects in the future. Below we highlight some of the current gaps which we think would benefit from further research.

#### 4.3.1 Further statistical analysis

We have applied simple ordination and statistical analyses to this dataset; however, we believe that further rigorous statistical analysis could be applied. Firstly, the analyses we present in this publication have been applied to mean values for each of the tephra samples (e.g. data from **Table 3**); there is no reason why these simple tests could not be applied to the full dataset, using all the individual values analysed for each sample. Secondly, for simplicity we chose to split up the assessment of major and trace elements, these could be run concurrently. Third, we chose very basic tests (PCA and ESC) to fit with our requirements, however there is likely some more appropriate statistical test that could be applied to get the most out of this exceptional dataset. For example, (extended) Canonical Variates Analysis (CVA); applying CVA to PCA results could determine optimal discrimination between multivariate data for single tephra deposits. This discrimination will increase the ability to identify an unknown tephra based on its similarity to known signatures plotted in multivariate space (e.g. discriminant function analysis; Tyron et al., 2009; 2010; Lowe et al., 2017; Bolton et al., 2020).



### 4.3.2 Whanganui Basin correlatives

A number of the tephras reported in this research were sampled from the Whanganui Basin, an uplifted Plio-Pleistocene basin margin sequence that preserves as many as 45-superposed cyclothems deposited since ~3 Ma (Naish and Kamp, 1997; Naish et al., 1996, 2005; Carter and Naish, 1998; Carter et al., 1999; Pillans, 2017; Grant et al., 2018, 2019; Tapia et al., 2019). The tephra deposits within the basin contribute to the robust chronological framework that has been constructed for this region (Seward, 1976; Beu and Edwards, 1984; Alloway et al., 1993; Naish and Kamp, 1995; Shane et al.,

1996; Saul et al., 1999; Pillans et al., 1994, 2005; Naish et al., 1996, 2005; Rees et al., 2018, 2019). These tephras also record a critical time in New Zealand's volcanological history – the transfer between activity from the Coromandel Volcanic Zone to the Taupō Volcanic Zone (Briggs et al., 2005). Deposits from this period are generally poorly exposed at source, and thus distal tephras could provide an insight into the eruptive history, geochemical evolution, and potentially even caldera evolution during this

period (Houghton et al., 1995). Most of the tephras reported in this research are well known and well dated, which is why they were included in the study. However, most do not have a known source caldera or source eruptives, or have only been variably correlated to other deposits in New Zealand (e.g. Lowe et al., 2001; Pearce et al., 2008). There are also numbers of tephra deposits in the Whanganui Basin that have yet to be studied, and thus a research project that is tephra focused, rather than using it

as an accessory to a different line of enquiry, is timely.

### 4.3.3 IODP and ODP correlatives

  At present there is a wealth of information that has yet to be fully investigated in the tephra deposits in ODP Leg 181 Sites 1122, 1123, 1124, 1125 (Carter et al., 2003, 2004; Alloway et al., 2005;

Allan et al., 2008) and IODP Expedition 372 and 375 sites U1517 and U1520 (Pecher et al., 2018; Saffer et al., 2018). Pioneering work includes that undertaken by Watkins and Huang (1977) and Nelson et al. (1985) and findings from more 'local' marine coring expeditions include those reported by Shane et al. (2006). The new reference material built by this project will allow more definitive identification and correlation of tephras within these cores, specifically post-2 Ma. However, the reports currently

published on these deposits suggest that there are many more tephra deposits to be found in these marine and offshore sites than we have in the TephraNZ dataset (Carter et al., 2003; Alloway et al., 2005; Holt et al., 2010, 2011). The TephraNZ dataset can provide a formalised correlation framework from which other unknown deposits can be determined, characterised, and integrated into a holistic tephrostratigraphic reconstruction. Allan (2008) and Allan et al. (2008) reported the major and trace

element geochemistry of glass shards for tephra deposits dating from ~1.65 Ma in the ODP 1123 core. They also give orbitally-tuned ages for these tephras. However, of the 38 identified tephras only seven were correlated to onshore equivalents. In addition, Alloway et al. (2005) reported over 100 tephra





layers in the four ODP Leg 181 cores, dating back through orbital tuning (astrochronology) to 1.81 Ma.
Using major element chemistry of glass, 13 tephras were correlated to equivalent onshore tephras
including KOT, Omataroa, Rangitawa/Onepuhi, Kaukatea, Kidnappers-B and -A/Potaka, Unit
D/Ahuroa, Ongatiti, Rewa, Sub-Rewa, Pakihikura, Ototoka and Table Flat. Analyses of glass from
some of these are currently not in the TephraNZ database but could be easily added if the appropriate
reference samples were available and capacity to analyse them were available. Alloway et al. (2005)
reported an additional six tephra deposits that are correlated between the cores, but not to onshore
equivalents, leaving potentially ~81 tephra horizons within the ODP cores that are uncorrelated. This
information could provide a detailed investigation into the timing and evolution of the TVZ eruptions
that is unobtainable from onshore deposits.

### 4.3.4 Mineral compositions

The TephraNZ reference dataset is only populated by glass major and trace element analyses at
present. This is because glass geochemistry is one of the most frequently used and accessible tools for
tephra correlation. Aerodynamic sorting of tephra componentry through transportation adds to the
favourability of glass shards as the dominant tool because glass shards tend to be the only phase that is
found at both proximal and distal sites. However, previous New Zealand-based studies have specified
how mineral assemblages and their geochemical compositions can be used to distinguish certain tephras
and their source (e.g. Nairn and Kohn, 1973; Lowe, 1988; Froggatt and Lowe, 1990; Froggatt and
Rogers, 1990; Shane, 1998; Shane et al., 2003b; Allan et al., 2008; Lowe et al., 2008; Lowe, 2011). For
example, the mineral cummingtonite, where predominant, is a known identifier for tephras from the
Haroharo complex of the OVC (Whakatane, Rotoma, Rotoehu/Rotoiti (**Table 4**); Ewart, 1968; Lowe,
1988; Froggatt and Lowe, 1990). At present, ferromagnesian mineralogical assemblages (following
Froggatt and Lowe, 1990; Smith et al., 2005; Lowe et al., 2008) for all the TephraNZ samples younger
than and including Rotoehu/Rotoiti have been published (see Table 4). Extending this tabulation to
include the older samples would add another useful criterion to the correlation toolbox .

Additionally, the fractional crystallisation of plagioclase, biotite, amphibole, zircon, hydrous
mineral phases, or Fe-Ti oxides has been shown to be the key impactor on the trace element chemistry
(Shane, 1998; Allan, 2008; Turner et al., 2009, 2011). Thus the prevalence of these minerals is also an
important potential fingerprinting tool. The information on the mineralogy of the tephras is not only
useful for fingerprinting but also can be used in determining the characteristics of the magma source
components, and potentially provide estimates for the temperature, pressure, and oxidation states of the
magmatic system before eruption (e.g. Lowe, 1988, 2011; Shane, 1998). Thus, this information can
allow hypotheses to be developed on the reactivation and triggering of these large-scale eruptions, an
important step for hazard and risk monitoring.



### 4.3.5 The New Zealand tephra "Bermuda Triangle"

At present the TephraNZ database is very well populated for samples from the Rotoiti/Rotoehu
through to Kaharoa eruption. It also has a high number of samples, but not an exhaustive list, from
Mamaku ignimbrite (~0.22-0.23 ka) to the Hikuroa Pumice (2 Ma). There is a stark deficit in tephras
between the Rotoiti/Rotoehu eruption and Mamaku ignimbrite (**Table 1**). This ~150 kyr gap in the
volcanic record (~220 ka to 45 ka) is intriguing as there is proximal evidence for activity during this

period. For example, Rosenberg et al. (2020) reported the occurrence of volcanic formations in cores
form the Taupō region in the age range of ~168 to 92 ka, including the Huka Falls formations,
Racetrack rhyolites, and the Te Mihi rhyolites. Tephra deposits, in some cases strongly weathered
successions of multiple units broadly lumped together  as a 'formation', such as the so-called Hamilton
Ash Formation, have been reported during this time period both terrestrially and in marine and

lacustrine sediment cores (Ward, 1967; Pain, 1975; Vucetich et al., 1978; Iso et al., 1982; Froggatt,
1983; Manning, 1996; Lowe et al., 2001; Newnham et al., 2004; Allan et al., 2008; Briggs et al., 2006;
Lowe, 2019; B. Laeuchli *pers. comms.* 2020). However, at present the authors are not aware of a
detailed, up-to-date study into the primary compositions of these tephra deposits. The key deposits
identified during this time period include (but are not limited to) Kaingaroa Ignimbrite (~0.18 Ma;

Froggatt, 1983), Tablelands Tephra Formation (~0.21–0.18 Ma; Iso et al., 1982, 0.34-0.39 Ma;
Manning, 1996), Hamilton Ash Formation (0.125–0.34 Ma; Lowe, 2019); Kutarere tephra (= Mamaku
ignimbrite 0.22-0.23 Ma; Shane et al., 1994; Houghton et al., 1995; Black et al., 1996; Tanaka et al.,
1996; Milner et al., 2003), Kukumoa Subgroup (~0.22-0.05 Ma; Manning, 1996), and Tikotiko Ash
(~0.125 ka; Lowe, 2019). A number of these studies are outdated, and with improved methodologies

(major and trace element analysis, potentially of melt inclusions where preserved, dating techniques,
and other measures to help construct time frames such as via phytolith studies to determine glacial vs
interglacial periods) it could be timely to further investigate this period of (apparent) deficit.

### 5. Conclusions

     Major and trace element geochemical compositions of glass shards for a large suite of
prominent, widespread New Zealand  rhyolitic tephras have been analysed systematically and published
for the first time as "TephraNZ". TephraNZ is a foundation dataset for collating geochemical data about
New Zealand tephras. The foundation reference dataset is made up of known deposits that have their
ages quantified through independent methods, or are from the type sites where tephras were first
defined, or well-documented reference sections. Detailed methodology is reported to allow subsequent
research to acquire comparable data to those in this database. Principal component analysis indicates



that for the TephraNZ database, as a whole, major elements CaO, $TiO_2$, $K_2O$, and $FeO_t$ are responsible for the variability in PC1 and PC2 space. For trace elements, Tb, Ho, Dy, Y, Sm, Nd, Tm, Yb, Hf, Gd and Rb, Th, Sr, U, Cs, Ta, Pb, La, Ce, Pr are responsible for ~ 69% of the variability, and trace element ratios, Ba/Zr, Ba/Hf, Ba/Eu, Zr, Nb, Rb/Zr, Ba/Y, Zr/Th, Zr/ Yb, Nb/Y, Zr/Y, and Ba/Th, Ce/Th Y/Th,

Ba/La, Ba/Ce, Zr/Th and Sr/Nb are responsible for ~73 % of the variance. Euclidean similarity coefficients can also be used to distinguish between some geochemically similar glass analyses. However, further detailed geochemical investigation is required to distinguish others. Geochemically indistinguishable tephras (on the basis of both major and trace element glass-shard compositions) are identified as Taupō and Waimihia; Poronui and Karapiti; Rotorua and Rerewhakaaitu; and KOT and

Okaia. Only Poronui and Karapiti are noted as entirely indistinguishable, with other methods of characterisation listed as alternative options, including mineralogy, age, and stratigraphic relationships.

### 6. Author contribution

JLH and RJW designed the project. DJL and BJP contributed samples from previous field campaigns, and DJL provided guidance on new and existing field locations for sample collection. JLH and JEB

undertook the field work, lab work, analysis and data reduction. ABHR advised on statistical analysis and R-coding. LA supervised and helped JEB develop LA-ICP-MS analysis and data reduction. FT supervised and helped JEB develop sample mounting and polishing procedures. JLH wrote the manuscript with contributions from all co-authors.

### 7. Competing interest

The authors declare that they have no conflict of interest.

### 8. Acknowledgements

This research was funded partly through the Victoria University of Wellington (VUW) Summer Scholarship programme of which JEB was the recipient (project code 136, 2019-2020), with a matched contribution from JLH's Marsden Fast start. JLH and RJW are also funded through JLH's Marsden Fast

Start project (Te Pūtea Rangahua a Marsden) from the Royal Society of New Zealand (Royal Society Te Apārangi) contract MFP-VUW1809. Some of the field work for tephra collection and analysis was supported by DJL's (2011-2014) Marsden Fund (Te Pūtea Rangahua a Marsden) from the Royal Society of New Zealand (Royal Society Te Apārangi) contract UOW1006. The paper is an output of the Commission on Tephrochronology (COT) of the International Association of Volcanology and

Chemistry of the Earth's Interior (IAVCEI). The authors would also like to thank James Crampton (VUW) and Grace Frontin-Rollet (NIWA) for statistical discussion and advice.





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



**Table 1. Overview of the foundation tephra included in TephraNZ database**

| Tephra name | Alternative name(s) | Caldera source | Age (cal. yr BP, 2sd) (unless otherwise shown) | Dating method* | Age reference | Proximal/distal | Easting | Southing | Site number (Fig. 1) | Site description (Fig. 1) |
|---|---|---|---|---|---|---|---|---|---|---|
| Kaharoa | | Okataina | 636 +/- 12 | C14 wiggle match | 1 | P | 176°31'81.7" | 38°18'56.8" | 1 | Ash Pit Road |
| Taupō | Unit Y | Taupo | 1718 +/- 10 | C14 wiggle match | 2, 3 | P | 176°11'58.5 | 38°44'50.5 | 2 | SH 5 |
| Waimihia | Unit S | Taupo | 3382 +/- 50 | C14 Bayes model 1 | 4 | D | 177°18'24.5" | 38°68'35.8" | 3 | Kaipo Bog |
| Unit K (Taupō series) | | Taupo | 5088 +/- 73 | C14 Bayes model 1 | 4 | D | 177°18'24.5" | 38°68'35.8" | 3 | Kaipo Bog |
| Whakatane | | Okataina | 5542 +/- 48 | C14 Bayes model 1 | 4 | D | 177°18'24.5" | 38°68'35.8" | 3 | Kaipo Bog |
| | | | | | | P | 176°28'82.3" | 38°17'90.5" | 1 | Ash Pit Road |
| Tuhua | | Mayor Island | 7637 +/- 100 | C14 Bayes model 2 | 5 | D | 177°18'24.5" | 38°68'35.8" | 3 | Kaipo Bog |
| Mamaku | | Okataina | 7992 +/- 58 | C14 Bayes model 1 | 4 | D | 177°18'24.5" | 38°68'35.8" | 3 | Kaipo Bog |
| Rotoma | | Okataina | 9472 +/- 40 | C14 Bayes model 1 | 4 | P | 176°28'82.3" | 38°17'90.5" | 1 | Ash Pit Road |
| | | | | | | D | 177°18'24.5" | 38°68'35.8" | 3 | Kaipo Bog |
| Opepe | Unit E | Taupo | 10,004 +/- 122 | C14 Bayes model 1 | 4 | D | 177°18'24.5" | 38°68'35.8" | 3 | Kaipo Bog |
| Poronui | Unit C | Taupo | 11,195 +/- 51 | C14 Bayes model 1 | 4 | D | 177°18'24.5" | 38°68'35.8" | 3 | Kaipo Bog |
| Karapiti | Unit B | Taupo | 11,501 +/- 104 | C14 Bayes model 1 | 4 | D | 177°18'24.5" | 38°68'35.8" | 3 | Kaipo Bog |
| Waiohau | | Okataina | 14,018 +/- 91 | C14 Bayes model 1 | 4 | D | 177°18'24.5" | 38°68'35.8" | 3 | Kaipo Bog |
| | | | | | | P | 176°19'29.3" | 38°10'11.8" | 4 | RNL pumice quarry |
| Rotorua | | Okataina | 15,738 +/- 263 | C14 Bayes model 1 | 4 | D | 177°18'24.5" | 38°68'35.8" | 3 | Kaipo Bog |
| | | | | | | P | 176°19'29.3" | 38°10'11.8" | 4 | RNL pumice quarry |
| Rerewhakaaitu | | Okataina | 17,209 +/- 249 | C14 Bayes model 1 | 4 | D | 177°18'24.5" | 38°68'35.8" | 3 | Kaipo Bog |
| | | | | | | P | 176°19'29.3" | 38°10'11.8" | 4 | RNL pumice quarry |
| Okareka | | Okataina | 21,858 +/- 290 | C14 Bayes model 3 | 4 | P | 176°19'29.3" | 38°10'11.8" | 4 | RNL pumice quarry |
| Te Rere | | Okataina | 25,171 +/- 964 | C14 Bayes model 3 | 4 | P | 176°19'29.3" | 38°10'11.8" | 4 | RNL pumice quarry |
| Kawakawa/Oruanui | | Taupo | 25,358 +/- 162 | C14 Bayes model 3 | 6 | P | 175°53'31.2" | 38°37'19.2" | 5 | Poihipi Road |
| Poihipi | | Taupo | 28,446 +/- 670 | C14 Bayes model 3 | 4 | P | 176°10'50.5" | 38°02'36.2" | 6 | Oturoa Road |
| Okaia | | Taupo | 28,621 +/- 1428 | C14 Bayes model 3 | 4 | P | 175°53'31.4" | 38°37'12.9" | 5 | Poihipi Road |
| Unit L (Mangaone Sgp series) | | Okataina | 30,900 +/- 1500 | Bayes model | 9 | P | 176°08'22.2" | 38°05'39.5" | 7 | Dansey Road |
| Awakeri | Unit J | Okataina | 30,900 +/- 560 | C14 + ZDD | 9 | P | 176°43'16.9" | 38°01'43.9" | 8 | Bowditch Quarry SH30 |
| Mangaone | Unit I | Okataina | 31,500 +/- 520 | C14 + ZDD | 9 | P | 176°43'16.9" | 38°01'43.9" | 8 | Bowditch Quarry SH30 |
| Hauparu | Unit F | Okataina | 36,165 +/- 450 | C14 | 8 | P | 176°14'43.9" | 37°46'13.1" | 9 | Little Waihi Road, Maketu |
| Maketu | Unit D | Okataina | 36,100 +/- 440 | C14 + ZDD | 9 | P | 176°14'43.9" | 37°46'13.1" | 9 | Little Waihi Road, Maketu |
| Tahuna | | Taupo | 38,400 +/- 1700 | Bayes model | 9 | P | 176°44'56.7" | 37°56'58.1" | 10 | Braemar Road |
| Ngamotu | Unit B | Okataina | 39.0 ka | | 7 | P | 176°44'56.7" | 37°56'58.1" | 10 | Braemar Road |
| Earthquake Flat Ig | | Kapenga | 45,160 +/- 2900 | (U-Th)/He | 10 | P | 176°15'02.9" | 38°16'52.9" | 13 | Tumunui Road |
| Rotoehu Ash | | Okataina | 45,170 +/- 3300 | (U-Th)/He | 10 | P | 175°53'04.2" | 37°59'47.5" | 11 | Tapapa Road |
| Rotoiti Ignimbrite | | | | | | | 176°43'33.8" | 37°52'42.2" | 12 | Mimiha Stream, SH2 |
| Ararata Gully 318 | ? Mamaku Ig | ? Rotorua | 0.235 Ma | MIS7c/b boundary | 11 | D | 174°21'26.9" | 39°33'17.2" | 14 | Ararata Gully |
| Kakariki 272 | < Rangitawa = ? Mamaku | | 0.235 - 0.3 Ma | Stratigraphy | 12, 13 | D | 175°44'51.0" | 40°09'52.4" | 15 | Rangitikei Valley |
| Fordell 449 | ? Whakamaru Ig | ?Whakamaru | 0.31 Ma | MIS9a | 13 | D | 175°15'17.8" | 39°56'33.3" | 16 | Kauangaroa Road |
| Rangitawa | Mt Curl, Mangaroa, Ohariu, Haywards ash, Lower Finnis Road, Ohinewai Tephra, Hamilton Ash H1 | Whakamaru | 345 +/- 12 ka (1sd) | glass-ITPFT | 14 | D | 175°19'01.9" | 37°47'26.5" | 17 | Univ of Waikato, Hamilton |
| Upper Griffin Road 307 | ? Whakamaru Ig | ? Whakamaru | 0.3 - 0.4 Ma | Stratigraphy | 13 | D | 175°13'18.4" | 39°58'02.4" | 16 | Kauangaroa Road |
| Lower Griffin Road 309 | ? Whakamaru Ig | | 0.3 - 0.4 Ma | Stratigraphy | 13 | D | 175°13'18.4" | 39°58'02.4" | 16 | Kauangaroa Road |
| Onepuhi 267 | | Unknown | 0.57 Ma | Astronomical | 15 | D | 175°28'38.9" | 40°04'32.2" | 15 | Rangitikei Valley |
| Kupe 481 | | Unknown | 0.63 +/- 0.08 Ma (1sd) | glass-ITPFT | 15, 16 | D | 175°18'09.5" | 40°00'12.9" | 18 | Turakina |
| Kaukatea 232 | | Unknown | 0.86 +/- 0.08 Ma (1sd) | glass-ITPFT | 15, 16 | D | 175°06'09.5" | 39°54'07.2" | 19 | Wanganui River |
| Potaka 305 | Kidnappers | Mangakino | 1.00 +/- 0.05 (1sd) | 40Ar/39Ar | 17 | D | 175°38'36.0" | 39°59'49.6" | 15 | Rangitikei Valley |
| Rewa 304 | Ahuroa? | ? Mangakino | 1.20 +/- 0.14 Ma (1sd) | glass-ITPFT | 15, 16 | D | 175°35'04.5" | 39°59'31.8" | 15 | Rangitikei Valley |
| Mangapipi 510 | ?Ig B | Mangakino | 1.51 +/- 0.16 Ma (1sd) | glass-ITPFT | 15, 16 | D | 175°20'35.2" | 39°55'14.7" | 18 | Turakina |
| Pakihikura 303 | ?Ngaroma | Mangakino | 1.58 +/- 0.16 Ma (1sd) | glass-ITPFT | 15 | D | 175°21'16.8" | 39°54'57.6" | 18 | Turakina |
| Birdgrove 511 | | Unknown | 1.6 Ma | Astronomical | 15 | D | 175°21'20.9" | 39°54'54.3" | 18 | Turakina |
| Mangahou 302 | | Unknown | 1.63 Ma | Astronomical | 15 | D | 175°38'42.1" | 39°57'00.8" | 15 | Rangitikei Valley |
| Ototoka 521 | | Unknown | 1.72 +/- 0.32 Ma (1sd) | glass-ITPFT | 15 | D | 175°50'22.9" | 39°52'10.6" | 19 | Ototoka Beach |
| Hikuroa Pumice member | | Unknown | 2.0 +/- 0.6 Ma | zircon-ITPFT | 18 | D | 176°49'36.8" | 39°14'37.2" | 20 | Darkys Spur |

Age references
1. Hogg et al., 2003
2. Hogg et al., 2012
3. Hogg et al., 2019
4. Lowe et al., 2013
5. Lowe et al., 2019
6. Vandergoes et al., 2013
7. Nairn 2002
8. Howorth 1975
9. Danišík et al., 2020
10. Danisik et al., 2012
11. Bussell and Pillans 1997
12. Bussell 1986
13. Pillans 1994
14. Pillans 1996
15. Pillans et al., 2005
16. Rees et al., 2019
17. Houghton et al., 1995
18. Hopkins and Seward 2019

Dating Methods
Bayesian model 1 = OxCal P_sequence
Bayesian model 2 = OxCal P_sequence isochron informed
Bayesian model 3 = OxCal Tau-boundary



**Table 2. Set up for electron microprobe analysis**

| Channel 1 | | | Channel 2 | | | Channel 3 | | | Channel 4 | | | Channel 5 | | |
|---|---|---|---|---|---|---|---|---|---|---|---|---|---|---|
| Element | Mineral | Time | Element | Mineral | Time | Element | Mineral | Time | Element | Mineral | Time | Element | Mineral | Time |
| **Ca** | VG-A99 | 30/15 | **Na\*** | VG-568 | 10/5 | **Si** | VG-568 | 30/15 | **K** | VG-568 | 30/15 | **Cl** | VG-568 | 30/15 |
| **Ti** | TiO2 | 30/15 | **Mg** | VG-A99 | 30/15 | **Al** | VG-568 | 30/15 | **Fe** | VG-A99 | 30/15 | **Mn** | MnO | 30/15 |

\* Na is run twice, on the second run the peak search is skipped to reduced the volatilisation of the element





**Table 3. Average and standard deviation for all tephra, see SM Table 1 for full dataset.**

| Tephra name | | $SiO_2$ (wt%) | $TiO_2$ (wt%) | $Al_2O_3$ (wt%) | $FeO_t$ (wt%) | MnO (wt%) | MgO (wt%) | CaO (wt%) | $Na_2O$ (wt%) | $K_2O$ (wt%) | Cl (wt%) | $H_2O_b$* | $SiO_2/K_2O$ | $Na_2O + K_2O$ |
|---|---|---|---|---|---|---|---|---|---|---|---|---|---|---|
| Kaharoa | Average | 77.8 | 0.09 | 12.6 | 0.92 | 0.07 | 0.10 | 0.63 | 3.75 | 3.98 | 0.17 | 3.66 | 19.5 | 7.73 |
| n=25 (18) | 2sd | 0.41 | 0.05 | 0.20 | 0.35 | 0.03 | 0.18 | 0.18 | 0.18 | 0.39 | 0.03 | 1.45 | 1.06 | 0.56 |
| Taupo Y5 | Average | 75.6 | 0.25 | 13.4 | 1.83 | 0.10 | 0.19 | 1.41 | 4.25 | 2.99 | 0.16 | 1.39 | 25.3 | 7.11 |
| n=25 (20) | 2sd | 0.67 | 0.07 | 0.45 | 0.53 | 0.04 | 0.08 | 0.13 | 0.30 | 0.12 | 0.05 | 1.28 | 1.65 | 0.34 |
| Waimihia K3 | Average | 76.2 | 0.21 | 13.2 | 1.83 | 0.10 | 0.15 | 1.33 | 4.11 | 2.92 | 0.18 | 2.61 | 26.1 | 7.02 |
| n=22 (20) | 2sd | 0.56 | 0.03 | 0.24 | 0.19 | 0.03 | 0.02 | 0.13 | 0.26 | 0.14 | 0.02 | 1.53 | 3.86 | 0.40 |
| Unit K | Average | 76.3 | 0.20 | 13.1 | 1.77 | 0.08 | 0.14 | 1.31 | 4.11 | 2.99 | 0.15 | 1.56 | 25.5 | 7.10 |
| n=24 (20) | 2sd | 0.29 | 0.02 | 0.13 | 0.13 | 0.02 | 0.03 | 0.11 | 0.10 | 0.10 | 0.03 | 1.71 | 2.87 | 0.20 |
| Whakatane K5 | Average | 78.3 | 0.11 | 12.5 | 0.84 | 0.04 | 0.10 | 0.69 | 3.58 | 3.85 | 0.16 | 1.97 | 20.4 | 7.43 |
| n=21 | 2sd | 0.18 | 0.02 | 0.08 | 0.05 | 0.02 | 0.01 | 0.04 | 0.13 | 0.09 | 0.02 | 1.49 | 2.0 | 0.23 |
| Whakatane - P | Average | 77.5 | 0.17 | 12.7 | 1.43 | 0.06 | 0.12 | 1.02 | 3.68 | 3.29 | 0.16 | 2.10 | 23.5 | 6.97 |
| n=25 (21) | 2sd | 0.78 | 0.04 | 0.23 | 0.35 | 0.02 | 0.03 | 0.21 | 0.51 | 0.36 | 0.02 | 2.18 | 2.20 | 0.86 |
| Tuhua K6 | Average | 74.4 | 0.30 | 9.7 | 5.80 | 0.12 | 0.01 | 0.27 | 5.18 | 4.19 | 0.23 | 0.95 | 17.8 | 9.37 |
| n=23 (23) | 2sd | 0.31 | 0.02 | 0.14 | 0.15 | 0.02 | 0.01 | 0.02 | 0.18 | 0.16 | 0.01 | 0.52 | 1.98 | 0.33 |
| Mamaku K7 | Average | 78.3 | 0.12 | 12.6 | 0.88 | 0.04 | 0.11 | 0.79 | 3.61 | 3.59 | 0.16 | 2.14 | 21.8 | 7.20 |
| n=23 (23) | 2sd | 0.18 | 0.02 | 0.15 | 0.05 | 0.02 | 0.02 | 0.06 | 0.11 | 0.13 | 0.01 | 1.54 | 1.33 | 0.25 |
| Rotoma - P | Average | 78.2 | 0.11 | 12.5 | 0.89 | 0.04 | 0.11 | 0.73 | 3.66 | 3.74 | 0.15 | 0.68 | 20.9 | 7.40 |
| n=24 | 2sd | 0.17 | 0.01 | 0.09 | 0.08 | 0.02 | 0.01 | 0.03 | 0.10 | 0.14 | 0.01 | 0.66 | 1.2 | 0.24 |
| Rotoma - K8 - D | Average | 78.2 | 0.12 | 12.6 | 0.87 | 0.05 | 0.13 | 0.84 | 3.73 | 3.45 | 0.15 | 0.83 | 22.7 | 7.19 |
| n=23 (20) | 2sd | 0.18 | 0.01 | 0.08 | 0.05 | 0.02 | 0.02 | 0.07 | 0.13 | 0.24 | 0.01 | 0.79 | 0.75 | 0.37 |
| Opepe K9 | Average | 76.0 | 0.23 | 13.4 | 1.80 | 0.07 | 0.17 | 1.74 | 3.77 | 2.76 | 0.16 | 4.02 | 27.6 | 6.53 |
| n=14 (8) | 2sd | 0.43 | 0.05 | 0.16 | 0.19 | 0.03 | 0.04 | 0.20 | 0.16 | 0.30 | 0.03 | 1.50 | 1.42 | 0.46 |
| Poronui K10 | Average | 76.4 | 0.21 | 13.2 | 1.64 | 0.07 | 0.16 | 1.53 | 3.75 | 3.02 | 0.14 | 1.46 | 25.3 | 6.77 |
| n=18 (15) | 2sd | 0.49 | 0.08 | 0.21 | 0.22 | 0.03 | 0.04 | 0.14 | 0.11 | 0.16 | 0.02 | 1.47 | 3.15 | 0.27 |
| Karapiti K11 | Average | 76.6 | 0.19 | 13.1 | 1.58 | 0.06 | 0.14 | 1.48 | 3.67 | 3.16 | 0.14 | 1.84 | 24.3 | 6.82 |
| n=25 (11) | 2sd | 0.26 | 0.02 | 0.11 | 0.16 | 0.02 | 0.02 | 0.14 | 0.18 | 0.36 | 0.02 | 1.27 | 0.72 | 0.54 |
| Waiohau - P | Average | 78.0 | 0.12 | 12.5 | 0.96 | 0.06 | 0.14 | 0.84 | 3.89 | 3.31 | 0.14 | 3.01 | 23.6 | 7.20 |
| n=19 (19) | 2sd | 0.27 | 0.02 | 0.26 | 0.13 | 0.01 | 0.03 | 0.07 | 0.19 | 0.14 | 0.03 | 2.33 | 1.06 | 0.23 |
| Waiohau - K14b D | Average | 78.1 | 0.12 | 12.4 | 1.03 | 0.05 | 0.16 | 0.88 | 3.87 | 3.23 | 0.14 | 1.02 | 24.2 | 7.10 |
| n=10 (10) | 2sd | 0.26 | 0.02 | 0.17 | 0.05 | 0.02 | 0.01 | 0.03 | 0.26 | 0.09 | 0.02 | 1.02 | 0.70 | 0.27 |
| Rotorua - P | Average | 77.6 | 0.14 | 12.5 | 1.15 | 0.05 | 0.14 | 1.00 | 3.75 | 3.51 | 0.15 | 4.58 | 22.5 | 7.26 |
| n=29 (28) | 2sd | 0.57 | 0.06 | 0.28 | 0.21 | 0.02 | 0.07 | 0.33 | 0.31 | 0.47 | 0.04 | 2.18 | 2.79 | 0.45 |
| Rotorua - K15 D | Average | 77.5 | 0.08 | 12.5 | 0.91 | 0.06 | 0.09 | 0.73 | 3.80 | 4.13 | 0.14 | 0.64 | 19.0 | 7.93 |
| n=10 (10) | 2sd | 0.35 | 0.05 | 0.19 | 0.15 | 0.01 | 0.07 | 0.21 | 0.23 | 0.43 | 0.02 | 0.63 | 2.47 | 0.34 |
| Rerewhakaaitu - P | Average | 78.0 | 0.09 | 12.3 | 0.98 | 0.05 | 0.10 | 0.77 | 3.61 | 3.84 | 0.15 | 3.06 | 20.5 | 7.45 |
| n=21 (19) | 2sd | 0.29 | 0.02 | 0.20 | 0.10 | 0.02 | 0.03 | 0.10 | 0.26 | 0.39 | 0.03 | 1.97 | 2.31 | 0.28 |
| Rerewhakaaitu - K17 D | Average | 77.7 | 0.08 | 12.6 | 0.94 | 0.05 | 0.10 | 0.79 | 3.71 | 3.94 | 0.15 | 3.31 | 19.9 | 7.65 |
| n=11 (10) | 2sd | 0.30 | 0.02 | 0.29 | 0.07 | 0.02 | 0.02 | 0.06 | 0.29 | 0.36 | 0.02 | 1.98 | 1.99 | 0.43 |
| Okareka | Average | 78.5 | 0.14 | 12.6 | 0.87 | 0.06 | 0.12 | 0.91 | 3.57 | 3.24 | 0.20 | 4.93 | 24.2 | 6.81 |
| n=22 (22) | 2sd | 0.20 | 0.02 | 0.11 | 0.06 | 0.02 | 0.02 | 0.02 | 0.11 | 0.18 | 0.02 | 0.73 | 1.09 | 0.29 |
| Te Rere | Average | 78.1 | 0.09 | 12.4 | 1.05 | 0.06 | 0.06 | 0.64 | 3.62 | 3.97 | 0.22 | 3.80 | 19.7 | 7.59 |
| n=24 (20) | 2sd | 0.18 | 0.02 | 0.10 | 0.08 | 0.03 | 0.02 | 0.09 | 0.16 | 0.24 | 0.02 | 1.18 | 0.73 | 0.41 |
| Kawakawa/Oruanui | Average | 77.9 | 0.17 | 12.8 | 1.27 | 0.06 | 0.14 | 1.24 | 3.34 | 3.14 | 0.17 | 5.18 | 24.8 | 6.48 |
| n=25 (23) | 2sd | 0.82 | 0.05 | 0.32 | 0.15 | 0.03 | 0.04 | 0.21 | 0.60 | 0.34 | 0.02 | 1.01 | 2.42 | 0.94 |
| Poihipi | Average | 78.1 | 0.12 | 12.6 | 0.96 | 0.07 | 0.11 | 0.90 | 3.52 | 3.65 | 0.50 | 5.71 | 21.4 | 7.17 |
| n=20 (18) | 2sd | 0.30 | 0.02 | 0.17 | 0.17 | 0.03 | 0.08 | 0.11 | 0.39 | 0.41 | 1.55 | 16.69 | 0.73 | 0.80 |
| Okaia | Average | 77.8 | 0.18 | 12.8 | 1.31 | 0.07 | 0.15 | 1.28 | 3.43 | 3.04 | 0.18 | 5.34 | 25.6 | 6.47 |
| n=23 (23) | 2sd | 0.64 | 0.06 | 0.26 | 0.17 | 0.03 | 0.05 | 0.18 | 0.24 | 0.26 | 0.06 | 1.89 | 2.45 | 0.50 |
| Unit L | Average | 77.8 | 0.15 | 12.7 | 1.18 | 0.06 | 0.14 | 1.16 | 3.62 | 3.16 | 0.22 | 4.48 | 24.6 | 6.78 |
| n=23 (20) | 2sd | 0.34 | 0.03 | 0.28 | 0.08 | 0.02 | 0.03 | 0.13 | 0.32 | 0.41 | 0.22 | 6.31 | 0.83 | 0.72 |
| Awakeri | Average | 77.4 | 0.18 | 13.0 | 1.16 | 0.07 | 0.18 | 1.07 | 3.99 | 2.97 | 0.15 | 3.57 | 26.1 | 6.95 |
| n=21 (19) | 2sd | 0.22 | 0.01 | 0.16 | 0.11 | 0.02 | 0.02 | 0.07 | 0.13 | 0.07 | 0.02 | 1.29 | 3.08 | 0.20 |
| Mangaone | Average | 76.7 | 0.20 | 13.3 | 1.25 | 0.08 | 0.22 | 1.20 | 4.14 | 2.86 | 0.18 | 2.56 | 26.8 | 7.00 |
| n=24 (19) | 2sd | 0.38 | 0.02 | 0.18 | 0.08 | 0.02 | 0.02 | 0.08 | 0.16 | 0.08 | 0.01 | 1.32 | 4.83 | 0.24 |
| Hauparu | Average | 75.2 | 0.38 | 13.7 | 1.86 | 0.08 | 0.39 | 1.78 | 3.87 | 2.73 | 0.15 | 3.05 | 27.6 | 6.60 |
| n=21 (12) | 2sd | 0.90 | 0.05 | 0.39 | 0.26 | 0.02 | 0.06 | 0.21 | 0.16 | 0.15 | 0.02 | 1.57 | 6.14 | 0.31 |
| Maketu | Average | 73.1 | 0.49 | 14.4 | 2.32 | 0.10 | 0.56 | 2.12 | 4.49 | 2.41 | 0.16 | 4.29 | 30.4 | 6.90 |
| n=3 (1) | 2sd | 0.26 | 0.05 | 0.20 | 0.26 | 0.01 | 0.06 | 0.14 | 0.35 | 0.13 | 0.03 | 6.25 | | |
| Ngamotu | Average | 75.7 | 0.34 | 13.7 | 1.54 | 0.09 | 0.36 | 1.73 | 3.92 | 2.62 | 0.15 | 3.93 | 28.8 | 6.54 |
| n=21 (19) | 2sd | 1.08 | 0.09 | 0.51 | 0.34 | 0.02 | 0.15 | 0.37 | 0.24 | 0.22 | 0.02 | 2.90 | 5.03 | 0.46 |
| Tahuna | Average | 78.2 | 0.13 | 12.5 | 0.95 | 0.04 | 0.12 | 0.86 | 3.49 | 3.75 | 0.18 | 3.84 | 20.8 | 7.24 |
| n=23 (17) | 2sd | 0.31 | 0.02 | 0.19 | 0.07 | 0.02 | 0.03 | 0.10 | 0.28 | 0.41 | 0.02 | 0.69 | 0.76 | 0.69 |
| Earthquake Flat Ig | Average | 77.9 | 0.11 | 12.5 | 0.95 | 0.03 | 0.10 | 0.79 | 3.23 | 4.43 | 0.21 | 3.83 | 17.6 | 7.65 |
| n=24 (11) | 2sd | 0.20 | 0.02 | 0.08 | 0.09 | 0.01 | 0.01 | 0.03 | 0.11 | 0.11 | 0.03 | 0.71 | 1.76 | 0.22 |
| Rotoehu tephra | Average | 78.3 | 0.14 | 12.6 | 0.91 | 0.05 | 0.13 | 0.83 | 3.66 | 3.42 | 0.16 | 3.57 | 22.9 | 7.08 |
| n=21 (14) | 2sd | 0.18 | 0.02 | 0.06 | 0.05 | 0.02 | 0.01 | 0.03 | 0.14 | 0.08 | 0.01 | 0.80 | 2.28 | 0.22 |
| Rotoiti Ig | Average | 78.3 | 0.14 | 12.6 | 0.87 | 0.05 | 0.14 | 0.86 | 3.72 | 3.35 | 0.21 | 6.00 | 23.4 | 7.08 |
| n=23 (19) | 2sd | 0.16 | 0.02 | 0.13 | 0.12 | 0.02 | 0.02 | 0.05 | 0.22 | 0.11 | 0.08 | 4.52 | 1.44 | 0.34 |
| Ararata Gully 318 | Average | 77.2 | 0.15 | 12.8 | 1.29 | 0.03 | 0.13 | 1.03 | 3.64 | 3.65 | 0.17 | 5.61 | 21.2 | 7.29 |
| n=24 (15) | 2sd | 0.70 | 0.04 | 0.30 | 0.19 | 0.01 | 0.05 | 0.17 | 0.09 | 0.09 | 0.02 | 0.79 | 7.60 | 0.18 |
| Kakariki 272 | Average | 78.1 | 0.13 | 12.5 | 0.98 | 0.02 | 0.11 | 0.91 | 3.50 | 3.74 | 0.16 | 4.99 | 20.9 | 7.24 |
| n=25 (18) | 2sd | 0.13 | 0.02 | 0.07 | 0.10 | 0.02 | 0.02 | 0.04 | 0.09 | 0.14 | 0.01 | 0.38 | 0.87 | 0.24 |
| Fordell 449 | Average | 78.0 | 0.11 | 12.4 | 1.07 | 0.02 | 0.07 | 0.64 | 3.74 | 3.97 | 0.20 | 5.16 | 19.7 | 7.70 |
| n=25 (19) | 2sd | 0.44 | 0.04 | 0.19 | 0.25 | 0.01 | 0.06 | 0.16 | 0.13 | 0.18 | 0.02 | 0.59 | 2.42 | 0.31 |
| Upper Griffin Road 307 | Average | 77.5 | 0.17 | 12.5 | 1.44 | 0.03 | 0.11 | 1.01 | 3.69 | 3.47 | 0.19 | 5.46 | 22.4 | 7.16 |
| n=21 (17) | 2sd | 0.19 | 0.02 | 0.09 | 0.10 | 0.02 | 0.02 | 0.04 | 0.11 | 0.10 | 0.02 | 0.47 | 1.93 | 0.21 |
| Lower Griffin Road 309 | Average | 76.5 | 0.28 | 13.2 | 1.30 | 0.03 | 0.28 | 1.39 | 3.72 | 3.32 | 0.17 | 5.14 | 23.1 | 7.04 |
| n=25 (21) | 2sd | 0.62 | 0.05 | 0.31 | 0.12 | 0.01 | 0.07 | 0.20 | 0.15 | 0.20 | 0.02 | 0.45 | 3.10 | 0.35 |
| Onepuhi 267 | Average | 77.1 | 0.14 | 12.8 | 1.32 | 0.03 | 0.09 | 0.96 | 3.49 | 4.04 | 0.24 | 5.20 | 19.1 | 7.53 |
| n=23 (10) | 2sd | 1.08 | 0.05 | 0.45 | 0.41 | 0.02 | 0.04 | 0.23 | 0.29 | 0.47 | 0.03 | 0.96 | 2.28 | 0.77 |
| Kupe 481 | Average | 77.5 | 0.16 | 12.5 | 1.34 | 0.03 | 0.09 | 0.88 | 3.54 | 3.96 | 0.25 | 5.22 | 19.6 | 7.50 |
| n=24 (17) | 2sd | 0.63 | 0.04 | 0.28 | 0.16 | 0.01 | 0.02 | 0.15 | 0.19 | 0.31 | 0.05 | 1.17 | 2.02 | 0.50 |
| Kaukatea 232 | Average | 76.4 | 0.17 | 13.3 | 1.61 | 0.04 | 0.09 | 1.02 | 4.01 | 3.41 | 0.19 | 6.02 | 22.4 | 7.43 |
| n=25 (20) | 2sd | 0.22 | 0.02 | 0.10 | 0.07 | 0.02 | 0.01 | 0.04 | 0.14 | 0.18 | 0.01 | 0.52 | 1.20 | 0.32 |
| Potaka 305 | Average | 77.6 | 0.14 | 12.6 | 1.24 | 0.03 | 0.10 | 1.02 | 3.58 | 3.65 | 0.24 | 5.65 | 21.3 | 7.23 |
| n=24 (21) | 2sd | 0.24 | 0.01 | 0.11 | 0.11 | 0.02 | 0.01 | 0.08 | 0.16 | 0.21 | 0.01 | 0.60 | 1.14 | 0.37 |
| Rewa 304 | Average | 75.6 | 0.19 | 13.4 | 1.95 | 0.04 | 0.12 | 1.25 | 4.01 | 3.36 | 0.20 | 6.13 | 22.6 | 7.37 |
| n=25 (18) | 2sd | 0.48 | 0.03 | 0.19 | 0.20 | 0.02 | 0.03 | 0.13 | 0.21 | 0.33 | 0.11 | 1.87 | 1.47 | 0.54 |
| Mangapipi 510 | Average | 74.8 | 0.20 | 13.8 | 2.28 | 0.05 | 0.10 | 1.16 | 4.17 | 3.50 | 0.22 | 5.84 | 21.3 | 7.67 |
| n=24 (11) | 2sd | 0.26 | 0.03 | 0.17 | 0.13 | 0.02 | 0.02 | 0.08 | 0.20 | 0.27 | 0.02 | 1.13 | 0.96 | 0.47 |
| Pakihikura 303 | Average | 77.7 | 0.10 | 12.6 | 1.33 | 0.03 | 0.07 | 1.21 | 3.53 | 3.34 | 0.17 | 5.71 | 23.3 | 6.88 |
| n=25 (21) | 2sd | 0.18 | 0.02 | 0.09 | 0.07 | 0.02 | 0.01 | 0.03 | 0.12 | 0.18 | 0.01 | 0.83 | 1.01 | 0.30 |
| Birdgrove 511 | Average | 75.7 | 0.16 | 13.4 | 1.72 | 0.04 | 0.09 | 1.08 | 3.76 | 4.09 | 0.19 | 4.93 | 18.5 | 7.85 |
| n=24 (19) | 2sd | 1.02 | 0.07 | 0.34 | 0.48 | 0.01 | 0.04 | 0.20 | 0.39 | 0.51 | 0.04 | 0.92 | 2.01 | 0.89 |
| Mangahou 302 | Average | 75.2 | 0.21 | 13.7 | 1.81 | 0.03 | 0.18 | 1.55 | 3.44 | 3.86 | 0.20 | 6.20 | 19.5 | 7.29 |
| n=24 (19) | 2sd | 0.49 | 0.05 | 0.14 | 0.18 | 0.02 | 0.04 | 0.22 | 0.32 | 0.51 | 0.02 | 0.86 | 0.95 | 0.84 |
| Ototaka 521 | Average | 74.4 | 0.23 | 13.8 | 2.38 | 0.06 | 0.11 | 1.25 | 4.08 | 3.64 | 0.26 | 6.11 | 20.4 | 7.72 |
| n=21 (19) | 2sd | 0.42 | 0.03 | 0.17 | 0.31 | 0.03 | 0.03 | 0.14 | 0.28 | 0.38 | 0.01 | 0.84 | 1.12 | 0.65 |
| Hikuroa Pumice member | Average | 77.6 | 0.08 | 12.3 | 1.45 | 0.03 | 0.06 | 0.85 | 3.71 | 3.74 | | 3.55 | 20.8 | 7.44 |
| n=18 (16) | 2sd | 0.43 | 0.02 | 0.29 | 0.11 | 0.02 | 0.02 | 0.04 | 0.25 | 0.17 | | 1.33 | 2.60 | 0.41 |

P = proximal
D = distal
K = Kaipo Bog sample
XXX = Pillans et al., 2005 sample number
n = number of shards analysed, (trace in parenthesis)
$H_2O_b$* = water and volatiles calculated by difference





**Table 3 (cont.) Average and standard deviation for all tephra, see SM Table 1 for full dataset.**

| Tephra name | | Sc (ppm) | Ti (ppm) | V (ppm) | Mn (ppm) | Co (ppm) | Cu (ppm) | Zn (ppm) | Ga (ppm) | Rb (ppm) | Sr86 (ppm) | Sr88 (ppm) | Y (ppm) | Zr90 (ppm) |
|---|---|---|---|---|---|---|---|---|---|---|---|---|---|---|
| Kaharoa | Average | 11.5 | 528 | 1.80 | 453 | 0.28 | 1.49 | 40.3 | 14.4 | 133 | 50.3 | 45.6 | 33.0 | 961 |
| n=25 (18) | 2sd | 6.36 | 165 | 1.00 | 84.9 | 0.17 | 1.22 | 11.1 | 3.11 | 37.7 | 15.3 | 20.2 | 29.8 | 3753 |
| Taupo Y5 | Average | 12.6 | 1175 | 1.27 | 531 | 0.67 | 2.85 | 49.3 | 13.7 | 110 | 112 | 111 | 24.2 | 162 |
| n=25 (20) | 2sd | 3.65 | 448 | 0.87 | 210 | 0.72 | 4.08 | 15.3 | 3.41 | 30.0 | 31.3 | 30.2 | 7.16 | 59.8 |
| Waimihia K3 | Average | 14.3 | 1189 | 3.03 | 731 | 0.43 | 7.11 | 71.8 | 18.8 | 102 | 112 | 113 | 30.7 | 211 |
| n=22 (20) | 2sd | 2.14 | 239 | 9.89 | 151 | 0.12 | 17.09 | 23.9 | 7.10 | 18.4 | 23.6 | 18.6 | 6.71 | 30.9 |
| Unit K | Average | 7.19 | 1188 | 0.78 | 595 | 0.67 | 5.37 | 58.4 | 17.1 | 110 | 119 | 118 | 32.8 | 218 |
| n=24 (20) | 2sd | 0.92 | 144 | 0.16 | 64.9 | 0.12 | 9.76 | 14.9 | 1.99 | 14.9 | 12.8 | 11.8 | 3.81 | 22.5 |
| Whakatane K5 | Average | | | | | | | | | | | | | |
| n=21 | 2sd | | | | | | | | | | | | | |
| Whakatane - P | Average | 6.57 | 985 | 1.01 | 528 | 0.55 | 2.67 | 41.6 | 17.3 | 123 | 80.0 | 81.9 | 31.5 | 173 |
| n=25 (21) | 2sd | 1.90 | 262 | 0.72 | 108 | 0.20 | 1.63 | 19.3 | 2.99 | 25.4 | 26.1 | 24.3 | 8.39 | 67.9 |
| Tuhua K6 | Average | 3.40 | 1677 | 0.71 | 1091 | 0.57 | 5.43 | 215 | 44.1 | 150 | 21.1 | 11.7 | 131 | 1171 |
| n=23 (22) | 2sd | 0.64 | 449 | 0.44 | 318 | 0.83 | 4.28 | 99.9 | 13.66 | 25.0 | 27.0 | 22.4 | 48.3 | 481 |
| Mamaku K7 | Average | 4.39 | 755 | 1.13 | 436 | 0.49 | 2.76 | 28.1 | 15.1 | 127 | 68.8 | 65.0 | 25.9 | 99.4 |
| n=23 (23) | 2sd | 0.80 | 133 | 0.26 | 64.9 | 0.10 | 1.95 | 7.3 | 2.23 | 20.0 | 11.9 | 10.2 | 3.49 | 17.0 |
| Rotoma - P | Average | | | | | | | | | | | | | |
| n=24 | 2sd | | | | | | | | | | | | | |
| Rotoma - K8 - D | Average | 4.08 | 776 | 1.27 | 464 | 0.45 | 1.25 | 37.9 | 14.0 | 115 | 71.2 | 71.6 | 27.1 | 96.9 |
| n=23 (20) | 2sd | 0.44 | 79 | 0.27 | 37.4 | 0.11 | 0.42 | 7.2 | 1.19 | 13.3 | 8.21 | 6.32 | 2.40 | 9.71 |
| Opepe K9 | Average | 8.02 | 1364 | 4.01 | 529 | 1.70 | 6.13 | 43.7 | 18.6 | 123 | 126 | 130 | 33.0 | 234 |
| n=14 (8) | 2sd | 0.31 | 131 | 2.01 | 66.4 | 0.21 | 3.92 | 13.8 | 1.76 | 11.7 | 10.9 | 12.9 | 3.00 | 20.7 |
| Poronui K10 | Average | 8.01 | 1297 | 2.87 | 463 | 1.62 | 4.23 | 46.9 | 17.1 | 126 | 118 | 118 | 32.9 | 230 |
| n=18 (15) | 2sd | 1.51 | 253 | 1.00 | 92.3 | 0.34 | 1.96 | 15.4 | 2.71 | 23.5 | 20.4 | 19.3 | 6.59 | 33.8 |
| Karapiti K11 | Average | 7.05 | 1162 | 2.60 | 395 | 1.57 | 2.93 | 44.1 | 16.3 | 109 | 109 | 112 | 30.6 | 209 |
| n=25 (11) | 2sd | 1.32 | 242 | 0.84 | 79.3 | 0.41 | 2.02 | 11.0 | 2.80 | 18.9 | 14.2 | 14.6 | 5.49 | 38.7 |
| Waiohau - P | Average | 13.4 | 652 | 1.35 | 443 | 0.12 | 0.73 | 39.1 | 14.2 | 110 | 68.7 | 69.3 | 21.5 | 83.6 |
| n=19 (19) | 2sd | 2.47 | 56 | 1.22 | 14.1 | 0.45 | 0.71 | 6.3 | 2.04 | 5.2 | 5.43 | 4.87 | 1.62 | 7.95 |
| Waiohau - K14b D | Average | 17.0 | 760 | 1.88 | 406 | 0.40 | 1.31 | 39.3 | 12.5 | 112 | 77.9 | 77.9 | 21.5 | 93.4 |
| n=10 (10) | 2sd | 1.94 | 28 | 0.81 | 16.7 | 0.37 | 0.80 | 8.1 | 0.88 | 2.9 | 4.78 | 2.40 | 0.90 | 1.53 |
| Rotorua - P | Average | 13.8 | 903 | 2.89 | 424 | 0.53 | 1.19 | 53.6 | 14.3 | 118 | 74.4 | 76.1 | 23.6 | 127 |
| n=29 (28) | 2sd | 2.83 | 356 | 2.20 | 65.6 | 0.45 | 1.00 | 26.8 | 2.04 | 18.0 | 34.0 | 33.3 | 5.63 | 37.7 |
| Rotorua -K15 D | Average | 17.9 | 470 | 1.79 | 483 | 0.44 | 2.63 | 30.7 | 13.7 | 146 | 76.5 | 73.7 | 20.9 | 74.3 |
| n=10 (10) | 2sd | 2.32 | 221 | 1.23 | 45.4 | 0.36 | 1.89 | 9.5 | 2.65 | 20.3 | 92.5 | 86.4 | 3.38 | 32.1 |
| Rerewhakaaitu - P | Average | 13.9 | 720 | 2.51 | 386 | 0.29 | 2.49 | 35.3 | 13.3 | 134 | 63.3 | 64.1 | 22.1 | 82.9 |
| n=21 (19) | 2sd | 2.64 | 446 | 3.30 | 67.7 | 0.67 | 3.41 | 9.6 | 2.83 | 18.1 | 43.8 | 43.7 | 6.97 | 20.5 |
| Rerewhakaaitu -K1 | Average | 18.5 | 499 | 2.54 | 481 | 0.56 | 2.05 | 34.0 | 13.9 | 142 | 60.3 | 58.1 | 18.9 | 75.3 |
| n=11 (10) | 2sd | 4.02 | 119 | 3.69 | 51.4 | 0.51 | 1.24 | 9.3 | 1.30 | 16.9 | 13.1 | 12.1 | 2.32 | 12.3 |
| Okareka | Average | 5.21 | 885 | 2.72 | 454 | 0.80 | 5.11 | 24.9 | 14.9 | 113 | 70.8 | 68.7 | 22.4 | 100 |
| n=22 (22) | 2sd | 0.97 | 100 | 0.49 | 44.3 | 0.18 | 5.17 | 7.6 | 1.65 | 11.7 | 8.45 | 7.36 | 2.60 | 11.4 |
| Te Rere | Average | 5.86 | 698 | 1.56 | 367 | 0.75 | 3.14 | 41.5 | 17.2 | 139 | 34.7 | 34.2 | 37.9 | 123 |
| n=24 (20) | 2sd | 1.68 | 150 | 2.05 | 62.0 | 0.69 | 1.83 | 10.8 | 2.71 | 28.2 | 9.56 | 8.81 | 6.85 | 18.6 |
| Kawakawa/Oruanui | Average | 4.92 | 1003 | 3.33 | 404 | 1.27 | 5.73 | 32.0 | 16.6 | 124 | 95.5 | 94.9 | 23.1 | 145 |
| n=25 (23) | 2sd | 1.36 | 222 | 3.09 | 76.6 | 0.51 | 7.60 | 9.9 | 5.71 | 15.8 | 18.1 | 17.3 | 3.16 | 31.7 |
| Poihipi | Average | 4.36 | 915 | 3.38 | 438 | 0.73 | 3.20 | 32.5 | 17.7 | 146 | 75.2 | 72.2 | 25.8 | 111 |
| n=20 (18) | 2sd | 1.17 | 369 | 6.28 | 82.0 | 0.53 | 2.58 | 9.5 | 9.58 | 50.7 | 13.8 | 14.0 | 5.47 | 28.3 |
| Okaia | Average | 5.21 | 1114 | 3.82 | 417 | 1.40 | 4.53 | 30.3 | 15.5 | 132 | 106 | 102 | 24.3 | 152 |
| n=23 (23) | 2sd | 0.88 | 333 | 2.44 | 72.5 | 0.70 | 2.78 | 10.2 | 2.32 | 23.7 | 19.9 | 19.1 | 3.92 | 33.1 |
| Unit L | Average | 4.96 | 1061 | 2.19 | 469 | 0.90 | 3.94 | 29.6 | 15.3 | 116 | 104 | 100 | 27.9 | 148 |
| n=23 (20) | 2sd | 0.81 | 167 | 0.68 | 121 | 0.36 | 2.22 | 9.4 | 1.87 | 19.7 | 18.0 | 17.8 | 7.78 | 15.0 |
| Awakeri | Average | 6.23 | 1216 | 1.48 | 630 | 0.40 | 2.18 | 45.8 | 17.7 | 103 | 107 | 107 | 35.3 | 163 |
| n=21 (19) | 2sd | 1.52 | 297 | 0.48 | 156 | 0.12 | 1.18 | 16.5 | 3.85 | 24.3 | 15.7 | 15.6 | 7.89 | 37.1 |
| Mangaone | Average | 4.96 | 1233 | 1.85 | 669 | 0.43 | 2.08 | 33.3 | 16.1 | 88.0 | 114 | 118 | 32.9 | 173 |
| n=24 (19) | 2sd | 0.48 | 94 | 0.23 | 58.7 | 0.07 | 1.34 | 10.4 | 1.61 | 7.1 | 9.37 | 10.6 | 2.78 | 16.2 |
| Haparu | Average | 6.57 | 2254 | 12.1 | 637 | 1.73 | 3.40 | 40.4 | 15.8 | 82.0 | 155 | 153 | 30.3 | 239 |
| n=21 (12) | 2sd | 1.11 | 240 | 1.87 | 64.2 | 0.38 | 1.82 | 18.7 | 2.06 | 11.5 | 29.4 | 28.4 | 3.37 | 28.5 |
| Maketu | Average | 8.80 | 2890 | 12.6 | 940 | 1.84 | 2.96 | 29.5 | 19.6 | 79.0 | 188 | 196 | 42.4 | 298 |
| n=3 (1) | 2sd | | | | | | | | | | | | | |
| Ngamotu | Average | 4.70 | 2065 | 16.8 | 646 | 3.21 | 6.27 | 42.5 | 16.0 | 72.0 | 193 | 189 | 23.5 | 179 |
| n=21 (19) | 2sd | 0.63 | 796 | 26.7 | 87.9 | 6.03 | 11.8 | 14.9 | 2.97 | 17.1 | 107 | 107 | 5.39 | 29.6 |
| Tahuna | Average | 4.05 | 895 | 2.61 | 442 | 1.03 | 3.62 | 25.9 | 18.2 | 136 | 61.4 | 61.5 | 21.0 | 102 |
| n=23 (17) | 2sd | 0.61 | 146 | 0.91 | 86.7 | 0.35 | 1.65 | 7.4 | 7.01 | 38.1 | 9.56 | 10.4 | 3.13 | 18.3 |
| Earthquake Flat Ig | Average | 4.14 | 884 | 2.40 | 490 | 0.84 | 2.14 | 19.6 | 15.8 | 122 | 67.4 | 64.9 | 20.5 | 93.1 |
| n=24 (11) | 2sd | 0.43 | 66 | 0.50 | 38.2 | 0.17 | 1.01 | 3.0 | 1.45 | 30.6 | 7.85 | 5.56 | 2.10 | 7.87 |
| Rotoehu tephra | Average | 4.33 | 1109 | 3.84 | 536 | 0.94 | 2.75 | 32.4 | 14.9 | 100 | 81.6 | 82.4 | 22.3 | 113 |
| n=21 (14) | 2sd | 0.62 | 564 | 3.50 | 141 | 0.30 | 0.99 | 9.5 | 2.15 | 11.9 | 50.4 | 48.8 | 3.43 | 46.8 |
| Rotoiti Ig | Average | 4.10 | 880 | 2.92 | 453 | 0.81 | 3.39 | 27.1 | 14.4 | 103 | 64.9 | 64.8 | 20.0 | 90.6 |
| n=23 (19) | 2sd | 0.46 | 84 | 2.37 | 47.8 | 0.27 | 1.98 | 8.3 | 1.90 | 15.6 | 8.09 | 6.81 | 2.50 | 10.1 |
| Ararata Gully 318 | Average | 5.42 | 945 | 2.77 | 415 | 0.97 | 10.2 | 28.7 | 18.1 | 144 | 82.6 | 84.9 | 26.0 | 147 |
| n=24 (15) | 2sd | 0.67 | 259 | 2.69 | 52.8 | 0.66 | 13.7 | 8.8 | 1.65 | 17.0 | 21.7 | 19.1 | 2.20 | 28.7 |
| Kakariki 272 | Average | 4.43 | 848 | 4.14 | 366 | 1.29 | 18.0 | 28.2 | 17.3 | 133 | 75.8 | 77.8 | 21.9 | 107 |
| n=25 (18) | 2sd | 0.59 | 146 | 4.56 | 40.1 | 0.97 | 22.0 | 7.6 | 2.94 | 20.5 | 10.7 | 7.75 | 2.53 | 11.3 |
| Fordell 449 | Average | 6.68 | 743 | 3.78 | 428 | 0.67 | 15.0 | 47.8 | 18.4 | 162 | 41.0 | 41.1 | 39.3 | 135 |
| n=25 (19) | 2sd | 1.24 | 174 | 3.81 | 85.8 | 0.37 | 22.7 | 10.5 | 3.87 | 36.6 | 12.4 | 12.0 | 8.71 | 30.2 |
| Upper Griffin Road | Average | 5.84 | 1107 | 2.51 | 394 | 1.05 | 5.52 | 50.9 | 16.1 | 132 | 70.8 | 75.7 | 35.2 | 154 |
| n=21 (17) | 2sd | 2.12 | 234 | 2.51 | 43.4 | 0.28 | 5.89 | 8.4 | 1.84 | 15.7 | 10.1 | 10.9 | 7.56 | 8.35 |
| Lower Griffin Road | Average | 5.21 | 1792 | 17.6 | 492 | 1.71 | 23.5 | 47.5 | 18.7 | 115 | 114 | 118 | 24.3 | 195 |
| n=25 (21) | 2sd | 1.36 | 948 | 35.1 | 95.4 | 2.72 | 68.1 | 16.3 | 12.4 | 9.9 | 22.9 | 25.0 | 2.43 | 53.4 |
| Onepuhi 267 | Average | 5.27 | 1135 | 16.6 | 435 | 2.40 | 42.7 | 49.3 | 17.9 | 150 | 75.0 | 77.3 | 28.1 | 163 |
| n=23 (10) | 2sd | 0.87 | 428 | 27.0 | 83.8 | 3.18 | 51.6 | 18.7 | 2.88 | 35.9 | 25.6 | 24.8 | 9.69 | 79.9 |
| Kupe 481 | Average | 5.38 | 973 | 3.38 | 450 | 3.94 | 21.9 | 37.4 | 17.2 | 152 | 83.1 | 65.6 | 29.6 | 144 |
| n=24 (17) | 2sd | 1.37 | 169 | 4.37 | 359 | 10.68 | 30.9 | 10.8 | 4.04 | 57.4 | 30.4 | 13.4 | 10.0 | 33.2 |
| Kaukatea 232 | Average | 6.12 | 1099 | 2.16 | 505 | 0.81 | 8.93 | 59.3 | 19.0 | 134 | 71.3 | 81.7 | 36.5 | 209 |
| n=25 (20) | 2sd | 0.46 | 112 | 3.16 | 34.1 | 0.28 | 15.96 | 7.2 | 1.20 | 10.7 | 17.8 | 7.15 | 2.17 | 13.3 |
| Potaka 305 | Average | 5.44 | 932 | 2.49 | 342 | 1.37 | 5.96 | 36.4 | 18.0 | 163 | 73.0 | 79.6 | 25.2 | 129 |
| n=24 (21) | 2sd | 0.49 | 103 | 0.89 | 34.6 | 0.17 | 32.6 | 8.5 | 2.05 | 17.9 | 21.6 | 9.59 | 2.73 | 17.0 |
| Rewa 304 | Average | 5.83 | 1087 | 2.39 | 529 | 1.39 | 5.96 | 49.6 | 19.3 | 117 | 107 | 105 | 34.3 | 226 |
| n=25 (18) | 2sd | 0.62 | 218 | 4.36 | 72.4 | 0.89 | 6.72 | 12.8 | 2.83 | 15.7 | 59.3 | 51.4 | 5.29 | 38.6 |
| Mangapipi 510 | Average | 6.66 | 1395 | 2.51 | 645 | 0.99 | 4.94 | 68.8 | 19.8 | 134 | 86.9 | 83.9 | 39.0 | 301 |
| n=24 (11) | 2sd | 0.67 | 494 | 3.80 | 87.0 | 0.74 | 4.44 | 12.7 | 1.76 | 16.5 | 13.0 | 10.1 | 2.42 | 21.9 |
| Pakihikura 303 | Average | 5.03 | 609 | 0.94 | 296 | 1.03 | 10.4 | 33.3 | 15.9 | 127 | 88.5 | 86.2 | 20.3 | 102 |
| n=25 (21) | 2sd | 0.43 | 63 | 0.24 | 22.1 | 0.15 | 12.3 | 6.9 | 1.89 | 12.7 | 7.01 | 6.62 | 1.21 | 5.33 |
| Birdgrove 511 | Average | 5.21 | 909 | 2.60 | 402 | 1.17 | 2.51 | 45.2 | 17.2 | 149 | 68.7 | 72.6 | 32.5 | 207 |
| n=24 (19) | 2sd | 0.87 | 238 | 3.44 | 74.2 | 0.55 | 1.89 | 13.4 | 2.80 | 38.6 | 17.2 | 17.9 | 6.83 | 60.6 |
| Mangahou 302 | Average | 5.23 | 1234 | 8.66 | 366 | 2.30 | 8.07 | 38.3 | 17.0 | 149 | 113 | 114 | 22.5 | 207 |
| n=24 (19) | 2sd | 0.55 | 266 | 4.79 | 101 | 0.73 | 19.2 | 15.0 | 1.81 | 23.0 | 16.6 | 15.3 | 5.06 | 39.2 |
| Ototaka 521 | Average | 6.25 | 1475 | 3.62 | 638 | 1.13 | 2.75 | 55.2 | 19.1 | 143 | 109 | 108 | 37.1 | 301 |
| n=21 (19) | 2sd | 0.94 | 481 | 4.46 | 144 | 0.45 | 0.84 | 13.0 | 2.27 | 19.7 | 22.5 | 24.0 | 4.83 | 54.6 |
| Hikuroa Pumice m | Average | 15.9 | 472 | 0.29 | 325 | | 2.09 | 46.6 | 14.9 | 129 | 56.5 | 56.3 | 25.5 | 114 |
| n=18 (16) | 2sd | 1.70 | 54 | 1.46 | 25.2 | | 1.62 | 4.9 | 1.67 | 5.1 | 6.35 | 5.62 | 0.55 | 12.5 |




**Table 3 (cont.) Average and standard deviation for all tephra, see SM Table 1 for full dataset.**

| Tephra name | | Zr$^{91}$ (ppm) | Nb (ppm) | Mo (ppm) | Cs (ppm) | Ba$^{137}$ (ppm) | Ba$^{138}$ (ppm) | La (ppm) | Ce (ppm) | Pr (ppm) | Nd (ppm) | Sm (ppm) | Eu$^{151}$ (ppm) | Eu$^{153}$ (ppm) |
|---|---|---|---|---|---|---|---|---|---|---|---|---|---|---|
| Kaharoa | Average | 850 | 7.95 | 1.51 | 5.38 | 945 | 933 | 21.0 | 46.4 | 5.08 | 19.0 | 3.99 | 0.55 | 0.59 |
| n=25 (18) | 2sd | 3282 | 1.99 | 0.65 | 1.81 | 243 | 208 | 4.18 | 9.23 | 0.97 | 3.89 | 1.35 | 0.20 | 0.20 |
| Taupo Y5 | Average | 168 | 7.17 | 1.07 | 6.07 | 609 | 601 | 21.3 | 45.4 | 5.25 | 20.4 | 4.24 | 0.88 | 0.92 |
| n=25 (20) | 2sd | 61.6 | 1.97 | 0.30 | 1.69 | 141 | 141 | 5.37 | 11.6 | 1.45 | 5.65 | 1.22 | 0.31 | 0.31 |
| Waimihia K3 | Average | 222 | 8.53 | 1.28 | 5.49 | 619 | 625 | 25.6 | 61.2 | 6.55 | 25.7 | 5.36 | 1.10 | 1.19 |
| n=22 (20) | 2sd | 33.6 | 1.82 | 0.22 | 1.41 | 116 | 123 | 4.42 | 11.3 | 1.19 | 5.27 | 1.19 | 0.21 | 0.20 |
| Unit K | Average | 219 | 8.99 | 1.67 | 6.24 | 655 | 653 | 26.4 | 56.2 | 6.37 | 26.0 | 5.76 | 1.12 | 1.14 |
| n=24 (20) | 2sd | 25.4 | 1.14 | 0.53 | 1.46 | 75.8 | 71.6 | 2.95 | 6.29 | 0.84 | 3.76 | 0.86 | 0.15 | 0.12 |
| Whakatane K5 | Average | | | | | | | | | | | | | |
| n=21 | 2sd | | | | | | | | | | | | | |
| Whakatane - P | Average | 175 | 8.83 | 1.75 | 8.76 | 750 | 737 | 31.6 | 66.7 | 7.66 | 30.2 | 6.18 | 1.00 | 0.96 |
| n=25 (21) | 2sd | 69.7 | 1.42 | 0.44 | 1.61 | 149 | 141 | 20.8 | 49.4 | 5.72 | 23.7 | 5.13 | 0.32 | 0.29 |
| Tuhua K6 | Average | 1178 | 102 | 10.6 | 6.08 | 161 | 158 | 85.6 | 182 | 21.6 | 86.8 | 20.1 | 2.13 | 1.90 |
| n=23 (22) | 2sd | 486 | 41.5 | 4.16 | 1.29 | 356 | 349 | 27.8 | 61.7 | 7.42 | 30.5 | 6.93 | 0.67 | 0.61 |
| Mamaku K7 | Average | 101 | 8.65 | 2.24 | 6.88 | 968 | 959 | 26.5 | 53.5 | 6.11 | 21.9 | 3.84 | 0.71 | 0.73 |
| n=23 (23) | 2sd | 18.3 | 1.28 | 0.95 | 1.34 | 161 | 159 | 3.4 | 7.00 | 0.94 | 2.64 | 0.61 | 0.20 | 0.18 |
| Rotoma - P | Average | | | | | | | | | | | | | |
| n=24 | 2sd | | | | | | | | | | | | | |
| Rotoma - K8 - D | Average | 98.6 | 8.84 | 1.81 | 5.68 | 970 | 940 | 26.1 | 54.7 | 6.22 | 23.8 | 4.54 | 0.77 | 0.82 |
| n=23 (20) | 2sd | 10.5 | 0.95 | 0.25 | 0.61 | 100 | 88.3 | 2.23 | 4.97 | 0.71 | 2.53 | 0.51 | 0.10 | 0.13 |
| Opepe K9 | Average | 236 | 8.82 | 1.94 | 8.13 | 700 | 674 | 26.1 | 56.4 | 6.38 | 25.6 | 5.15 | 1.01 | 1.05 |
| n=14 (8) | 2sd | 25.0 | 0.57 | 0.33 | 1.16 | 69.1 | 66.3 | 3.08 | 5.72 | 0.64 | 2.65 | 0.65 | 0.18 | 0.08 |
| Poronui K10 | Average | 231 | 9.16 | 1.83 | 7.48 | 738 | 737 | 28.1 | 57.9 | 6.65 | 26.7 | 5.41 | 1.06 | 0.98 |
| n=18 (15) | 2sd | 33.8 | 1.88 | 0.27 | 1.20 | 149 | 155 | 5.71 | 11.1 | 1.32 | 6.15 | 1.26 | 0.28 | 0.18 |
| Karapiti K11 | Average | 189 | 8.41 | 1.73 | 7.99 | 675 | 688 | 25.8 | 55.2 | 6.03 | 25.4 | 5.19 | 0.99 | 1.00 |
| n=25 (11) | 2sd | 37.6 | 1.62 | 0.38 | 1.86 | 135 | 135 | 4.61 | 9.68 | 1.11 | 4.94 | 1.03 | 0.16 | 0.15 |
| Waiohau - P | Average | 84.3 | 7.71 | 1.38 | 4.77 | 858 | 860 | 22.5 | 47.5 | 5.04 | 19.4 | 4.12 | 0.69 | 0.68 |
| n=19 (19) | 2sd | 5.73 | 0.30 | 1.89 | 0.50 | 38.6 | 48.5 | 1.41 | 2.77 | 0.36 | 1.54 | 0.61 | 0.13 | 0.20 |
| Waiohau - K14b D | Average | 92.6 | 7.75 | 1.82 | 4.53 | 867 | 861 | 22.4 | 47.6 | 5.10 | 19.6 | 3.76 | 0.73 | 0.81 |
| n=10 (10) | 2sd | 5.46 | 0.36 | 1.24 | 0.22 | 18.4 | 18.4 | 0.72 | 0.99 | 0.25 | 0.73 | 0.68 | 0.12 | 0.25 |
| Rotorua - P | Average | 132 | 8.12 | 2.20 | 5.31 | 793 | 812 | 23.4 | 49.3 | 5.30 | 20.1 | 4.05 | 0.62 | 0.68 |
| n=29 (28) | 2sd | 41.9 | 0.98 | 2.84 | 1.11 | 36.0 | 39.8 | 3.43 | 7.96 | 0.99 | 3.68 | 1.18 | 0.21 | 0.29 |
| Rotorua -K15 D | Average | 79.4 | 8.30 | 2.47 | 7.15 | 821 | 819 | 25.9 | 53.0 | 5.38 | 18.8 | 3.53 | 0.45 | 0.63 |
| n=10 (10) | 2sd | 36.3 | 0.93 | 1.13 | 1.17 | 43.2 | 44.7 | 3.19 | 6.48 | 0.74 | 2.74 | 0.50 | 0.22 | 0.41 |
| Rerewhakaaitu - P | Average | 89.3 | 7.69 | 2.31 | 5.89 | 794 | 798 | 24.1 | 50.3 | 5.25 | 19.5 | 4.04 | 0.45 | 0.54 |
| n=21 (19) | 2sd | 29.0 | 1.11 | 2.02 | 1.08 | 64.3 | 65.8 | 3.11 | 7.61 | 0.90 | 5.12 | 1.49 | 0.16 | 0.26 |
| Rerewhakaaitu -K1 | Average | 76.1 | 7.94 | 2.18 | 7.73 | 849 | 836 | 24.9 | 51.2 | 5.00 | 18.2 | 2.70 | 0.30 | 0.51 |
| n=11 (10) | 2sd | 14.2 | 1.24 | 1.61 | 3.07 | 50.9 | 43.8 | 1.29 | 2.88 | 0.24 | 2.22 | 0.80 | 0.30 | 0.37 |
| Okareka | Average | 115 | 8.58 | 1.92 | 4.78 | 1083 | 1036 | 25.6 | 51.1 | 5.79 | 19.5 | 3.72 | 0.51 | 0.55 |
| n=22 (22) | 2sd | 20.4 | 0.75 | 0.39 | 0.64 | 112 | 108 | 2.81 | 4.95 | 0.69 | 1.81 | 0.55 | 0.10 | 0.12 |
| Te Rere | Average | 132 | 11.1 | 2.32 | 7.41 | 805 | 785 | 30.9 | 66.5 | 7.71 | 28.9 | 6.09 | 0.66 | 0.61 |
| n=24 (20) | 2sd | 48.9 | 1.90 | 0.31 | 1.14 | 119 | 111 | 4.80 | 11.2 | 1.64 | 4.83 | 1.14 | 0.19 | 0.11 |
| Kawakawa/Oruanui | Average | 147 | 7.55 | 1.77 | 7.48 | 663 | 653 | 22.9 | 45.4 | 5.01 | 18.6 | 3.72 | 0.68 | 0.62 |
| n=25 (23) | 2sd | 49.6 | 1.00 | 0.41 | 1.40 | 70.0 | 74.0 | 2.83 | 5.62 | 0.62 | 2.71 | 0.75 | 0.20 | 0.17 |
| Poihipi | Average | 111 | 9.62 | 2.05 | 7.12 | 962 | 949 | 27.6 | 60.3 | 6.20 | 22.4 | 4.42 | 0.74 | 0.74 |
| n=20 (18) | 2sd | 29.1 | 1.76 | 0.48 | 1.51 | 232 | 212 | 3.44 | 25.2 | 1.08 | 3.89 | 0.96 | 0.45 | 0.34 |
| Okaia | Average | 153 | 8.30 | 1.91 | 8.44 | 717 | 721 | 24.7 | 46.5 | 5.29 | 20.3 | 3.99 | 0.71 | 0.69 |
| n=23 (23) | 2sd | 33.7 | 1.46 | 0.52 | 5.01 | 94.5 | 93.5 | 3.39 | 6.42 | 0.90 | 3.93 | 0.97 | 0.14 | 0.19 |
| Unit L | Average | 146 | 8.55 | 1.92 | 6.26 | 758 | 762 | 25.2 | 51.2 | 5.84 | 22.0 | 4.51 | 0.84 | 0.90 |
| n=23 (20) | 2sd | 15.0 | 1.83 | 0.42 | 1.70 | 109 | 101 | 3.64 | 8.47 | 1.22 | 5.05 | 1.14 | 0.31 | 0.29 |
| Awakeri | Average | 163 | 10.9 | 2.05 | 4.45 | 918 | 894 | 29.0 | 62.9 | 7.24 | 28.4 | 5.95 | 1.22 | 1.14 |
| n=21 (19) | 2sd | 39.1 | 2.83 | 0.53 | 1.30 | 209 | 205 | 7.11 | 14.7 | 1.66 | 6.03 | 1.62 | 0.48 | 0.28 |
| Mangaone | Average | 171 | 9.54 | 1.76 | 3.88 | 812 | 799 | 25.9 | 53.1 | 6.24 | 25.1 | 5.23 | 1.11 | 1.13 |
| n=24 (19) | 2sd | 15.6 | 0.77 | 0.31 | 0.79 | 50.8 | 47.7 | 1.86 | 2.93 | 0.37 | 2.04 | 0.57 | 0.13 | 0.09 |
| Hauparu | Average | 237 | 9.15 | 1.65 | 3.83 | 986 | 979 | 24.1 | 46.6 | 5.48 | 22.7 | 4.76 | 1.13 | 0.99 |
| n=21 (12) | 2sd | 29.8 | 1.09 | 0.32 | 1.14 | 794 | 765 | 2.87 | 5.23 | 0.73 | 2.60 | 0.89 | 0.17 | 0.14 |
| Maketu | Average | 304 | 10.8 | 1.96 | 2.66 | 786 | 782 | 27.4 | 51.8 | 5.97 | 29.0 | 6.14 | 1.58 | 1.45 |
| n=3 (1) | 2sd | | | | | | | | | | | | | |
| Ngamotu | Average | 184 | 7.85 | 1.80 | 2.62 | 734 | 732 | 20.3 | 43.5 | 4.81 | 19.8 | 4.08 | 0.98 | 0.93 |
| n=21 (19) | 2sd | 31.3 | 0.93 | 0.43 | 0.61 | 169 | 164 | 3.76 | 8.31 | 0.92 | 3.64 | 0.75 | 0.16 | 0.11 |
| Tahuna | Average | 102 | 8.42 | 2.15 | 7.92 | 941 | 923 | 26.4 | 50.8 | 5.31 | 18.5 | 3.42 | 0.42 | 0.50 |
| n=23 (17) | 2sd | 17.0 | 1.07 | 0.71 | 3.11 | 108 | 103 | 3.35 | 7.05 | 0.88 | 2.71 | 0.82 | 0.11 | 0.11 |
| Earthquake Flat Ig | Average | 96.45 | 8.94 | 2.04 | 4.87 | 1064 | 1053 | 26.0 | 51.7 | 5.58 | 19.4 | 3.66 | 0.51 | 0.53 |
| n=24 (11) | 2sd | 8.53 | 0.90 | 0.43 | 1.74 | 87.0 | 80.1 | 2.69 | 3.72 | 0.55 | 1.49 | 0.51 | 0.10 | 0.12 |
| Rotoehu tephra | Average | 116 | 8.73 | 1.88 | 3.64 | 1043 | 1056 | 25.6 | 51.8 | 5.54 | 20.6 | 3.81 | 0.64 | 0.66 |
| n=21 (14) | 2sd | 49.2 | 0.80 | 0.46 | 0.82 | 125 | 121 | 2.75 | 6.00 | 0.62 | 2.10 | 0.43 | 0.24 | 0.25 |
| Rotoiti Ig | Average | 91.6 | 8.42 | 1.77 | 3.82 | 1065 | 1066 | 25.2 | 49.8 | 5.21 | 19.4 | 3.57 | 0.49 | 0.51 |
| n=23 (19) | 2sd | 11.1 | 1.02 | 0.40 | 0.65 | 128 | 123 | 3.09 | 5.15 | 0.73 | 2.92 | 0.65 | 0.12 | 0.14 |
| Ararata Gully 318 | Average | 145 | 9.62 | 3.80 | 7.96 | 827 | 829 | 28.3 | 58.4 | 6.25 | 23.9 | 4.27 | 0.71 | 0.58 |
| n=24 (15) | 2sd | 30.2 | 0.98 | 2.10 | 1.15 | 96.6 | 91.2 | 2.43 | 4.73 | 0.65 | 2.93 | 0.61 | 0.10 | 0.16 |
| Kakariki 272 | Average | 106 | 8.52 | 2.59 | 5.68 | 869 | 878 | 25.3 | 50.1 | 5.42 | 19.8 | 3.99 | 0.63 | 0.65 |
| n=25 (18) | 2sd | 12.4 | 0.81 | 0.88 | 0.86 | 245 | 246 | 3.03 | 4.75 | 0.59 | 2.61 | 0.62 | 0.12 | 0.13 |
| Fordell 449 | Average | 138 | 11.8 | 2.04 | 8.60 | 874 | 884 | 33.7 | 71.1 | 8.21 | 31.1 | 6.53 | 0.73 | 0.75 |
| n=25 (19) | 2sd | 32.5 | 2.67 | 0.78 | 2.20 | 194 | 189 | 7.59 | 15.7 | 1.85 | 6.76 | 1.53 | 0.18 | 0.28 |
| Upper Griffin Road | Average | 164 | 9.16 | 1.87 | 7.66 | 714 | 723 | 29.3 | 57.3 | 6.82 | 24.8 | 4.72 | 0.77 | 0.86 |
| n=21 (17) | 2sd | 55.8 | 1.19 | 0.32 | 0.76 | 131 | 52.9 | 6.80 | 4.18 | 0.75 | 2.74 | 0.80 | 0.08 | 0.11 |
| Lower Griffin Road | Average | 193 | 8.25 | 2.07 | 5.61 | 743 | 749 | 23.6 | 44.9 | 5.02 | 18.8 | 3.82 | 0.84 | 0.91 |
| n=25 (21) | 2sd | 53.0 | 2.41 | 0.31 | 0.83 | 170 | 183 | 2.84 | 3.06 | 0.82 | 3.10 | 0.77 | 0.19 | 0.22 |
| Onepuhi 267 | Average | 162 | 9.67 | 2.41 | 8.38 | 790 | 783 | 26.3 | 52.5 | 5.99 | 22.9 | 4.68 | 0.78 | 0.80 |
| n=23 (10) | 2sd | 79.4 | 2.65 | 0.58 | 2.03 | 208 | 211 | 5.98 | 13.4 | 1.39 | 7.48 | 2.22 | 0.22 | 0.25 |
| Kupe 481 | Average | 144 | 9.58 | 2.18 | 7.69 | 820 | 1055 | 26.4 | 59.7 | 7.13 | 24.0 | 4.78 | 0.70 | 0.72 |
| n=24 (17) | 2sd | 36.3 | 2.63 | 0.58 | 2.50 | 240 | 580 | 8.18 | 15.2 | 4.08 | 7.98 | 1.51 | 0.17 | 0.16 |
| Kaukatea 232 | Average | 210 | 12.0 | 2.11 | 7.37 | 841 | 704 | 30.6 | 66.1 | 6.75 | 29.6 | 6.40 | 1.05 | 1.13 |
| n=25 (20) | 2sd | 17.8 | 1.06 | 0.25 | 0.63 | 65.4 | 119 | 2.19 | 3.67 | 1.13 | 2.30 | 0.69 | 0.09 | 0.12 |
| Potaka 305 | Average | 130 | 10.1 | 2.24 | 9.72 | 955 | 1087 | 30.1 | 60.4 | 8.95 | 23.5 | 4.58 | 0.66 | 0.70 |
| n=24 (21) | 2sd | 15.8 | 1.14 | 0.33 | 1.08 | 91.3 | 365 | 3.16 | 7.31 | 4.05 | 2.38 | 0.72 | 0.12 | 0.12 |
| Rewa 304 | Average | 216 | 10.7 | 1.71 | 7.31 | 765 | 773 | 29.0 | 59.4 | 6.46 | 26.6 | 5.73 | 1.09 | 1.05 |
| n=25 (18) | 2sd | 38.5 | 2.10 | 0.37 | 0.87 | 106 | 136 | 4.49 | 7.56 | 0.74 | 4.66 | 1.13 | 0.17 | 0.14 |
| Mangapipi 510 | Average | 313 | 12.8 | 1.97 | 8.62 | 816 | 799 | 30.0 | 66.3 | 7.86 | 31.1 | 6.31 | 1.10 | 1.17 |
| n=24 (11) | 2sd | 29.5 | 1.72 | 0.27 | 0.98 | 74.7 | 67.5 | 1.64 | 5.28 | 0.58 | 3.19 | 0.65 | 0.14 | 0.16 |
| Pakihikura 303 | Average | 101 | 6.32 | 1.29 | 7.33 | 824 | 828 | 23.0 | 46.8 | 4.56 | 17.1 | 3.37 | 0.56 | 0.63 |
| n=25 (21) | 2sd | 7.18 | 0.66 | 0.25 | 0.80 | 73.6 | 65.6 | 1.63 | 3.52 | 0.36 | 1.57 | 0.48 | 0.06 | 0.07 |
| Birdgrove 511 | Average | 204 | 10.1 | 2.26 | 10.5 | 756 | 764 | 29.4 | 60.2 | 6.66 | 24.7 | 5.30 | 0.82 | 0.83 |
| n=24 (19) | 2sd | 59.2 | 2.56 | 0.59 | 3.22 | 160 | 170 | 6.85 | 12.8 | 1.57 | 6.22 | 1.43 | 0.27 | 0.24 |
| Mangahou 302 | Average | 208 | 8.63 | 2.51 | 10.1 | 787 | 790 | 25.5 | 49.3 | 5.38 | 19.5 | 3.85 | 0.71 | 0.72 |
| n=24 (19) | 2sd | 36.5 | 1.48 | 1.85 | 1.76 | 100 | 95.5 | 3.56 | 6.76 | 0.84 | 3.43 | 0.96 | 0.13 | 0.14 |
| Ototaka 521 | Average | 310 | 11.8 | 2.17 | 9.49 | 781 | 765 | 28.7 | 59.2 | 7.06 | 28.3 | 5.86 | 1.03 | 1.09 |
| n=21 (19) | 2sd | 65.6 | 2.06 | 0.39 | 1.47 | 117 | 108 | 4.11 | 9.04 | 1.15 | 4.50 | 1.10 | 0.19 | 0.20 |
| Hikuroa Pumice m | Average | 115 | 7.24 | | 6.17 | 805 | 806 | 23.3 | 49.7 | 5.36 | 20.8 | 4.40 | 0.52 | 0.69 |
| n=18 (16) | 2sd | 14.4 | 0.34 | | 0.32 | 30.2 | 21.2 | 0.82 | 1.05 | 0.25 | 1.08 | 0.33 | 0.12 | 0.23 |





**Table 3 (cont.) Average and standard deviation for all tephra, see SM Table 1 for full dataset.**

| Tephra name | | Gd (ppm) | Tb (ppm) | Dy (ppm) | Ho (ppm) | Er (ppm) | Tm (ppm) | Yb (ppm) | Lu (ppm) | Hf (ppm) | Ta (ppm) | W (ppm) | Pb (ppm) | Th (ppm) | U (ppm) |
|---|---|---|---|---|---|---|---|---|---|---|---|---|---|---|---|
| Kaharoa | Average | 3.95 | 0.64 | 5.00 | 1.19 | 3.97 | 0.75 | 6.15 | 1.58 | 18.4 | 0.75 | 1.55 | 17.7 | 12.3 | 5.27 |
| n=25 (18) | 2sd | 1.32 | 0.18 | 3.47 | 1.31 | 5.31 | 1.27 | 11.5 | 3.11 | 65.3 | 0.18 | 0.54 | 4.48 | 5.18 | 8.96 |
| Taupo Y5 | Average | 4.31 | 0.66 | 4.21 | 0.86 | 2.72 | 0.40 | 2.94 | 0.74 | 4.35 | 0.53 | 1.45 | 17.3 | 9.16 | 2.42 |
| n=25 (20) | 2sd | 1.28 | 0.21 | 1.43 | 0.29 | 0.83 | 0.14 | 1.05 | 0.22 | 1.54 | 0.15 | 0.46 | 4.87 | 2.48 | 0.63 |
| Waimihia K3 | Average | 5.18 | 0.84 | 5.10 | 1.13 | 3.30 | 0.52 | 3.46 | 0.95 | 5.69 | 0.66 | 1.80 | 19.9 | 10.6 | 2.59 |
| n=22 (20) | 2sd | 1.25 | 0.20 | 1.01 | 0.24 | 0.63 | 0.11 | 0.80 | 0.21 | 0.89 | 0.13 | 0.53 | 3.29 | 1.64 | 0.46 |
| Unit K | Average | 5.61 | 0.86 | 5.61 | 1.18 | 3.41 | 0.50 | 3.60 | 0.55 | 6.41 | 0.70 | 1.70 | 20.5 | 10.9 | 2.89 |
| n=24 (20) | 2sd | 0.91 | 0.12 | 0.66 | 0.16 | 0.46 | 0.08 | 0.59 | 0.08 | 0.78 | 0.11 | 0.37 | 5.00 | 1.39 | 0.41 |
| Whakatane K5 | Average | | | | | | | | | | | | | | |
| n=21 | 2sd | | | | | | | | | | | | | | |
| Whakatane - P | Average | 5.83 | 0.88 | 5.31 | 1.16 | 3.20 | 0.50 | 3.40 | 0.58 | 5.07 | 0.73 | 1.77 | 21.1 | 11.6 | 2.96 |
| n=25 (21) | 2sd | 2.86 | 0.36 | 1.67 | 0.31 | 0.99 | 0.18 | 0.92 | 0.18 | 2.27 | 0.18 | 0.72 | 4.94 | 2.52 | 0.54 |
| Tuhua K6 | Average | 21.3 | 3.41 | 23.0 | 4.81 | 14.2 | 2.24 | 14.0 | 2.07 | 23.6 | 6.69 | 2.01 | 28.0 | 17.7 | 5.89 |
| n=23 (22) | 2sd | 7.89 | 1.28 | 8.46 | 1.76 | 5.22 | 0.82 | 4.93 | 0.78 | 9.77 | 2.66 | 0.45 | 6.77 | 3.59 | 1.87 |
| Mamaku K7 | Average | 3.87 | 0.64 | 4.18 | 0.85 | 2.69 | 0.40 | 3.05 | 0.45 | 3.32 | 0.81 | 1.68 | 16.5 | 11.9 | 2.94 |
| n=23 (23) | 2sd | 0.98 | 0.18 | 0.81 | 0.14 | 0.39 | 0.10 | 0.51 | 0.11 | 0.47 | 0.09 | 0.42 | 3.46 | 1.54 | 0.47 |
| Rotoma - P | Average | | | | | | | | | | | | | | |
| n=24 | 2sd | | | | | | | | | | | | | | |
| Rotoma - K8 - D | Average | 4.15 | 0.66 | 4.17 | 0.92 | 2.90 | 0.42 | 3.24 | 0.48 | 3.31 | 0.77 | 1.47 | 16.1 | 11.5 | 2.86 |
| n=23 (20) | 2sd | 1.63 | 0.09 | 0.50 | 0.11 | 0.45 | 0.07 | 0.38 | 0.07 | 0.51 | 0.10 | 0.25 | 2.29 | 1.13 | 0.39 |
| Opepe K9 | Average | 5.78 | 0.86 | 5.20 | 1.09 | 3.57 | 0.49 | 3.27 | 0.53 | 5.89 | 0.69 | 1.55 | 18.9 | 11.1 | 2.88 |
| n=14 (8) | 2sd | 1.17 | 0.11 | 0.27 | 0.17 | 0.41 | 0.11 | 0.49 | 0.09 | 0.57 | 0.10 | 0.31 | 1.96 | 2.15 | 0.37 |
| Poronui K10 | Average | 9.40 | 0.89 | 5.69 | 1.24 | 3.53 | 0.56 | 3.89 | 0.57 | 6.13 | 0.73 | 1.79 | 21.2 | 12.3 | 3.03 |
| n=18 (15) | 2sd | 9.69 | 0.19 | 1.18 | 0.23 | 0.83 | 0.14 | 0.72 | 0.13 | 0.94 | 0.16 | 0.39 | 3.82 | 2.68 | 0.61 |
| Karapiti K11 | Average | 5.48 | 0.78 | 5.18 | 1.07 | 4.78 | 0.53 | 3.40 | 0.52 | 5.70 | 0.66 | 1.66 | 19.6 | 12.0 | 2.96 |
| n=25 (11) | 2sd | 1.25 | 0.15 | 1.07 | 0.20 | 0.98 | 0.11 | 0.66 | 0.09 | 1.09 | 0.16 | 0.35 | 3.75 | 2.13 | 0.59 |
| Waiohau - P | Average | 3.53 | 0.58 | 3.74 | 0.78 | 2.34 | 0.38 | 2.56 | 0.36 | 2.69 | 0.62 | 1.36 | 15.3 | 9.37 | 2.37 |
| n=19 (19) | 2sd | 0.58 | 0.12 | 0.50 | 0.12 | 0.26 | 0.06 | 0.40 | 0.06 | 0.32 | 0.10 | 0.28 | 0.88 | 0.65 | 0.18 |
| Waiohau - K14b D | Average | 3.17 | 0.55 | 3.72 | 0.71 | 2.40 | 0.31 | 2.60 | 0.40 | 3.08 | 0.65 | 1.50 | 14.9 | 9.72 | 2.47 |
| n=10 (10) | 2sd | 0.58 | 0.12 | 0.43 | 0.10 | 0.23 | 0.05 | 0.38 | 0.08 | 0.24 | 0.09 | 0.30 | 0.91 | 0.47 | 0.10 |
| Rotorua - P | Average | 3.65 | 0.64 | 4.09 | 0.87 | 2.47 | 0.40 | 2.77 | 0.38 | 3.70 | 0.68 | 1.52 | 16.1 | 10.8 | 2.73 |
| n=29 (28) | 2sd | 1.11 | 0.23 | 1.16 | 0.21 | 0.72 | 0.11 | 0.58 | 0.14 | 0.70 | 0.13 | 0.36 | 2.27 | 1.90 | 0.48 |
| Rotorua -K15 D | Average | 2.83 | 0.49 | 3.39 | 0.70 | 2.14 | 0.33 | 2.44 | 0.37 | 2.69 | 0.81 | 2.14 | 18.0 | 13.4 | 3.38 |
| n=10 (10) | 2sd | 0.88 | 0.10 | 0.51 | 0.10 | 0.43 | 0.10 | 0.48 | 0.09 | 0.80 | 0.15 | 0.47 | 1.68 | 2.75 | 0.57 |
| Rerewhakaaitu - P | Average | 3.25 | 0.47 | 3.59 | 0.80 | 2.32 | 0.38 | 2.66 | 0.38 | 2.85 | 0.73 | 1.70 | 16.9 | 11.6 | 3.05 |
| n=21 (19) | 2sd | 1.13 | 0.22 | 1.31 | 0.33 | 0.79 | 0.14 | 0.75 | 0.12 | 0.72 | 0.13 | 0.42 | 2.37 | 2.19 | 0.59 |
| Rerewhakaaitu -K1 | Average | 2.90 | 0.51 | 3.07 | 0.73 | 2.03 | 0.39 | 2.15 | 0.35 | 2.74 | 0.77 | 1.90 | 18.8 | 13.2 | 3.32 |
| n=11 (10) | 2sd | 0.43 | 0.20 | 0.63 | 0.25 | 0.38 | 0.12 | 0.56 | 0.09 | 0.33 | 0.12 | 0.24 | 3.49 | 1.51 | 0.41 |
| Okareka | Average | 3.64 | 0.54 | 3.33 | 0.75 | 2.93 | 0.37 | 2.85 | 0.41 | 3.07 | 0.81 | 1.37 | 13.2 | 10.7 | 2.47 |
| n=22 (22) | 2sd | 0.59 | 0.10 | 0.45 | 0.12 | 1.12 | 0.07 | 0.52 | 0.08 | 0.44 | 0.16 | 0.38 | 1.39 | 1.24 | 0.33 |
| Te Rere | Average | 5.87 | 0.97 | 6.14 | 1.26 | 4.51 | 0.64 | 4.34 | 0.62 | 4.32 | 1.25 | 1.74 | 19.7 | 14.0 | 3.55 |
| n=24 (20) | 2sd | 1.10 | 0.16 | 1.10 | 0.21 | 1.90 | 0.16 | 0.90 | 0.14 | 1.05 | 0.54 | 0.23 | 2.75 | 1.91 | 0.57 |
| Kawakawa/Oruanui | Average | 3.59 | 0.60 | 3.54 | 0.75 | 2.34 | 0.38 | 2.52 | 0.39 | 4.15 | 0.62 | 1.59 | 15.3 | 11.2 | 2.85 |
| n=25 (23) | 2sd | 0.70 | 0.12 | 0.71 | 0.12 | 0.71 | 0.07 | 0.46 | 0.10 | 0.81 | 0.16 | 0.37 | 2.10 | 1.59 | 0.40 |
| Poihipi | Average | 3.75 | 0.64 | 3.97 | 0.86 | 2.54 | 0.40 | 3.14 | 0.42 | 3.64 | 0.91 | 1.85 | 21.5 | 13.3 | 3.34 |
| n=20 (18) | 2sd | 0.85 | 0.18 | 1.18 | 0.19 | 0.49 | 0.10 | 0.75 | 0.10 | 0.75 | 0.24 | 0.57 | 12.0 | 2.99 | 0.73 |
| Okaia | Average | 3.95 | 0.58 | 4.00 | 0.80 | 2.59 | 0.39 | 2.88 | 0.41 | 4.20 | 0.71 | 1.66 | 16.9 | 12.0 | 3.11 |
| n=23 (23) | 2sd | 0.71 | 0.13 | 0.82 | 0.16 | 0.66 | 0.13 | 0.56 | 0.06 | 0.73 | 0.13 | 0.37 | 2.93 | 1.90 | 0.58 |
| Unit L | Average | 4.34 | 0.68 | 4.34 | 0.94 | 3.06 | 0.47 | 3.22 | 0.49 | 4.32 | 0.74 | 1.54 | 16.0 | 11.7 | 2.90 |
| n=23 (20) | 2sd | 1.20 | 0.21 | 1.50 | 0.30 | 0.98 | 0.13 | 0.88 | 0.17 | 0.58 | 0.15 | 0.25 | 2.74 | 1.85 | 0.47 |
| Awakeri | Average | 5.60 | 0.93 | 5.69 | 1.17 | 3.68 | 0.59 | 3.93 | 0.59 | 4.69 | 0.79 | 1.48 | 17.6 | 10.5 | 2.56 |
| n=21 (19) | 2sd | 1.51 | 0.26 | 1.26 | 0.30 | 0.87 | 0.16 | 0.91 | 0.16 | 1.00 | 0.23 | 0.41 | 4.30 | 2.25 | 0.61 |
| Mangaone | Average | 5.19 | 0.81 | 5.23 | 1.09 | 3.40 | 0.54 | 3.68 | 0.56 | 4.79 | 0.71 | 1.20 | 14.3 | 8.94 | 2.16 |
| n=24 (19) | 2sd | 0.64 | 0.11 | 0.63 | 0.14 | 0.46 | 0.07 | 0.39 | 0.08 | 0.54 | 0.07 | 0.20 | 1.54 | 0.85 | 0.18 |
| Hauparu | Average | 4.41 | 0.76 | 4.89 | 1.01 | 3.03 | 0.49 | 3.34 | 0.48 | 5.58 | 0.61 | 1.12 | 12.6 | 8.08 | 1.95 |
| n=21 (12) | 2sd | 0.89 | 0.10 | 0.61 | 0.13 | 0.38 | 0.08 | 0.56 | 0.12 | 0.90 | 0.15 | 0.19 | 2.40 | 1.22 | 0.41 |
| Maketu | Average | 6.15 | 1.05 | 7.60 | 1.33 | 4.20 | 0.76 | 4.27 | 0.55 | 7.38 | 0.72 | 1.13 | 13.9 | 7.60 | 1.71 |
| n=3 (1) | 2sd | | | | | | | | | | | | | | |
| Ngamotu | Average | 3.87 | 0.62 | 3.78 | 0.78 | 2.50 | 0.38 | 2.87 | 0.45 | 4.58 | 0.58 | 0.99 | 11.8 | 7.02 | 1.86 |
| n=21 (19) | 2sd | 0.89 | 0.13 | 0.84 | 0.16 | 0.57 | 0.11 | 0.71 | 0.11 | 0.83 | 0.10 | 0.17 | 1.61 | 1.89 | 0.38 |
| Tahuna | Average | 3.11 | 0.48 | 3.06 | 0.67 | 2.18 | 0.34 | 2.49 | 0.38 | 3.12 | 0.86 | 1.55 | 16.4 | 13.0 | 3.26 |
| n=23 (17) | 2sd | 0.49 | 0.10 | 0.59 | 0.15 | 0.51 | 0.06 | 0.65 | 0.09 | 0.69 | 0.20 | 0.47 | 4.31 | 3.67 | 1.00 |
| Earthquake Flat Ig | Average | 3.05 | 0.53 | 3.08 | 0.61 | 2.08 | 0.32 | 2.31 | 0.38 | 2.71 | 0.76 | 1.50 | 14.8 | 11.4 | 2.93 |
| n=24 (11) | 2sd | 0.47 | 0.09 | 0.46 | 0.11 | 0.44 | 0.11 | 0.34 | 0.07 | 0.50 | 0.19 | 0.61 | 2.79 | 3.42 | 0.87 |
| Rotoehu tephra | Average | 3.63 | 0.58 | 3.60 | 0.77 | 2.21 | 0.38 | 2.75 | 0.45 | 3.21 | 0.72 | 1.40 | 13.8 | 9.95 | 2.56 |
| n=21 (14) | 2sd | 0.76 | 0.11 | 0.65 | 0.15 | 0.45 | 0.09 | 0.48 | 0.08 | 1.06 | 0.10 | 0.24 | 2.36 | 1.41 | 0.41 |
| Rotoiti Ig | Average | 3.06 | 0.46 | 3.11 | 0.63 | 1.96 | 0.30 | 2.57 | 0.34 | 2.67 | 0.70 | 1.33 | 12.9 | 9.52 | 2.36 |
| n=23 (19) | 2sd | 0.75 | 0.07 | 0.41 | 0.11 | 0.34 | 0.06 | 0.33 | 0.06 | 0.43 | 0.16 | 0.38 | 1.96 | 1.25 | 0.34 |
| Ararata Gully 318 | Average | 4.08 | 0.60 | 4.26 | 0.85 | 2.53 | 0.40 | 2.78 | 0.41 | 4.04 | 0.85 | 1.76 | 22.4 | 13.5 | 3.37 |
| n=24 (15) | 2sd | 0.72 | 0.08 | 0.53 | 0.10 | 0.26 | 0.06 | 0.50 | 0.07 | 0.52 | 0.15 | 0.38 | 3.96 | 1.26 | 0.30 |
| Kakariki 272 | Average | 3.39 | 0.55 | 3.48 | 0.72 | 2.22 | 0.37 | 2.41 | 0.40 | 3.39 | 0.77 | 1.51 | 19.5 | 13.7 | 3.06 |
| n=25 (18) | 2sd | 0.58 | 0.11 | 0.54 | 0.11 | 0.37 | 0.07 | 0.28 | 0.06 | 0.37 | 0.15 | 0.34 | 2.29 | 1.86 | 0.50 |
| Fordell 449 | Average | 6.38 | 1.05 | 6.69 | 1.35 | 3.93 | 0.62 | 4.30 | 0.66 | 4.74 | 1.08 | 2.03 | 24.9 | 14.9 | 3.68 |
| n=25 (19) | 2sd | 1.44 | 0.24 | 1.98 | 0.32 | 0.91 | 0.18 | 1.07 | 0.17 | 1.12 | 0.27 | 0.58 | 4.97 | 3.40 | 0.81 |
| Upper Griffin Road | Average | 5.04 | 0.82 | 5.23 | 1.23 | 3.64 | 0.53 | 3.37 | 0.57 | 4.82 | 0.76 | 1.67 | 17.3 | 12.3 | 3.10 |
| n=21 (17) | 2sd | 1.38 | 0.09 | 1.37 | 0.14 | 0.38 | 0.08 | 0.77 | 0.07 | 0.37 | 0.12 | 0.28 | 5.29 | 1.01 | 0.31 |
| Lower Griffin Road | Average | 3.64 | 0.58 | 4.06 | 0.82 | 2.46 | 0.37 | 2.73 | 0.43 | 5.37 | 0.74 | 1.40 | 17.4 | 12.6 | 2.97 |
| n=25 (21) | 2sd | 0.56 | 0.11 | 0.69 | 0.13 | 0.43 | 0.06 | 0.42 | 0.07 | 1.39 | 0.25 | 0.37 | 6.91 | 2.77 | 0.51 |
| Onepuhi 267 | Average | 4.44 | 0.70 | 4.61 | 0.92 | 2.87 | 0.45 | 3.24 | 0.46 | 4.56 | 0.85 | 1.68 | 19.7 | 15.8 | 3.91 |
| n=23 (10) | 2sd | 1.65 | 0.35 | 1.79 | 0.41 | 1.01 | 0.22 | 1.21 | 0.14 | 2.06 | 0.21 | 0.33 | 3.29 | 6.35 | 0.98 |
| Kupe 481 | Average | 4.57 | 0.75 | 4.74 | 1.02 | 3.00 | 0.49 | 3.29 | 0.46 | 4.10 | 0.74 | 1.55 | 19.61 | 12.35 | 4.22 |
| n=24 (17) | 2sd | 1.56 | 0.25 | 1.69 | 0.37 | 1.09 | 0.18 | 1.25 | 0.17 | 1.20 | 0.27 | 0.60 | 5.02 | 4.52 | 2.95 |
| Kaukatea 232 | Average | 6.22 | 0.97 | 6.26 | 1.27 | 3.94 | 0.58 | 3.95 | 0.59 | 5.89 | 0.91 | 1.67 | 23.0 | 13.1 | 2.68 |
| n=25 (20) | 2sd | 0.59 | 0.09 | 0.47 | 0.09 | 0.33 | 0.07 | 0.41 | 0.06 | 0.49 | 0.09 | 0.25 | 2.11 | 0.85 | 0.43 |
| Potaka 305 | Average | 3.99 | 0.65 | 4.04 | 0.86 | 2.72 | 0.38 | 2.95 | 0.44 | 3.96 | 0.90 | 1.92 | 22.4 | 14.3 | 5.58 |
| n=24 (21) | 2sd | 0.78 | 0.08 | 0.51 | 0.13 | 0.38 | 0.06 | 0.38 | 0.07 | 0.58 | 0.12 | 0.34 | 2.92 | 1.90 | 3.11 |
| Rewa 304 | Average | 5.16 | 0.89 | 5.69 | 1.17 | 3.36 | 0.51 | 3.62 | 0.56 | 5.93 | 0.75 | 1.50 | 21.5 | 12.3 | 2.75 |
| n=25 (18) | 2sd | 0.88 | 0.15 | 0.95 | 0.19 | 0.42 | 0.10 | 0.75 | 0.09 | 0.85 | 0.10 | 0.43 | 3.78 | 1.81 | 0.40 |
| Mangapipi 510 | Average | 6.99 | 1.04 | 6.35 | 1.39 | 4.28 | 0.65 | 4.23 | 0.63 | 7.73 | 0.92 | 1.51 | 20.0 | 13.0 | 3.57 |
| n=24 (11) | 2sd | 0.93 | 0.11 | 0.58 | 0.10 | 0.36 | 0.06 | 0.44 | 0.04 | 0.74 | 0.14 | 0.24 | 1.98 | 1.42 | 0.41 |
| Pakihikura 303 | Average | 3.13 | 0.51 | 3.14 | 0.69 | 2.01 | 0.32 | 2.16 | 0.34 | 3.36 | 0.57 | 1.56 | 17.3 | 11.8 | 2.87 |
| n=25 (21) | 2sd | 0.39 | 0.08 | 0.32 | 0.07 | 0.28 | 0.05 | 0.22 | 0.06 | 0.35 | 0.09 | 0.32 | 2.46 | 0.69 | 0.28 |
| Birdgrove 511 | Average | 5.44 | 0.80 | 5.31 | 1.13 | 3.02 | 0.49 | 3.48 | 0.54 | 5.85 | 0.87 | 1.87 | 23.0 | 16.1 | 3.36 |
| n=24 (19) | 2sd | 1.53 | 0.20 | 1.26 | 0.24 | 0.71 | 0.11 | 0.84 | 0.12 | 1.46 | 0.22 | 0.56 | 5.51 | 3.52 | 1.26 |
| Mangahou 302 | Average | 3.72 | 0.57 | 3.74 | 0.77 | 2.41 | 0.35 | 2.62 | 0.39 | 5.39 | 0.81 | 1.49 | 20.1 | 14.8 | 3.74 |
| n=24 (19) | 2sd | 0.78 | 0.15 | 0.84 | 0.20 | 0.55 | 0.07 | 0.60 | 0.08 | 0.99 | 0.14 | 0.30 | 2.87 | 1.60 | 0.59 |
| Ototaka 521 | Average | 6.05 | 0.92 | 6.07 | 1.27 | 3.67 | 0.59 | 4.02 | 0.57 | 7.56 | 0.89 | 1.84 | 21.0 | 14.1 | 3.65 |
| n=21 (19) | 2sd | 1.10 | 0.21 | 1.12 | 0.19 | 0.76 | 0.11 | 0.63 | 0.14 | 1.53 | 0.19 | 0.36 | 3.76 | 2.68 | 0.73 |
| Hikuroa Pumice m | Average | 3.94 | 0.68 | 4.31 | 0.91 | 2.64 | 0.46 | 2.92 | 0.46 | 3.71 | 0.58 | 1.55 | 18.6 | 10.7 | 2.72 |
| n=18 (16) | 2sd | 0.43 | 0.10 | 0.39 | 0.10 | 0.23 | 0.07 | 0.26 | 0.06 | 0.32 | 0.09 | 0.31 | 0.49 | 0.46 | 0.19 |





**Table 4. Dominant ferroganesian mineral assemblages for late Quaternary silicic tephra deposits updated from Froggatt and Lowe 1990**

| Assemblage 1 **Hyp** +/- aug +/- hbl | Assemblage 2 Hyp + **hbl** +/- aug | Assemblage 3 Hyp + hbl + **bio** | Assemblage 4 Hyp + **cgt** +/- hbl | Assemblage 5 Hyp + **aug** +/- hbl | Assemblage 6 **Aegirine**\* |
|---|---|---|---|---|---|
| **Taupō VC** | **Okataina VC** | **Okataina VC** | **Okataina VC** | **Okataina VC** | **Tuhua VC (Mayor Is)** |
| Taupō (all) | Mamaku | Kaharoa | Whakatane | Hauparu | Tuhua |
| (Mapara) | Waiohau | Rotorua (upper) | Rotoma | Te Mahoe | |
| (Whakaipo) | Rotorua (lower) | Rerewhakaaitu | Rotoiti/Rotoehu (all) | Maketu | |
| Waimihia | Unit L | Okareka | | | |
| Unit K | Te Rere | Rotoiti/Rotoehu (upper) | | | |
| Motutere (all?) | (Omatoroa) | | | | |
| Opepe | Awakeri | **Kapenga VC** | | | |
| Poronui | Mangaone | Earthquake Flat | | | |
| Karapiti | Tahuna | | | | |
| | Ngamotu | **Maroa VC** | | | |
| | | Puketarata | | | |
| | **Taupō VC** | | | | |
| | Kawakawa (all) | | | | |
| | Poihipi | | | | |
| | Okaia | | | | |
| | (Tihoi) | | | | |
| | (Waihoroa) | | | | |
| | (Otake) | | | | |

The assemblages are listed with mineral species in order of abundance, the diagnostic mineral in each assemblage is in bold, tephra are listed multiple times
if their mineral assemblage changes through the eruption sequence, and deposits in brackets are not included in the TephraNZ database
\*Assembly 6 Aegirine +/- riebeckite +/- aenigmatite +/- olivine +/- tuhualite
Motutere was listed in Froggatt and Lowe 1990 as a single unit with Assemblage 1 mineraology,  but this has subsequently been redefined by Wilson 1993 into two
subunits G&H, which do not have their independent assemblages defined.



**Table 5. Geochemically similar tephra and their distinctions**

| Tephra | Ages (ka)[a] | Magma volume (km$^3$)* | Geochemical distinction | Other distinctions |
|---|---|---|---|---|
| Taupo (Unit Y, TVC) | 1,718 +/- 10 | 13.4 | Indistinguishable | age, volume, stratigraphic relationship |
| Waimihia (Unit S, TVC) | 3,382 +/- 50 | 5.1 | | |
| Waimihia (Unit S, TVC) | 3,382 +/- 50 | 5.1 | Lu, Sc, Mn, Co, Ba | age, volume, stratigraphic relationship to Stent Tephra |
| Unit K (TVC) | 5,088 +/- 73 | 0.12 | | |
| Waimihia (Unit S, TVC) | 3,401 +/- 108 | 5.1 | FeO vs. CaO, Na$_2$O+K$_2$O, Lu, Sc, Mn, Co, | age, volume, stratigraphic relationships |
| Poronui (Unit C, TVC) | 11,159 +/- 51 | 0.23 | | |
| Waimihia (Unit S, TVC) | 3,382 +/- 50 | 5.1 | FeO vs. CaO, Na$_2$O+K$_2$O, Lu, Sc, Mn, Co, | age, stratigraphic relationships |
| Karapiti (Unit B, TVC) | 11,501 +/- 104 | 0.42 | | |
| Unit K (TVC) | 5,088 +/- 73 | 0.12 | FeO vs. CaO, Na$_2$O+K$_2$O, Lu, Sc, Mn, Co, | age, stratigraphic relationships |
| Poronui (Unit C, TVC) | 11,170 +/- 115 | 0.23 | | |
| Unit K (TVC) | 5,088 +/- 73 | 0.12 | FeO vs. CaO, Na$_2$O+K$_2$O, Lu, Sc, Mn, Co, | age, stratigraphic relationships |
| Karapiti (Unit B, TVC) | 11,501 +/- 104 | 0.42 | | |
| Mamaku | 7,992 +/- 58 | 13.0 | Ba/Th vs. Rb/Sr, Rb/Zr vs. Rb/Sr | volume, chemistry |
| Rotoma | 9,472 +/- 40 | 8.0 | | |
| Poronui (Unit C, TVC) | 11,195 +/- 51 | 0.23 | Indistinguishable | |
| Karapiti (Unit B, TVC) | 11,501 +/- 104 | 0.4 | | |
| Waiohau | 14,018 +/- 91 | 3.3 | Ba, Hf, Ba/Zr vs. Ba/Th, Rb/Sr, Rb/Zr; Rotorua is bi-modal | volume, chemistry, mineralogy |
| Rotorua | 15,738 +/- 263 | 1.0 | | |
| Waiohau | 14,018 +/- 91 | 3.3 | Cs, La, Ce, Nd, Eu, Rb/Zr, Ba/Th; Rerewhakaaitu is bimodal | volume, chemistry, mineralogy |
| Rerewhakaaitu | 17,209 +/- 249 | 5.0 | | |
| Rotorua | 15,738 +/- 263 | 1.0 | Indistinguishable | age |
| Rerewhakaaitu | 17,209 +/- 249 | 5.0 | | |
| Poihipi | 28,446 +/- 670 | 0.5 | La/Yb vs. Ba/Y, Rb/Zr vs. Ba/Th | age |
| Tahuna | 39,268 +/- 1193 | 2.0[b] | | |
| Rotoma | 9,472 +/- 40 | 8.0 | Ba, Cs, Y, Sm, Nd, Pr, Er, Ho, Dy, Tb, Eu, Tm, Yb, Pb, U, Th | age, volume, stratigraphic relationships |
| Rotoehu | 45,170 +/- 3300 | 90[c] | | |
| KOT | 25,358 +/- 162 | 530.0 | Indistinguishable | volume, shard morphology |
| Okaia | 28,621 +/- 1428 | 3.0 | | |
| KOT | 25,358 +/- 162 | 530.0 | Unit L bimodal in Rb/Zr, Ba/Th, Ce/Th, Y/Th | volume, mineral geochemisty (Smith et al., 2002) |
| Unit L (Mangaone sgp OVC) | 31.4 ka | c. 7.0[d] | | |

[a] Ages from Table 1
* Volumes from Lowe et al., 2008 and references therein,
except for Tahuna, Rotoiti/Rotoehu, and Unit L (Mangaone Subgroup)
[b] Tahuna airfall volume estimate in km$^3$ from Froggatt and Lowe 1990
[c] Rotoehu airfall volume in km$^3$ from Froggatt and Lowe 1990
[d] Unit L airfall volume in km3 from Jurado-Chichay and Walker 2000