# Peer review of "TephraNZ: a major and trace element reference dataset for glassshard analyses from prominent Quaternary rhyolitic tephras in New Zealand, and implications for correlation"

_Geochronology, 2020_

## Referee Comment (RC1) · Maxim Portnyagin (Referee) · 2 Dec 2020

This paper reports a new self-consistent (obtained in one lab) dataset for major tephras in New Zealand and its discussion aimed at defining robust criteria to distinguish different tephras on the basis of chemical data. This work is important step toward a global database of tephra compositions and should be published. The dataset is proposed to be a reference for future studies and must be of high quality. Therefore my comments mainly focus on the analytical procedures and the quality of data included in the tables. I recognized some issues with the data, which should be solved before the manuscript

and the database are accepted for publication. Discussion based on the data sounds valid. However, I cannot fully evaluate the discussion as I am not familiar with details of tephrastratigraphy in New Zealand.

Samples studied Table 1 – list of tephras should be available as a part of the database in Excel format. This table can be linked to data tables.

SM Table missing/required: list of samples with coordinates, their description, name of tephra, age etc.

EMPA data Table 2. Analytical crystals should be indicated. Compositions of reference materials should also be included.

Line 228: "... and Cl".

Line 230: Source of reference data? I see in Table SM2 that reference data for VG-568 differ from Jarosevich et al. 1980. It is said in line 235 that the reference data came from GeoRem. However, Georem does not provide any recommended (preferred) values for this reference glass (and for VGA99 either). So, how was this derived from the GeoRem data and why do the authors believe that the data is better than originally provided by the Smithsonian Institution? Anyway, the data for VG-A99 and VG-568 used for calibration and correction for instrumental drift must be clearly presented in table and justified if they differ from previously recommended values.

Line 238: Some analyses have H2OD as high as 32% (line 566. This analysis has also 1.2% Cl = a lot of epoxy trapped). This is not acceptable for reference data. The authors should exclude all data with calculated H2O>8 wt% (more secondary water can hardly enter glass structure).

Line 241: I recognized some problems with ATHO-G data, which must have influenced the calculated precision.

SM Table 2: Analysis date should be indicated here and for standards.

SM Table 3: poor reproducibility of ATHO-G. The data look very bad for Na. Drift is seen for Si, Ca, Al, Mn. What happened between points 51 and 100? Did the analytical conditions change? Was this different chip of ATHO-G?

No data for VG-A99 provided.

Was Cl calibrated on rhyolite glass VG-568? This cannot be precise calibration as Cl content is 0.1 wt% in the glass.

LA-ICP-MS SM Table 1: Very high level of oxides (ThO/Th=1.3-1.8%, whereas acceptable ratio is 0.5-0.7%). This must have effect on some mass number measured, which have interference with oxides. For example, Sc45 signal was likely strongly affected by interference with Si29O16. This maybe the reason of large deviation of Sc from reference data in STHS standard (SM Table 4). As Sc was calibrated on ATHO with large input from SiO+, Sc measured on samples with lower SiO2 (as STHS) is underestimated.

SM Table 2: For some elements two isotopes were measured. The authors should report just one value: the best or average value for these elements.

Check cells [AB24] and [AC24] – misprints?

Poor agreement between EMP and LA-ICP-MS data for Ti, for example, in lines 43, 45, 46 (contamination by orthopyroxene?), line 67 (contamination by Ti-magnetite) and in many other lines. About 5% of all data have this problem (See plot attached). This indicates contamination of LA-ICP-MS analyses by mineral phases. Outliers must be excluded based on some quantitative criteria. For example, data included in the dataset have better than 10% (or 15%, or 20% - this should be authors choice to place the level of their accuracy) agreement between EMP and LA-ICP-MS for Si and Ti, which were the only elements precisely analysed by both technique. EMP data for Mn are probably not precise enough for such comparison and data screening. Thus, the authors should manually check every LA-ICP-MS analysis and exclude outliers, which cannot

be reliably proved as representing natural melt variability.

In the future, I strongly recommend to analyse all major elements by LA-ICP-MS (Agilent7900 is excellent for this job) as described, for example, by Portnyagin et al. 2020. This will ensure reliable identification of contaminated analyses.

Fig. 2 Very strange data for Rb and Zr and calibration based on NIST-610 in runs # 8 and 9. The outliers do not look representative. NIST610 and NIST612 normally show very consistent calibration, but not in this study. This is unclear. In general, there are several comprehensive investigations of the matrix effect in LA-ICP-MS analysis of glasses. It is usually of the order of 10-20% or less, rarely larger as for Zn, which is volatile (low condensation temperature) and terribly difficult element for LA-ICP-MS analysis. Of course, NIST glasses differ compositionally very much from natural rhyolites. ATHO-G (Askjy rhyolite) should be better standard for natuarl samples.

Line 347: 2sd value depends on analytical conditions and on concentration. Thus, comparison with the secondary standard maybe misleading if the tephras have different composition. ATHO has 3.5%FeO, whereas the glasses from this work have on average 1.4% FeO. Using 2sd from ATHO is too conservative approach to define homogeneous populations in this case. CaO is also typically lower in the glasses compared to ATHO-G.

Fig. 3 -7: all should be updated after cleaning the dataset.

Line 399 and Figure 8: I strongly recommend to NOT use chondrite composition for multi-element plots like this one (Chondrite is Ok for REE plots). This is because chondrites are strongly enriched in some volatile elements, which were lost from the Earth mantle during accretion of our planet or shortly after it (e.g. Pb, Cs, Mo). Better use primitive mantle composition (e.g. McDonough and Sun, 1995, Chem Geology). In this case, normalized Pb, Mo, Cs will be much higher (Mo) and even strongly elevated (Pb) compared to neighboring elements (i.e. LREE for Pb) as it must be in typical subduction related magma. Chondrite normalized spectra like Fig. 8 look very confusing

for geochemists working with trace elements in magmas from different tectonic settings. This figure also should be updated after cleaning the dataset (removing outliers contaminated by solid phases during analysis).

Line 403: Not Pr anomaly, but negative Pb anomaly relative to LREE. Not negative Sm anomaly, but positive Zr-Hf anomaly relative to Sm.

Line 410: Er and Lu peaks are likely analytical artefacts. We do not know high-T processes fractionating these elements from the neighboring HREE.

Line 445 and below: What is the advantage of using PCA analysis in comparison with simple "old-fashion" bi-plots of major elements and spidergrams for trace elements? Do we really need so many elements to distinguish different tephras?

Fig. 15: Some of these plots make little sense to me. For example, Tb and Er are nearly elements-twins, their behaviour is always coupled.
* * *
[Figure]

This figure illustrates poor agreement between Ti content in glass obtained by EMP and LA-ICP-MS in some analyses included in the dataset. Acceptable deviation is about 20% relative or less. Larger discrepancy indicates contamination by mineral phases, which are present in glass and were likely ablated during analysis. These bad analyses should be excluded from the dataset.

**Fig. 1.** Ti EMPA versus Ti La-ICP-MS

---

## Referee Comment (RC2) · Anonymous Referee #2 · 23 Dec 2020

Review of gchron-2020-34

This manuscript sets out to be a database for the geochemical composition of glass in Quaternary rhyolite tephra layers in New Zealand. It provides an extensive data suite of major, minor and trace elements that is proposed to be a reference standard for identifying unknown layers (tephra correlation). The authors rightly point-out that the NZ tephra are well-studied, but there is no standard database, as previous studies are based on differing materials, techniques and labs, and the raw data is not always available. From this point of view the goal of work is valuable. But the question is

whether it has been executed correctly here considering the data suite and its quality (see below).

In addition, it is not clear whether a catalogue or inventory of data makes an actual scientific publication. Indeed, some the high quality tephra databases from elsewhere that the authors quote (line 95-100) are not actually published in refereed journals. The authors do not make a case for new discovery or methods – normally a requirement for science journals. Many (not all) of the conclusions about the homogeneity or heterogeneity of individual tephra and differences between volcanic centres are already thoroughly published by original workers (and re-cycled by previous reviewers – including reviews of reviews). In addition, it is unlikely that many of these local NZ tephra layers would be found beyond the NZ region, and hence the work is of regional rather than global interest. It should be noted the same applies to the other tephra databases.

Regardless, if the database is to become a standard, it must be comprehensive (even if unpublished but publicly available). This requires (1) relatively complete sampling of the known tephra both (i) regionally and (ii) sub-sampling within deposits; and (2) very robust geochemical results/methods.

1 (i) Geographic/stratigraphic coverage. There are some regional gaps in their sampling or sample choice. There is literature on well dated and documented tephra in SE North Island (e.g., Wairarapa and Cape Kidnappers) and northern NI (early Quaternary Auckland region). These have not been analysed and this needs to be explained. They cannot be dismissed as limited localities known or poor age control, because some of tephra layers from the Wanganui basin examined here have very limited known exposure and limited age accuracy.

1 (ii) Eruption deposit character. Although acknowledged as a limitation (near line 185), the work has not attempted to sub-sample within thick deposits despite numerous petrological studies showing time-sequential and azimuth differences in mineralogy and glass composition for some OVC and Taupo tephra layers. This becomes more
evident in proximal outcrops. Thus, creating a dilemma as to what is actual reference material for characterisation. The authors simply dismiss the problem by saying that distal deposits are the largest volume of the eruption (line 580). This is not always correct, and no published examples are offered in support – the old Walker citation provides no documented examples. It is possible that some of the widely dispersed glass from an eruption is homogeneous, but this is not always the case. There are published NZ examples of heterogeneity. Regardless, it becomes a circular argument unless it is documented that there is no spatiotemporal variation. This is difficult to solve. But the authors should expand the discussion to highlight the problem (rather than doing the reverse). As a result, the work implies some of the tephra are chemically very similar (e.g. near line 655). But there are petrology studies showing them to differ. This becomes an issue when you are attempting to make a reference database.

(2) Issues with the robustness of the geochemical data (a) Table 2 shows channels for elements analysed on the probe. This is meaningless unless it is spelt-out in the table which spectrometer crystals (e.g. TAP etc) were used. Its not obviously why channels are highlighted because it is glass standards that actually show achieved accuracy and precision. Spectrometer crystal set-ups will always vary between labs. A table of resulting standards would be better.

(b) (near line 235) It is stated that water in the glass is magmatic. This is unlikely. Most studies show modern pyroclastic glass as relatively anhydrous, and the high and variable water contents (by difference) like in this study is due to variable meteoric hydration. Actually, that in itself is a problem not discussed here.

(c) Water by difference in Supplementary files. A quick scan of the file reveals individual shards with water contents of in the range 10-32 wt %. This is impossible and reflects poor analytical data that should be removed. Most of the listed water data is in the range 0-7 wt %. But it is not clear why even that variation is so wide (within samples). Whether the water was magmatic (unlikely) or meteoric, why would some shards occurring side-by-side in a deposit have widely differing contents e.g. 0.7 and 7.5 wt

% water in sample Kaharoa P? This shows-up the problem with not directly measuring water (which is difficult). The water "by difference" is simply the sum of the analytical error. Unfortunately, this data suite highlights the problem. This needs to be discussed.

(d) The trace element variation is VERY wide in some samples. For example, Ti ∼300-1775 ppm and Sr 62-148 ppm in Taupo PY (also see wide Rb range in Waimihia and range Sr range in Whakatane). Is it likely that the melt is that highly heterogeneous on a micro-scale and is maintained during magmatic transport without mingling. Regardless, some of the co-variation elemental is not consistent with AFC processes. Petrologic studies at Taupo and Okataina volcanoes do not support such micro-variations. If fact many (not all) pumices are relatively homogeneous for the young Taupo eruption and other deposits examined here (OVC etc). The variability presented here could partly reflect the nature of the sample analysed. Were individual pumice or lapilli used? If not, then the matrix of many of these deposits contains xenolithic material including obsidians, dome glasses and other volcanogenic detritus. It is likely such fragments have been probed here, but they would not be representative of widely dispersed vitric ash found in deep-sea cores and elsewhere (as already published). Hence, a reference set of data must be based on juvenile lapilli where available. Lapilli is available for most of the post-20 ka deposits. Perhaps an appendix of lithology information is needed. Contributing to the problem is that some of the data are analytical outliers and reflect analytical problems. For example, data line 67 in Waimihia has V = 45 ppm and Cu = 79 ppm – wildly different to the other analyses and does not reflect petrologic processes. The entire dataset would need to be filtered for these types of errors. But this also raises the issue of how to filter it. It would need to be fully explained. I suspect the cause of the problem is ablation of microlites and micro-voids in the glass. This is a major problem in previous tephra studies using laser ablation and should be discussed in the text if you argue this is a reference database.

(e) There are some poor comparisons between element contents determined by EMA versus laser ICPMS (e.g. Ti). This needs to be discussed or explained. Data may need

to be filtered. But that requires a robust approach that is clearly explained.

In summary, (1) the geographic/stratigraphic sampling is not uniform (2) spatiotemporal within-deposit heterogeneity is not explored but is highlighted by other workers (3) the geochemical database has variable errors and outliers, and has not been filtered (4) filtering of the analytical database would require a robust rationale to avoid bias (5) suitable geologic material (individual lapilli) are needed to avoid xenolithic sources when proposing a reference standard

---

## Referee Comment (RC3) · Stephen Kuehn (Referee) · 8 Jan 2021

The core purpose of this paper is to present and evaluate a set of New Zealand tephra reference data to provide an openly-available framework and database to support tephra-using multidisciplinary research. To facilitate unambiguous eruption identification, samples from key reference locations have been characterized using a consistent, well-described methodology. This dataset is then examined to outline where tephra identifications may still be challenging, even with such consistent data.

[Figure]

The paper follows many recommended best practices in tephra data reporting (e.g. including detailed methods, reporting H2OD, reporting results on quality control standards), but it could go further. With additional documentation, better data filtering, and better exploration of deposit heterogeneity it could become outstanding example.

The authors also discuss the importance of data repositories, yet this dataset appears to exist only as Excel tables currently. Are there specific plans to house the TephraNZ dataset in an open database system that supports data search functions (e.g. Petlab)? The authors discuss the potential for this to be a living data set that expands over time. What plans are there for long-term curation of such an expanding reference dataset? Additionally, interoperability across and linking between data systems is a goal that has been expressed by multiple scientific communities. Careful documentation of data provenance that is recorded in a consistent fashion is one part of achieving this. Unique identifiers like IGSNs for samples and DOIs for papers are another. In volcanology, globally unique IDs exist for volcanic systems and are maintained by the Smithsonian GVP, and these could be incorporated into the dataset. For the Holocene, volcanic eruption and episode IDs also exist.

Analytical and sample processing methodologies are relatively well-described, but the paper could go further. Be sure to include the detection limits for all elements and how these were determined. Additionally, the supplemental spreadsheet file contains a table outlining the LA-ICP-MS method. Please add a similar table for EPMA. Even better, review Abbott et al. (2020, https://zenodo.org/record/4075613) "Community Established Best Practice Recommendations for Tephra Studies-from Collection through Analysis" for the documents on methodology reporting. Version 1 of these recommendations was released in May, 2020. Version 2 and some worked examples (e.g. https://doi.org/10.5281/zenodo.4074289 with EPMA methodology detailed) were released in early October. For EPMA, this includes details such as spectrometer assignments, diffracting crystals, and sequence. As there is not yet a worked example for LA-ICP-MS, the Tephra NZ project could supply the first one for this international

tephra community effort. Such a stand-alone methods document with DOI could then also be re-usable and could be linked with all papers and datasets that use that same methodology now and in the future.

The paper states that a robust data set with geochemical consistency resulting from consistent methodology is necessary for the optimal use of statistical techniques to correlate tephras. This is because these methods generally do not account for variations resulting from changes in analytical methods over time within a single lab or from biases between different laboratories. Although perhaps ideal, this is not the only approach. Routine use of common reference materials across laboratories can be used to remove most laboratory bias and harmonize datasets, provided that alkali element migration effects (e.g. sodium loss) are sufficiently limited. This is a key reason why the International Focus Group on Tephrochronology and Volcanism (INTAV, recently renamed Commission on Tephrochronology or COT) distributed a set of four reference glasses to more than 25 laboratories for analysis and why a similar effort is currently underway with additional glasses for LA-ICP-MS.

When using the reference glass ATHO-G, it is important to be aware that the published preferred $Na_2O$ concentration of 3.75 wt% (Jochum et al., 2006, Table 13h) is too low. Several analyses with significant sodium loss were not screened out prior to calculating the preferred composition. A majority of newer analyses in the GeoReM database exceed the above reference concentration as do XRF and INAA analyses which are immune to alkali element migration. Kimura et al (2018) also report a concentration of 4.6 by FS-LA-ICP-MS. Together, these data suggest that a $Na_2O$ concentration of 4.3 to 4.6 wt% for ATHO-G is probably more accurate. Using the 3.75 concentration during EPMA standardization or offline correction will bias all results to too low values. Using the same 3.75 concentration when assessing whether Na-loss is sufficiently controlled by an analytical method may also produce an inaccurate interpretation.

Careful removal (but not deletion) of outliers is important before computing summary statistics. The data tables need to be further screened for these. For example, SM

Table 2 contains obviously problematic trace element analyses like line Kaharoa-P_22 which has elevated Zr. Such points should be set aside and labeled as outliers within the dataset, following recommended practice.

Analyses with very low totals should be considered as outliers, labeled, and set aside. Often, a 90% total (10% water and unanalyzed elements) is used as the threshold for this, but that is not appropriate in all cases. Some maximally hydrated tephras yield mean totals of 90-91 wt%, and require a somewhat lower cutoff to allow for the range of analytical precision. If a Cl-bearing epoxy is used, elevated Cl concentrations on points that also have low totals indicate that the beam spot is not entirely on tephra glass. Often, these points will compare well to others when normalized to 100% totals, but the Cl will be inaccurate.

I am also concerned about the removal of individual data values that fall below the single analysis detection limits, despite this being commonly applied. It is much better to include the actual data values and also report what the detection limits are. This way, information is not discarded, and each end user can determine what is fit for their own purpose. Fundamentally, analytical techniques like EPMA and ICP-MS are based on signal counting and essentially follow Poisson statistics. Consequently, replicate analyses on a homogeneous material follow a probability distribution. At low enough concentrations, a portion of that distribution will begin to fall below the single analysis detection limit at some selected level of confidence (often chosen as 99% or 3 sigma, but 2 sigma and 1 sigma are valid confidence levels as well). At this point, those specific analyses that fall below the detection limit are no less valid as members of the population than those analyses which fall above the limit. It is therefore completely inappropriate to compute summary statistics like mean, median, mode, or standard deviation when the low end of that population has been removed. Removing those below detection limit values and then computing the mean and standard deviation of those that remain biases the mean to too high a value and reduces the standard deviation, misrepresenting the true precision.
Another community-recommended best practice is to have all tephra analyses and glass standard analyses linked on an analytical session (run) basis. This is because results on common reference glasses are perhaps the best demonstration of analytical quality and because instrument performance can vary from session to session. In SM Table 4, trace element standard analyses are clearly labeled with the individual runs. Please add equivalent information to the other tables for both standards and samples.

Where multiple isotopes are measured for the same element, could these be used in combination rather that reporting concentrations from each with assumed isotopic ratios? (Or alternately use the one less prone to interferences, or an average.)

SM Table 2 puts some EPMA and LA-ICP-MS analyses together on the same line. Presumably this means that both were done on the same shards. If so, state this explicitly.

SM Table 3 Major Standards omits results on VG-A99 despite the paper text indicating that this was run as a secondary (and primary) standard. Please add the data for VG-A99 to the table. This same table lists some individual analyses with surprisingly low Na2O concentrations for ATHO-G. What is going on with these? Could these represent cases where the electron beam analyzed the same location twice? These problematic results should be set aside and labeled as such but not entirely removed from the table.

SM Table 3 also includes some plots which apparently show results on reference glasses over time. Please add explanation for these. Also, the plots for ATHO-G show some discontinuities and trends that suggest problems with the standardization and/or drift. These results on ATHO-G are worrying. Please explain what is going on here. Do analyses of VG-A99 show the same patterns? The text mentions that offline data reduction was used to correct for variability in the VG-568 primary calibration. Are the ATHO-G results in this table the raw or corrected values?

SM Table 4 – Offset is a percentage. Label it as such for clarity. Relabel 2*Std as 2*StDev for clarity. It may be more useful to compare the offset from reference in ppm

to the StDev in ppm as a ratio rather than reporting an offset in % of the reference ppm (or do both). This way, if a result is e.g. 3 standard deviations from reference, this would be more readily apparent.

Line 192 – The proximal-distal differences mentioned raise the question of how it would be known that these are the same tephra?

Lines 224- – What was the kV? What is total time for an entire analysis, including spectrometer movement and other instrument overhead?

Lines 232-34 – The text mentions monitoring for drift. Was there drift? How was the drift correction implemented – linear interpolation?

Line 235 – By "applied to all the data" does this mean just all data lines, or all lines and all elements? Where an element is at low concentration on the reference glass (e.g. MnO, MgO in VG-568), the precision may not be sufficient to apply a reliable correction to other analyses. Additionally, such a correction will make little difference where VG-568 is both the primary calibration standard and the offline correction standard for the same element.

Line 237-238 – For older tephra glasses, most of this water can be secondary hydration (water absorbed from the environment post deposition) rather than magmatic. This is acknowledged later in equation 3.

Lines 245-255 – Equations 1 and 2 essentially outline a standards-based (reference material based) normalization using a single standard. This could also be done using a consensus of the three EPMA reference glasses (VG-568, VG-A99, and ATHO-G) for even better consistency. Of course, any bad analyses would have to be set aside prior to computing consensus corrections.

Line 315 – DFA with cross-validation is another way to test this by looking for high rates of misclassification.

Line 321 – "of" instead of "or"

Lines 347-49 – This is perhaps OK as a rough criterion, but precision also scales with concentration due to signal intensity and counting statistics. At lower concentrations, standard deviations will be smaller in wt%, but the relative standard deviation (as a percentage of the analyzed concentration) will be greater. The reverse is true when going to higher concentrations such that a 4x increase in signal intensity corresponds to a relative standard deviation that is reduced by a factor of 1/2 (i.e. scaling with sqrt of total x-rays recorded). So, the standard deviation on a secondary standard only provides an estimate of what should be expected for a homogeneous sample when the concentrations of the same element are similar. Consequently the stated +/- 0.23wt% homogeneity cutoff for Fe would not be appropriate for a sample with a much higher or lower concentration of Fe than the standard.

Line 364 – Crystallization of biotite is not the only way to affect $K_2O$ concentrations. Does the Shane et al (2008) reference provide evidence for biotite fractionation?

Figure 4 – The $TiO_2$ and MgO appear "quantized" on the plots, probably due to rounding everything to the nearest 0.01 wt% prior to plotting. To avoid artifacts like this, it is often better to carry one extra decimal place.

Line 415 – Perhaps replace "can be used to maintain" with "may exhibit"

Line 453 – Not coincidentally, some of John Westgate's early work in the 1960 and 1970s (e.g. Westgate & Evans 1978) also used the same three elements.

Line 454 – Perhaps replace "presence" with "inclusion"

Figure 10 – Was Ba not included in the histograms for some reason? Some of the text and later plots point to Ba as a useful discriminator.

Lines 504-505 – Remove "ratios"

Line 606 – Perhaps replace "acts to effectively reduce the variability of" with "exhibits little variability in"

Line 607 – Perhaps replace "can be used to maintain" with "exemplifies"

Line 610 – Replace "causes the" with "exhibits"

Figure 13 (and supplemental tables) – Where multiple populations are clearly evident as in this figure, it would be useful to clearly identify such populations in the data tables, e.g. as Pop 1, Pop 2. Means and standard deviations should then be computed and reported separately for each major population. Data repositories also need to support archival of such details.

Lines 739-757 – Yes, discovering more tephra layers in marine and lacustrine sections than are known from proximal deposits at potential source volcanoes appears to be common in many volcanic regions. (Multiple examples have been reported for e.g. the Cascades arc too such as at Summer Lake, Oregon.) Therefore, developing more complete records of volcanic events (and also understanding their spatial distribution, timing, and eruptive/dispersal processes) requires the integration of proximal and distal tephra records, ideally into accessible databases.

Tables 1-5 – To facilitate data re-use, provide in the supplemental file spreadsheet versions of all of these tables in addition to the versions embedded within the manuscript.

Table 1 – Relabel easting and northing as longitude and latitude. Also convert to decimal degrees for simpler presentation and easier reuse. Abbott et al. (2020) noted above also contains recommendations regarding reporting of sample details.

Table 2 – This table lists acquisition time, but it does not specify which is peak and which is background or whether one or two backgrounds were measured.

---

## Author Comment (AC1) · 3 Mar 2021

We thank Dr Portnyagin for taking the time to give such a thorough review of our research, and appreciate his positivity toward the results. Dr Portnyagin highlights that as we are proposing this research as a reference set, the data should be of high quality. We wholeheartedly agree and we therefore very much appreciate his detailed comments on the analytical procedures and quality of the data. Below we have addressed the key themes of this review.

[Figure]

Samples studied Table 1 – list of tephras should be available as a part of the database in Excel format. This table can be linked to data tables. SM Table missing/required: list of samples with coordinates, their description, name of tephra, age etc. EMPA data Table 2. Analytical crystals should be indicated. Compositions of reference materials should also be included. Table 1 with all the sample names and co-ordinate locations will be available as an excel spreadsheet, downloadable with the SM material. All the tables will be accessible in this format to ensure they are useful, and usable for future studies. The transparency of the glass-based data underpins the presentation of this research. We will update Table 2 to include the analytical crystals used and compositions of reference materials. The analysis dates for the standards are listed in SM Table 3 (EPMA) and SM Table4 (LA-ICP-MS); we will add analysis dates for the samples to Table 3 for clarity.

Line 228: ". . . and Cl". Amended

Line 230: Source of reference data? I see in Table SM2 that reference data for VG-568 differ from Jarosevich et al. 1980. It is said in line 235 that the reference data came from GeoRem. However, Georem does not provide any recommended (preferred) values for this reference glass (and for VGA99 either). So, how was this derived from the GeoRem data and why do the authors believe that the data is better than originally provided by the Smithsonian Institution? Anyway, the data for VG-A99 and VG-568 used for calibration and correction for instrumental drift must be clearly presented in table and justified if they differ from previously recommended values. The reference data values for VG-568 and VG-A99 come from GeoREM. GeoREM reported 53 published analyses between 2006 and 2019 for VG-568, and 21 published analyses for VG-A99 between 2006 and 2015. We believe these to be more up to date than those provided by Jarosevich et al. (1980) but all are within error of Jarosevich et al's data, and are publicly accessible, unlike the Jarosevich et al. (1980) publication. We have produced averaged data for the GeoREM values reported. However, because these values are updated frequently we chose to use the values reported by Streck et al. (2006) for

consistency and continuity. In the database, Streck et al., are the only group to publish values for both VG-568 and VG-A99 for EPMA analysis. These values will be reported and appropriately referenced in the SM Table 3. Standard data for VG-A99 in SM Table 3 will also be provided as requested by Dr Portnyagin.

Line 238: Some analyses have H2OD as high as 32% (line 566. This analysis has also 1.2% Cl = a lot of epoxy trapped). This is not acceptable for reference data. The authors should exclude all data with calculated H2O>8 wt% (more secondary water can hardly enter glass structure). Yes, we have identified 19 data points throughout the 1190 that have H2OD values $\geq$ 8 wt%. These will indeed be removed from our dataset. This is an oversight on our part. There is only 1 sample with Cl value $\geq$ 0.72 wt%, and this is also one of the samples with high H2OD that will be removed.

Line 241: I recognized some problems with ATHO-G data, which must have influenced the calculated precision. We thank Dr Portnyagin for pointing this out.

SM Table 2: Analysis date should be indicated here and for standards. Amended

SM Table 3: poor reproducibility of ATHO-G. The data look very bad for Na. Drift is seen for Si, Ca, Al, Mn. What happened between points 51 and 100? Did the analytical conditions change? Was this different chip of ATHO-G? No data for VG-A99 provided. Was Cl calibrated on rhyolite glass VG-568? This cannot be precise calibration as Cl content is 0.1 wt% in the glass. SM Table 3 shows the standard data for VG-568 (run as a primary standard) and ATHOG (run as the secondary standard). There is a jump in the data at point #51 and some drift observed after this point, as the machine was re-standardised after being left idle for two days (13/12–16/12). Besides this, no analytical conditions changed. The average and standard deviation for these data between these two periods are indistinguishable except for SiO2, which show a variance in the average recorded value of 0.7 wt% . We will re-standardise using these separate analysis sessions to ensure the values reported are as accurate as they can be.

[Figure]

LA-ICP-MS SM Table 1: Very high level of oxides (ThO/Th=1.3-1.8%, whereas acceptable ratio is 0.5-0.7%). This must have effect on some mass number measured, which have interference with oxides. For example, Sc45 signal was likely strongly affected by interference with Si29O16. This maybe the reason of large deviation of Sc from reference data in STHS standard (SM Table 4). As Sc was calibrated on ATHO with large input from SiO+, Sc measured on samples with lower SiO2 (as STHS) is underestimated. The oxide levels reported for our analyses (SM Table 1) are ThO/Th = 1.3–1.8 %. These values are comparable to those reported by other publications at similar operating conditions. For example, Pearce et al. (2011) reported ThO/Th values of "typically ∼1.5" using a 193 nm laser ablation system with 5 Hz repetition rate and a 20s acquisition time. We note that these are higher than those reported by Portnyagin et al. (2019). However, we also report both 153Eu and 151Eu values (which can be affected by 137BaO). The concentrations measured for these isotopes are very similar and show no relationship with Ba, thus we assume no problem with oxide interference on either the standard or sample data.

SM Table 2: For some elements two isotopes were measured. The authors should report just one value: the best or average value for these elements. We will report the most common isotope for these elements.

Check cells [AB24] and [AC24] – misprints? Poor agreement between EMP and LA-ICP-MS data for Ti, for example, in lines 43, 45, 46 (contamination by orthopyroxene?), line 67 (contamination by Ti-magnetite) and in many other lines. About 5% of all data have this problem (See plot attached). This indicates contamination of LA-ICP-MS analyses by mineral phases. Outliers must be excluded based on some quantitative criteria. For example, data included in the dataset have better than 10% (or 15%, or 20% - this should be authors choice to place the level of their accuracy) agreement between EMP and LA-ICP-MS for Si and Ti, which were the only elements precisely analysed by both technique. EMP data for Mn are probably not precise enough for such comparison and data screening. Thus, the authors should manually check every LA-

ICP-MS analysis and exclude outliers, which cannot be reliably proved as representing natural melt variability. We realise that it is a common theme throughout our reviews: the data have not been thoroughly reduced. This is an oversight on our part, however, in some cases we did not remove outliers as we were unsure if these were indeed anomalous or if they were simple natural variability in the data. There is evidence for fractional crystallisation of mineral phases within the data, and we wanted the reference data to reflect (where appropriate) this natural variability within the samples. In some cases, this geochemical variability is the unique identifier for the tephra. We will add in a Supplementary file that shows a comparison between the elements that were analysed on both EPMA and LA-ICP-MS for clarity.

In the future, I strongly recommend to analyse all major elements by LA-ICP-MS (Agilent7900 is excellent for this job) as described, for example, by Portnyagin et al. 2020. This will ensure reliable identification of contaminated analyses. We thank Dr Portnyagin for this advice and will endeavour to do this in subsequent analyses.

Fig. 2 Very strange data for Rb and Zr and calibration based on NIST-610 in runs # 8 and 9. The outliers do not look representative. NIST610 and NIST612 normally show very consistent calibration, but not in this study. This is unclear. In general, there are several comprehensive investigations of the matrix effect in LA-ICP-MS analysis of glasses. It is usually of the order of 10-20% or less, rarely larger as for Zn, which is volatile (low condensation temperature) and terribly difficult element for LA-ICP-MS analysis. Of course, NIST glasses differ compositionally very much from natural rhyolites. ATHO-G (Askjy rhyolite) should be better standard for natuarl samples. This figure outlines our investigation into the most appropriate standard to use for our analyses, and we agree that our data show ATHO-G to be the most appropriate.

Line 347: 2sd value depends on analytical conditions and on concentration. Thus, comparison with the secondary standard maybe misleading if the tephras have different composition. ATHO has 3.5%FeO, whereas the glasses from this work have on average 1.4% FeO. Using 2sd from ATHO is too conservative approach to define homogeneous populations in this case. CaO is also typically lower in the glasses compared to ATHO-G. We thank Dr Portnyagin for pointing this out.

Fig. 3 -7: all should be updated after cleaning the dataset. Amended

Line 399 and Figure 8: I strongly recommend to NOT use chondrite composition for multi-element plots like this one (Chondrite is Ok for REE plots). This is because chondrites are strongly enriched in some volatile elements, which were lost from the Earth mantle during accretion of our planet or shortly after it (e.g. Pb, Cs, Mo). Better use primitive mantle composition (e.g. McDonough and Sun, 1995, Chem Geology). In this case, normalized Pb, Mo, Cs will be much higher (Mo) and even strongly elevated (Pb) compared to neighboring elements (i.e. LREE for Pb) as it must be in typical subduction related magma. Chondrite normalized spectra like Fig. 8 look very confusing for geochemists working with trace elements in magmas from different tectonic settings. This figure also should be updated after cleaning the dataset (removing outliers contaminated by solid phases during analysis). Amended

Line 403: Not Pr anomaly, but negative Pb anomaly relative to LREE. Not negative Sm anomaly, but positive Zr-Hf anomaly relative to Sm. Amended

Line 410: Er and Lu peaks are likely analytical artefacts. We do not know high-T processes fractionating these elements from the neighboring HREE. Noted

Line 445 and below: What is the advantage of using PCA analysis in comparison with simple "old-fashion" bi-plots of major elements and spidergrams for trace elements? Do we really need so many elements to distinguish different tephras? We have found in our research on New Zealand tephras that more elements from the glass analyses are often required to distinguish between tephras from different eruptions. This feature is discussed in the text. For this reason, we have chosen to run PCA analysis to further distinguish between samples that simple biplots cannot. This approach uses the theory of "handprinting" rather than "fingerprinting", and it is often very successful, especially when statistical methods are used. In this way data can be looked at in multidimensional space, and thus variations in the data can be more readily distinguished and correlations made more robustly.

Fig. 15: Some of these plots make little sense to me. For example, Tb and Er are nearly elements-twins, their behaviour is always coupled. These elemental couples, and all the plots, were chosen because they show the best distinction between the two tephras.

Please also note the supplement to this comment:
https://gchron.copernicus.org/preprints/gchron-2020-34/gchron-2020-34-AC1-supplement.pdf

---

## Author Comment (AC2) · 3 Mar 2021

This manuscript sets out to be a database for the geochemical composition of glass in Quaternary rhyolite tephra layers in New Zealand. It provides an extensive data suite of major, minor and trace elements that is proposed to be a reference standard for identifying unknown layers (tephra correlation). The authors rightly point-out that the NZ tephra are well-studied, but there is no standard database, as previous studies are based on differing materials, techniques and labs, and the raw data is not always available. From this point of view the goal of work is valuable. We thank Reviewer2 for

taking the time to read and comment on our manuscript. Below we address some of the key concerns highlighted by R2 in their comments.

But the question is whether it has been executed correctly here considering the data suite and its quality (see below). In addition, it is not clear whether a catalogue or inventory of data makes an actual scientific publication. Indeed, some the high quality tephra databases from elsewhere that the authors quote (line 95-100) are not actually published in refereed journals. The authors do not make a case for new discovery or methods – normally a requirement for science journals. Many (not all) of the conclusions about the homogeneity or heterogeneity of individual tephra and differences between volcanic centres are already thoroughly published by original workers (and re-cycled by previous reviewers – including reviews of reviews). In addition, it is unlikely that many of these local NZ tephra layers would be found beyond the NZ region, and hence the work is of regional rather than global interest. It should be noted the same applies to the other tephra databases. Despite agreeing that this manuscript provides an "extensive data suite" and that we "rightly point-out that there. . ..is no standard database" for New Zealand tephra, we are surprised that R2 thinks this research is not valuable enough for a scientific publication. We (and the other two invited reviewers) strongly disagree that this research has no new "discovery or method" as described by R2. This manuscript includes many firsts for New Zealand tephrochronology that are clearly worthy of publication including (but not limited to): 1. The first time almost all of these tephra (of known source) have been analysed for glass trace element compositions – allowing more robust correlations where major elements are not distinct enough, as shown by our work and previous publications. 2. The first time a comprehensive and systematically analysed data set has been produced, which allows rigorous assessment of the relationship between the geochemical signatures of glass not only from different tephras but also from different caldera systems and their evolution through time. 3. The first open access, fully holistic dataset that has been produced, including not just mean and standard deviation values, which have limitations when producing robust correlations, but also full standard data information (i.e.

individual shard analyses) and data reduction methods, which are critical for producing comparable data in future projects.

Regardless, if the database is to become a standard, it must be comprehensive (even if unpublished but publicly available). This requires (1) relatively complete sampling of the known tephra both (i) regionally and (ii) sub-sampling within deposits; and (2) very robust geochemical results/methods

(i) Geographic/stratigraphic coverage. There are some regional gaps in their sampling or sample choice. There is literature on well dated and documented tephra in SE North Island (e.g., Wairarapa and Cape Kidnappers) and northern NI (early Quaternary Auckland region). These have not been analysed and this needs to be explained. They cannot be dismissed as limited localities known or poor age control, because some of tephra layers from the Wanganui basin examined here have very limited known exposure and limited age accuracy. Reviewer 2 rightly points out that there is extensive research on tephra in many regions of New Zealand. However, we believe R2 misses one of the main themes of this reference dataset, namely providing reference data from known and well-documented tephra deposits. Even in the abstract we plainly state that "we target original type sites or reference locations where the tephra's identification is unequivocally known based on independent dating or mineralogical techniques". Therefore, many of the sites highlighted by R2 (e.g. Auckland, Wairarapa, Hawke Bay), and indeed other sites in New Zealand, were deemed inappropriate for this study as their assigned eruptives are commonly just correlations based on other research. By adding such sites could potentially perpetuate misinformation and miscorrelation, which is why we have chosen to pointedly target the type sites for these tephras, or where their correlation is unequivocally known through multiple criteria (e.g. Kaipo bog sites).

(ii) Eruption deposit character. Although acknowledged as a limitation (near line 185), the work has not attempted to sub-sample within thick deposits despite numerous petrological studies showing time-sequential and azimuth differences in mineralogy
and glass composition for some OVC and Taupo tephra layers. This becomes more paper evident in proximal outcrops. Thus, creating a dilemma as to what is actual reference material for characterisation. The authors simply dismiss the problem by saying that distal deposits are the largest volume of the eruption (line 580). This is not always correct, and no published examples are offered in support – the old Walker citation provides no documented examples. It is possible that some of the widely dispersed glass from an eruption is homogeneous, but this is not always the case. There are published NZ examples of heterogeneity. Regardless, it becomes a circular argument unless it is documented that there is no spatiotemporal variation. This is difficult to solve. But the authors should expand the discussion to highlight the problem (rather than doing the reverse). As a result, the work implies some of the tephra are chemically very similar (e.g. near line 655). But there are petrology studies showing them to differ. This becomes an issue when you are attempting to make a reference database. We recognise that past literature (which we have cited) has identified variability in composition for the same eruption's deposits, linked dominantly to changing wind direction during the eruption coupled with magma source dynamics and compositional variations, and this limitation is discussed in Section 4.1.3. Because of this we sampled both proximal and distal locations (where possible) to emphasise that there is sometimes variability in the geochemical compositions of these deposits from a single eruptive episode, and our results show this is true for some, but not all, of our samples, which is discussed in the text. As R2 suggests, we will expand our discussion on this to give more clarity to the reader with regards to the complexity of geochemical variations throughout an eruption.

We would like to reiterate that this foundation dataset, is proposed to be the beginning of what we hope will be a constantly updating and evolving tephra data base for New Zealand (in our case, based on glass-shard analyses). Indeed, we have financial support for the next two years for projects to gather more samples and data where appropriate.

(2) Issues with the robustness of the geochemical data (a) Table 2 shows channels for elements analysed on the probe. This is meaningless unless it is spelt-out in the table which spectrometer crystals (e.g. TAP etc) were used. Its not obviously why channels are highlighted because it is glass standards that actually show achieved accuracy and precision. Spectrometer crystal set-ups will always vary between labs. A table of resulting standards would be better. Additional information will be added to this table. We highlight that SM Table 3 shows all the standard data for major elements run on the EPMA.

(b) (near line 235) It is stated that water in the glass is magmatic. This is unlikely. Most studies show modern pyroclastic glass as relatively anhydrous, and the high and variable water contents (by difference) like in this study is due to variable meteoric hydration. Actually, that in itself is a problem not discussed here. Amended in the text

(c) Water by difference in Supplementary files. A quick scan of the file reveals individual shards with water contents of in the range 10-32 wt %. This is impossible and reflects poor analytical data that should be removed. Most of the listed water data is in the range 0-7 wt %. But it is not clear why even that variation is so wide (within samples). Whether the water was magmatic (unlikely) or meteoric, why would some shards occurring side-by-side in a deposit have widely differing contents e.g. 0.7 and 7.5 wt% water in sample Kaharoa P? This shows-up the problem with not directly measuring water (which is difficult). The water "by difference" is simply the sum of the analytical error. Unfortunately, this data suite highlights the problem. This needs to be discussed. Yes, there are 19 data points in the 1190 that have H2OD values $\geq$ 8 wt%. These have been removed from our dataset and additional discussion will be added to the text to explain this.

(d) The trace element variation is VERY wide in some samples. For example, Ti âĹij300- 1775 ppm and Sr 62-148 ppm in Taupo PY (also see wide Rb range in Waimihia and range Sr range in Whakatane). Is it likely that the melt is that highly heterogeneous on a micro-scale and is maintained during magmatic transport without mingling. Regardless, some of the co-variation elemental is not consistent with AFC processes. Petrologic studies at Taupo and Okataina volcanoes do not support such micro-variations. If fact many (not all) pumices are relatively homogeneous for the young Taupo eruption and other deposits examined here (OVC etc). The variability presented here could partly reflect the nature of the sample analysed. Were individual pumice or lapilli used? If not, then the matrix of many of these deposits contains xenolithic material including obsidians, dome glasses and other volcanogenic detritus. It is likely such fragments have been probed here, but they would not be representative of widely dispersed vitric ash found in deep-sea cores and elsewhere (as already published). Hence, a reference set of data must be based on juvenile lapilli where available. Lapilli is available for most of the post-20 ka deposits. Perhaps an appendix of lithology information is needed. Contributing to the problem is that some of the data are analytical outliers and reflect analytical problems. For example, data line 67 in Waimihia has V = 45 ppm and Cu = 79 ppm – wildly different to the other analyses and does not reflect petrologic processes. The entire dataset would need to be filtered for these types of errors. But this also raises the issue of how to filter it. It would need to be fully explained. I suspect the cause of the problem is ablation of microlites and micro-voids in the glass. This is a major problem in previous tephra studies using laser ablation and should be discussed in the text if you argue this is a reference database.

We are aware of the high variability in the trace element data we have produced. As R2 points out, it is well known in the literature the difficulty in analysing glass shards without accidentally including microcrysts. Therefore, we realise we need to be even more rigorous in our data reduction. Additional data reduction will be fully explained in the text, as has been done for data reduction and standardisation already (sections 2.3-2.6).

We highlight to R2, as discussed frequently in the text, that these analyses were run on glass shards in 60–250 $\mu$m size fraction, thus negating contamination by what R2 suggests could include "xenolithic material including obsidian, dome glasses and other

volcanogenic detritus". The whole point of this study is to provide a comparable data set to distal deposits (including from deep sea cores as identified by R2), hence our exclusive use of the same material (juvenile glass) found at these distal sites.

We also note that heterogeneity and homogeneity are often defining features of an eruption and, by analysing a juvenile clast (e.g. lapilli) you are effectively producing a whole rock analysis, which can homogenise the dataset. For example, small scale heterogeneity is less likely to be picked up as the juvenile clast will represent only a small part of the magmatic system. Selecting glass shards from ashfall is far more likely to sample a wider range of the magmatic system, depending on which part of the magma chamber they come from (and what minerals they are in equilibrium with). This is confirmed by melt inclusion work. For example, glass shards that formed interstitially to plagioclase-dominated crystal mush will be very different in composition to those formed when surrounded by biotite, or those from areas of high melt proportion and few crystals. We believe that many previous studies have "over-reduced" their data, removing "outliers" or "inherited shards" and therefore skewing the geochemical composition results. This detail is also lost where means and standard deviations of individual shards are reported rather than the full data set. The point of reporting all of our data (even when the internal variability seems high) is in the hope that these details are retained.

(e) There are some poor comparisons between element contents determined by EMA versus laser ICPMS (e.g. Ti). This needs to be discussed or explained. Data may need to be filtered. But that requires a robust approach that is clearly explained. This data will be plotted and added as an appendix into the Supplementary Files once further data reduction has been undertaken.

In summary, (1) the geographic/stratigraphic sampling is not uniform (2) spatiotemporal within-deposit heterogeneity is not explored but is highlighted by other workers (3) the geochemical database has variable errors and outliers, and has not been filtered (4) filtering of the analytical database would require a robust rationale to avoid bias

(5) suitable geologic material (individual lapilli) are needed to avoid xenolithic sources when proposing a reference standard.

Please also note the supplement to this comment:
https://gchron.copernicus.org/preprints/gchron-2020-34/gchron-2020-34-AC2-supplement.pdf

―――――――――――――――――――

---

## Author Comment (AC3) · 3 Mar 2021

The core purpose of this paper is to present and evaluate a set of New Zealand tephra reference data to provide an openly-available framework and database to support tephra-using multidisciplinary research. To facilitate unambiguous eruption identification, samples from key reference locations have been characterized using a consistent, well-described methodology. This dataset is then examined to outline where tephra identifications may still be challenging, even with such consistent data. The paper follows many recommended best practices in tephra data reporting (e.g. including

detailed methods, reporting H2OD, reporting results on quality control standards), but it could go further. With additional documentation, better data filtering, and better exploration of deposit heterogeneity it could become outstanding example. Our thanks to Assoc. Prof Kuehn for his thoughtful and valuable comments on our manuscript. The insight into "community best practise" will be invaluable in our revised manuscript.

The authors also discuss the importance of data repositories, yet this dataset appears to exist only as Excel tables currently. Are there specific plans to house the TephraNZ dataset in an open database system that supports data search functions (e.g. Petlab)? The authors discuss the potential for this to be a living data set that expands over time. What plans are there for long-term curation of such an expanding reference dataset? Additionally, interoperability across and linking between data systems is a goal that has been expressed by multiple scientific communities. Careful documentation of data provenance that is recorded in a consistent fashion is one part of achieving this. Unique identifiers like IGSNs for samples and DOIs for papers are another. In volcanology, globally unique IDs exist for volcanic systems and are maintained by the Smithsonian GVP, and these could be incorporated into the dataset. For the Holocene, volcanic eruption and episode IDs also exist. In its current "in review" status, the data linked to this manuscript are indeed only available through the supplementary file. However, once this manuscript is published, we have in place an online data repository. In addition, all these data will be available through PetLab. We also have a website set up to disseminate information and highlight when new data are added to the set. At present our plan is to update the PetLab data base as we continue to grow the data, but our idea is that this original foundation glass-shard analyses dataset of known tephra act as the "primary" dataset. These will be highlighted on PetLab, and any additional samples added to the set by ourselves or other studies will be labelled as "secondary" if a correlation is made of an unknown/uncorrelated tephra deposit. Thank you for bringing the Smithsonian GVP number system to our attention. This information has been added into Table 1 for TaupÅ■ = 241070 and Okataina = 241050; no others are listed.

[Figure]

Analytical and sample processing methodologies are relatively well-described, but the paper could go further. Be sure to include the detection limits for all elements and how these were determined. Additionally, the supplemental spreadsheet file contains a table outlining the LA-ICP-MS method. Please add a similar table for EPMA. Even better, review Abbott et al. (2020, https://zenodo.org/record/4075613) "Community Established Best Practice Recommendations for Tephra Studies-from Collection through Analysis" for the documents on methodology reporting. Version 1 of these recommendations was released in May, 2020. Version 2 and some worked examples (e.g. https://doi.org/10.5281/zenodo.4074289 with EPMA methodology detailed) were released in early October. We thank Assoc. Prof Kuehn for bringing these studies to our attention and have included the guidance given in Abbott et al. (2020a&b).

For EPMA, this includes details such as spectrometer assignments, diffracting crystals, and sequence. As there is not yet a worked example for LA-ICP-MS, the Tephra NZ project could supply the first one for this international tephra community effort. Such a stand-alone methods document with DOI could then also be re-usable and could be linked with all papers and datasets that use that same methodology now and in the future. Table 2 will be updated for more information as to the EPMA setup and running protocols.

The paper states that a robust data set with geochemical consistency resulting from consistent methodology is necessary for the optimal use of statistical techniques to correlate tephras. This is because these methods generally do not account for variations resulting from changes in analytical methods over time within a single lab or from biases between different laboratories. Although perhaps ideal, this is not the only approach. Routine use of common reference materials across laboratories can be used to remove most laboratory bias and harmonize datasets, provided that alkali element migration effects (e.g. sodium loss) are sufficiently limited. This is a key reason why the International Focus Group on Tephrochronology and Volcanism (INTAV, recently renamed Commission on Tephrochronology or COT) distributed a set of four reference

glasses to more than 25 laboratories for analysis and why a similar effort is currently underway with additional glasses for LA-ICP-MS. We (Victoria Univ. of Wellington) took part in, and look forward to the publication of the Commission on Tephrochronology (COT) reference glass study, and hope that it can provide a more appropriate reference glass standard for our analyses.

When using the reference glass ATHO-G, it is important to be aware that the published preferred Na2O concentration of 3.75 wt% (Jochum et al., 2006, Table 13h) is too low. Several analyses with significant sodium loss were not screened out prior to calculating the preferred composition. A majority of newer analyses in the GeoReM database exceed the above reference concentration as do XRF and INAA analyses which are immune to alkali element migration. Kimura et al (2018) also report a concentration of 4.6 by FS-LA-ICP-MS. Together, these data suggest that a Na2O concentration of 4.3 to 4.6 wt% for ATHO-G is probably more accurate. Using the 3.75 concentration during EPMA standardization or offline correction will bias all results to too low values. Using the same 3.75 concentration when assessing whether Na-loss is sufficiently controlled by an analytical method may also produce an inaccurate interpretation. We were unaware of this problem with the ATHO-G standard values for Na2O, and appreciate this issue being brought to our attention. The discrepancy will be in the order of $\sim 1$ wt% higher than our current values measured for ATHO-G, and we will further investigate this issue. We also do not hang any interpretations on the Na2O data in these rhyolitic glasses.

Careful removal (but not deletion) of outliers is important before computing summary statistics. The data tables need to be further screened for these. For example, Table 2 contains obviously problematic trace element analyses like line Kaharoa-P_22 which has elevated Zr. Such points should be set aside and labeled as outliers within the dataset, following recommended practice. Analyses with very low totals should be considered as outliers, labeled, and set aside. Often, a 90% total (10% water and unanalyzed elements) is used as the threshold for this, but that is not appropriate in all

cases. Some maximally hydrated tephras yield mean totals of 90-91 wt%, and require a somewhat lower cutoff to allow for the range of analytical precision. If a Cl-bearing epoxy is used, elevated Cl concentrations on points that also have low totals indicate that the beam spot is not entirely on tephra glass. Often, these points will compare well to others when normalized to 100% totals, but the Cl will be inaccurate. We agree that there remain some outliers in the data that need to be removed, but appreciate Assoc. Prof Kuehn's suggestion they should be "set aside and labelled". Once this further data reduction is undertaken, we will revise the statistical analyses and the figures and tables presented.

We are also aware, as also highlighted by both R1 and R2, that some of the H2OD values are 'too high' in the major element data and these will be removed. We agree that in some circumstances a standard cut-off is not necessarily applicable, however, because only ~1.6% of our data are $\geq$ 8 wt% and this 1.6% exists in a range of samples, from a range of sites, we think this is appropriate for our purpose here.

I am also concerned about the removal of individual data values that fall below the single analysis detection limits, despite this being commonly applied. It is much better to include the actual data values and also report what the detection limits are. This way, information is not discarded, and each end user can determine what is fit for their own purpose. Fundamentally, analytical techniques like EPMA and ICP-MS are based on signal counting and essentially follow Poisson statistics. Consequently, replicate analyses on a homogeneous material follow a probability distribution. At low enough concentrations, a portion of that distribution will begin to fall below the single analysis detection limit at some selected level of confidence (often chosen as 99% or 3 sigma, but 2 sigma and 1 sigma are valid confidence levels as well). At this point, those specific analyses that fall below the detection limit are no less valid as members of the population than those analyses which fall above the limit. It is therefore completely inappropriate to compute summary statistics like mean, median, mode, or standard deviation when the low end of that population has been removed. Removing those

below detection limit values and then computing the mean and standard deviation of those that remain biases the mean to too high a value and reduces the standard deviation, misrepresenting the true precision. We greatly appreciate this detail from Assoc. Prof Kuehn and will include more details about detection limits within our text.

Another community-recommended best practice is to have all tephra analyses and glass standard analyses linked on an analytical session (run) basis. This is because results on common reference glasses are perhaps the best demonstration of analytical quality and because instrument performance can vary from session to session. In SM Table 4, trace element standard analyses are clearly labeled with the individual runs. Please add equivalent information to the other tables for both standards and samples. We have split our data into analytical sessions to allow further clarity to the data reduction and standardisation practises.

Where multiple isotopes are measured for the same element, could these be used in combination rather that reporting concentrations from each with assumed isotopic ratios? (Or alternately use the one less prone to interferences, or an average.) We will provide a single isotope value for the duplicate analyses of trace elements on multiple isotopes. This method is run as a standard in our lab to test for interferences, but the data do not necessarily need to be presented here.

SM Table 2 puts some EPMA and LA-ICP-MS analyses together on the same line. Presumably this means that both were done on the same shards. If so, state this explicitly. Yes, this is our process. We have added a statement about this in the text.

SM Table 3 Major Standards omits results on VG-A99 despite the paper text indicating that this was run as a secondary (and primary) standard. Please add the data for VGA99 to the table. Amended

This same table lists some individual analyses with surprisingly low Na2O concentrations for ATHO-G. What is going on with these? Could these represent cases where the electron beam analyzed the same location twice? These problematic results should be

set aside and labeled as such but not entirely removed from the table. Yes, we believe the low Na2O values are indeed where the points have not been plotted far enough away and the area has effectively been run twice. We will remove these points from the statistical assessment of the data.

SM Table 3 also includes some plots which apparently show results on reference glasses over time. Please add explanation for these. Amended

Also, the plots for ATHO-G show some discontinuities and trends that suggest problems with the standardization and/or drift. These results on ATHO-G are worrying. Please explain what is going on here. Do analyses of VG-A99 show the same patterns? We believe this to be an artefact of re-standardisation after leaving the machine idle for two days. This appears to have only impacted Channel 3 (SiO2 and Al2O3 values), and no other parameters were changed. But, regardless, the data have been standardised separately for these two run sessions.

The text mentions that offline data reduction was used to correct for variability in the VG-568 primary calibration. Are the ATHO-G results in this table the raw or corrected values? They are corrected values. A note has been made in the table to clarify this.

SM Table 4 – Offset is a percentage. Label it as such for clarity. Relabel 2*Std as 2*StDev for clarity. It may be more useful to compare the offset from reference in ppm to the StDev in ppm as a ratio rather than reporting an offset in % of the reference ppm (or do both). This way, if a result is e.g. 3 standard deviations from reference, this would be more readily apparent. Amended

Line 192 – The proximal-distal differences mentioned raise the question of how it would be known that these are the same tephra? Agreed, hence the need to publish these data to allow rigorous investigation for future studies.

Lines 224- – What was the kV? What is total time for an entire analysis, including spectrometer movement and other instrument overhead? Amended

Lines 232-34 – The text mentions monitoring for drift. Was there drift? How was the drift correction implemented – linear interpolation? No drift was identified

Line 235 – By "applied to all the data" does this mean just all data lines, or all lines and all elements? Where an element is at low concentration on the reference glass (e.g. MnO, MgO in VG-568), the precision may not be sufficient to apply a reliable correction to other analyses. Additionally, such a correction will make little difference where VG568 is both the primary calibration standard and the offline correction standard for the same element. We apply the correction to all the elements – we have noted this comment.

Line 237-238 – For older tephra glasses, most of this water can be secondary hydration (water absorbed from the environment post deposition) rather than magmatic. This is acknowledged later in equation 3. Noted.

Lines 245-255 – Equations 1 and 2 essentially outline a standards-based (reference material based) normalization using a single standard. This could also be done using a consensus of the three EPMA reference glasses (VG-568, VG-A99, and ATHO-G) for even better consistency. Of course, any bad analyses would have to be set aside prior to computing consensus corrections. Noted.

Line 315 – DFA with cross-validation is another way to test this by looking for high rates of misclassification. Noted.

Line 321 – "of" instead of "or" Amended

Lines 347-49 – This is perhaps OK as a rough criterion, but precision also scales with concentration due to signal intensity and counting statistics. At lower concentrations, standard deviations will be smaller in wt%, but the relative standard deviation (as a percentage of the analyzed concentration) will be greater. The reverse is true when going to higher concentrations such that a 4x increase in signal intensity corresponds to a relative standard deviation that is reduced by a factor of 1/2 (i.e. scaling with sqrt

of total x-rays recorded). So, the standard deviation on a secondary standard only provides an estimate of what should be expected for a homogeneous sample when the concentrations of the same element are similar. Consequently the stated +/- 0.23wt% homogeneity cutoff for Fe would not be appropriate for a sample with a much higher or lower concentration of Fe than the standard. We agree and we will edit the text to be more explicit.

Line 364 – Crystallization of biotite is not the only way to affect K2O concentrations. Does the Shane et al (2008) reference provide evidence for biotite fractionation? Shane et al. (2008), Shane et al. (2003 – reference added to the text) and Nairn et al. (2004 – reference added to the text), described two different biotite populations each of which could be used to model fractional crystallisation trends from a zoned magma chamber. We have added extra details into the text to clarify this.

Figure 4 – The TiO2 and MgO appear "quantized" on the plots, probably due to rounding everything to the nearest 0.01 wt% prior to plotting. To avoid artifacts like this, it is often better to carry one extra decimal place. Amended

Line 415 – Perhaps replace "can be used to maintain" with "may exhibit" Amended

Line 453 – Not coincidentally, some of John Westgate's early work in the 1960 and 1970s (e.g. Westgate & Evans 1978) also used the same three elements. Reference added.

Line 454 – Perhaps replace "presence" with "inclusion" Amended

Figure 10 – Was Ba not included in the histograms for some reason? Some of the text and later plots point to Ba as a useful discriminator. Ba is a very useful discriminator, and this attribute can be seen clearly by the PCA graph. We have amended the histograms to reflect this.

Lines 504-505 – Remove "ratios" Removed

Line 606 – Perhaps replace "acts to effectively reduce the variability of" with "exhibits

little variability in" Amended

Line 607 – Perhaps replace "can be used to maintain" with "exemplifies" Amended

Line 610 – Replace "causes the" with "exhibits" Amended

Figure 13 (and supplemental tables) – Where multiple populations are clearly evident as in this figure, it would be useful to clearly identify such populations in the data tables, e.g. as Pop 1, Pop 2. Means and standard deviations should then be computed and reported separately for each major population. Data repositories also need to support archival of such details. A really good suggestion, thank you - amended

Lines 739-757 – Yes, discovering more tephra layers in marine and lacustrine sections than are known from proximal deposits at potential source volcanoes appears to be common in many volcanic regions. (Multiple examples have been reported for e.g. the Cascades arc too such as at Summer Lake, Oregon.) Therefore, developing more complete records of volcanic events (and also understanding their spatial distribution, timing, and eruptive/dispersal processes) requires the integration of proximal and distal tephra records, ideally into accessible databases. Agreed, we are so pleased you recognise the value in this research, thank you.

Tables 1-5 – To facilitate data re-use, provide in the supplemental file spreadsheet versions of all of these tables in addition to the versions embedded within the manuscript. The full excel file with all the Tables and SM Tables will be provided.

Table 1 – Relabel easting and northing as longitude and latitude. Also convert to decimal degrees for simpler presentation and easier reuse. Abbott et al. (2020) noted above also contains recommendations regarding reporting of sample details. Amended

Table 2 – This table lists acquisition time, but it does not specify which is peak and which is background or whether one or two backgrounds were measured. Amended

Please also note the supplement to this comment:

https://gchron.copernicus.org/preprints/gchron-2020-34/gchron-2020-34-AC3-supplement.pdf

**GChronD**

Interactive
comment

---

## Author Response (AR1)

**SCHOOL OF GEOGRAPHY ENVIRONMENT AND EARTH SCIENCE**
TE KURA TĀTAI ARE WHENUA
**VICTORIA UNIVERSITY OF WELLINGTON,** PO Box 600, Wellington 6140, New Zealand
**Phone** + 64 4 463 5337 **Email** geo-enquiries@vuw.ac.nz **Web** wgtn.ac.nz/sgees

1st June 2021

Dear Assistant Professor Jensen,

Thank you for your detailed comments on our response to reviewers for our manuscript submitted for open review to *Geochronology*. Your depth of comment and support are much appreciated.

We have resubmitted a version of the manuscript with tracked changed to allow the revisions to be more apparent. We have addressed all the reviewer comments in this revised version of the manuscript. We have taken particular note of Abbott et al., 2020 "Community Best Practises" manuscript as advised by Reviewer 3, and have made some additional changes based on this guidance.

Primarily, as was requested by all three reviewers, we have spent time re-reducing all the data, including re-standardisation and further outlier removal, to produce what we now believe to be a very robust, accurate dataset. We have also added in extra text to improve the clarity and transparency of the data collection, and reduction, and have updated all the figures and tables to reflect this additional work on the data revisions. We hope you agree the article now suitable for publication as a foundation reference data set for New Zealand tephra studies.

We also note some additional comments from the Editor (shown in Green) which we address below:
Dr. Portnyagin points out the issues using VGA568 as calibration for Cl (too low to be used) and Na issues with ATHO-G in the same run that Dr. Kuehn also discusses. You mention that steps you will take to clarify why there are differences and adjust as necessary, however, you do not address the specific issue with Na and Cl. These should be responded to directly – I do not suggest re-analyses, but there has to be more clarity behind the quality of some of the analyses where standards suggest potential issues. This is also the case with the potential interference on Sc due to the oxide interferences. ThO/Th 1.3-1.8% is quite high, regardless of other reports and is a potential explanation despite the lack of other apparent oxide interferences. I just want to emphasize here that in an analytical dataset this large there is an inevitability that not every run was perfect, but when problems crop up that don't disqualify using the data but complicate it from a particular day, it is best to note all the reasons those problems may have arisen and state them clearly. I believe this is largely what both Dr. Portnyagin and Dr. Kuehn would like to see in the revisions. More acknowledgement, clarity and clear reporting of potential problems
We have added an additional subsection to the results section (Sect. 3.1 Data Quality; line 316- 402) in which we discuss the quality control on the data, and the variability seen in the analyses. This new text allows for more transparency in the dataset. We have also added a number of extra figures in the Supplementary Material (SM Tables and Figs 6) which we have used to show more clearly the quality of the data. This work aims to address the comments by the reviewers on the data quality and the transparency of our data collection and analytical accuracy and precision. As advised we hope we have acknowledged the key issues brought up by the reviewers.

As you have noted, the screening of the data is an issue raised by all the reviewers. I suggest (as seems to be your intent) to take the path suggested by Dr. Kuehn, which is to remove outliers from the main population

but still report them (just noted as outliers). This overcomes some of the interpretive challenges and allow you to make specific notes for the outliers as well (i.e., potential phenocryst contamination vs. outlier glass composition etc.).

However, Dr. Portnyagin does make some helpful suggestions as to how to parse the trace element data in particular, and in my opinion, does not force an overly heavy discrimination of the data as you argue happens on occasion. There are natural trends and variation that can be expected but then there are some that are much more likely contamination.

We have extensively screened the data and removed the outliers – which are now reported at the end of the data tables in the Supplementary Material.

The anonymous reviewer does have some challenging comments, several of which I see as beyond the scope of what your paper is attempting to achieve (as you note). However, I do think it would be helpful to make sure to explicitly address some of their points about geographic/stratigraphic coverage. You give a strong response to their comment, but it is not clear how you will address it in the paper itself. In particular I would be curious about their point that some of the localities they mention cannot be dismissed because of your reporting of layers from the Wanganui Basin - which is apparently relatively poorly understood. They appear to anticipate your response in their comment and you do not quite address it.

We have added some extra details in Section 2.1. (now entitled "Sample selection, collation, and collection"; line 181-198) to address the anonymous reviewers concerns about the geographic coverage of the data.

There seems to be a potential misinterpretation about the comments on the juvenile clasts. Juvenile 'clasts' in the way that the anonymous reviewer is discussing them refers specifically to the magmatic ejecta (i.e., just the tephra). So when they discuss the proximal deposit lapilli and pumice they are simply discussing the size fraction of the proximal tephra. In this context a pumice/lapilli clast is no different than a smaller glass shard – just bigger (and potentially more heterogeneous). And just like distal glass, they require numerous analyses to gain a true sense of the geochemical heterogeneity in an eruption. In that context do not quite understand your statement that an analysis on lapilli is essentially equal to a whole rock analysis– an analysis on a single lapilli is no different and an analysis on a distal glass shard. What I think the reviewer is getting at is that in your proximal deposit samples, some of the heterogeneity maybe due to the analysis of non-tephra glass, which I think is a fair interpretation. Reading the methods, you chose a sieved size fraction – a relatively fine one – and there was no componentry in the choice of material to analyse in the proximal samples. In this case there certainly is a possibility that this size fraction could contain shattered glassy material from a dome or obsidian - when broken they can look like (generally blocky) glass shards.

Regardless, in most of these cases I think simply a clear acknowledgement of some of the challenges in interpreting the analyses and organizing them as noted by Dr. Kuehn would be sufficient.

We have added extra clarity into the methods sections for EPMA analysis (Sect. 2.3) and LA-ICP-MS analysis (Sect 2.4). In addition, as reported above, we have added an additional paragraph to the results section (Sect 2.1) in which we discuss the quality control on the data, and the variability seen in the analyses. This allows for more transparency in the dataset. We have also added a few extra figures in the Supplementary Material (SM Tables and Figs 6).

Some minor comments:

For Dr. Kuehns comment on line 192 – this is an important question, and you agree with it but don't offer how you are going to address it in the paper (with a statement that some of these are uncertain?)

See text added lines 210 to 212

Additionally, for some other comments (e.g., Line 245-255) you mention they are "noted" – does that mean his suggestion will be utilized or simply you acknowledge the option of doing it this way but will stick with the original method?

At this stage we have just noted these as we saw these as "suggestions" rather than requirements: lines 245-255 - to use VG-A99 and ATHOG as normalisation standards would preclude their use as secondary standards and hence would likely require a further standard to be used as a secondary standard; line 315 – we intend on having a follow up manuscript that looks into the statistical options for this data, hence why only simple PCA and ESC have been used at this stage.

Below we also detail the key changes made to the Figures and Tables:

Figure edits
Below we detail all the edits made to the figures in the updated version of this manuscript:
- Figure 1 no changes made
- Figure 2 edited with outliers removed for points 8 and 9 of NIST 610, note made in the figure caption
- Figure 3 updated after outlier removal and further data reduction
- Figure 4 updated after outlier removal and further data reduction; rounding improvement could not be made, as requested by R3, but this doesn't impact the data just the image aesthetics.
- Figure 5 updated after outlier removal and further data reduction
- Figure 6 updated after outlier removal and further data reduction
- Figure 7 updated after outlier removal and further data reduction
- Figure 8 updated after outlier removal and further data reduction; also now plotted as sample/primitive mantle (Sun and McDonough 1995) rather than sample/chondrite as requested.
- Figure 9 PCA analysis has been re-run after performing centre log ratio on the data set to account for the closure effect on the data therefore this is a new plot for this data.
- Figure 10 PCA analysis has been re-run after performing centre log ratio on the data set to account for the closure effect on the data therefore this is a new plot for this data.
- Figure 11 updated after outlier removal and further data reduction
- Figure 12 updated after outlier removal and further data reduction
- Figure 13 updated after outlier removal and further data reduction
- Figure 14 ESC rerun after outlier removal, and new figures made to reflect this change
- Figure 15 updated after outlier removal and further data reduction

Table edits
The following table edits have been made, and the table references have been updated in the text:
-Table 1 locations converted to Lat, Long in decimal degrees; Smithsonian GVP numbers added; date of analysis for EPMA and LA-ICP-MS added.

- Table 2 moved to SM Table 1.1b to be read alongside the EPMA set up information, additional data added to show crystals on which each element was run.

- Table 2 (new; original Table 3) updated after removal of outliers this information is also added in the text lines 260 to 261 and line 303.

- Table 3 new; original Table 4 – updated in the text. Additional details added for the Taupō eruption which there is new data for from the work of Barker et al., 2015

-Table 4 new; original Table 5 – updated references in the text, no changes made to this Table.

- SM Table 1 updated after Abbott et al., 2020. This now shows full details for EPMA (SM Table 1.1a and 1.1b) and LA-ICP-MS (SM Table 1.2) set up as prescribed by "Best Practise guidelines" and requested by all Reviewers. High oxide levels now discussed in text lines 383 to 393 with reference to SM Table 6, Figures 6.2.3, 6.2.4, and 6.2.5 which show plots of isotope comparisons where multiple isotopes of the same element were run, and between Eu and Ba. The isotope comparison plots (6.2.3 and 6.2.4) show R2 values ≥ 95% suggesting no impact from oxides. In addition, the plot of Eu vs Ba (6.2.5) shows now correlation also showing little to no impact from oxide production.

- SM Table 2 updated with outliers removed ($H_2O_D \geq 8$ wt%); removal of double isotopes; statement of clarity about data presented detailed in the text lines 260 to 261

- SM Table 3 values for A99 added, and details added to give more information on the figures show, outliers removed from results and added to list at bottom of the tables, and references to standard values given. SM Table 6 added to give more justification as to the reason for the use of a different standard value than the Jarosewich et al., 1979 paper. Details for this have been added in the text lines 362 to 382.

- SM Table 4 updated with outliers removed, 2*std changed to "2*stdev" and Offset now reported in (ppm) rather than (%).

-SM Table 6 (new) Figure 6.2.1 and 6.2.2 show plots of the data for elements Ti and Mn, which were run on both the LA-ICP-MS and the EPMA.

- Two versions of data included, individual tables for manuscript publication, and full tables submitted as supplementary files.

We look forward to hearing from you with regards to the improvements to the paper.

Many Thanks,
Dr Jenni L Hopkins (for the Authors)

Jenni.hopkins@vuw.ac.nz

---

## Referee Report (RR1)

The authors have made extensive changes in response to the reviewers and editor, and these has substantially improved the paper. A few key changes include:

- Substantial additional method information and data quality discussion; Includes an updated methods description table covering EPMA and LA-ICP-MS following the approach of Abbott et al. (2020, https://doi.org/10.5281/zenodo.3866266)
- Reworking of data tables and figures and the addition of more supplementary material
- Outlier analyses are retained, commented, and moved to a separate section in the supplemental data tables

The manuscript also now indicates that the full data are accessible via the Pet Lab database at https://pet.gns.cri.nz.  An earlier version was also archived at EarthChem as a file submission. Will this also be updated to the latest version of the dataset?

Since the manuscript and dataset were modified in accordance with the Abbott et al best practice document, those best practices should be cited in the manuscript.

The data tables still contain many entries labeled "Below LOD." These still need to be replaced with the actual analyzed concentrations.  Note the concluding statement of Kirchmer (1994, *Limits of detection and accuracy in trace elements analysis*): **"When reporting data, particularly monitoring data, the critical level, limit of detection, or limit of quantitation should not be used to censor data. To avoid information loss and biased calculations of mean sample concentrations, all data should be reported, together with an estimate of the uncertainty in the results. The critical level should be provided separately as an aid in interpreting the reported results."**

In the track changes version:

Line 212 – If chemistry has differences, how are two samples known to be the same tephra?

Line 219 – Perhaps grain size and tephra samples would be better than glass-shard size and glass samples? Unless density separates are performed to concentrate glass or the original deposits are almost pure glass already, these samples technically aren't composed of just glass.

Line 438 – Do you mean two different magmatic liquids (and therefore glasses) formed due to fractionation of different amounts of biotite?

Lines 906-908 – Perhaps "most responsible for the separation in PC1 and PC2 space" ?

Line 943 – At the time of this re-review, www.tephranz.com resolves to a page with an "Oops! That page can't be found" message. However, the header on this page does allow navigation to the data.

Excel reports a formula error upon opening the Excel file which contains the tables and supplemental data. The problem may lie in the plots associated with supplemental Table 5.

Additional comparative sets of analyses for VG-568 may be found in the following references. The latter uses a high-precision approach with additional EPMA elements (Rb, Sr, Zr).

- Rowe et al., 2008, Catalog of Mount St. Helens 2004–2005 Tephra Samples with Major- and Trace-Element Chemistry, U.S. Geological Survey Open File Report 2008-1131. (see appendix in the pdf)
- Kuehn and Lyon, 2020, June Lake Tephra Dataset https://zenodo.org/record/4074290#.YNEuSi2cZ38 (specifically see June Lake Glass DATA.xlsx)

---

## Author Response (AR2)

**Response to - S. Kuehn – 2nd Review**

The manuscript also now indicates that the full data are accessible via the Pet Lab database at https://pet.gns.cri.nz. An earlier version was also archived at EarthChem as a file submission. Will this also be updated to the latest version of the dataset?
Yes it will, as soon as the manuscript is accepted for publication

Since the manuscript and dataset were modified in accordance with the Abbott et al best practice document, those best practices should be cited in the manuscript.
Details and reference added into the manuscript Ln 97-99, and in SM Tables 1.1 and 1.2, we thank Dr Kuehn for point out this oversight.

The data tables still contain many entries labeled "Below LOD." These still need to be replaced with the actual analyzed concentrations. Note the concluding statement of Kirchmer (1994, Limits of detection and accuracy in trace elements analysis): "When reporting data, particularly monitoring data, the critical level, limit of detection, or limit of quantitation should not be used to censor data. To avoid information loss and biased calculations of mean sample concentrations, all data should be reported, together with an estimate of the uncertainty in the results. The critical level should be provided separately as an aid in interpreting the reported results."
Unfortunately, the output from our LA-ICP-MS programme and iolite does not report these values, we have however added text at the base of **SM Table 2** to show the LOD values for the elements where "below LOD" is reported – we hope this is acceptable.

Line 212 – If chemistry has differences, how are two samples known to be the same tephra?
They are known to be the same through accurate 14C dating of the distal sites, the geochemistry varies due to the potential different phases (and hence chemistries) of the eruptions captured by the different sites. This is discussed further in the text lines 454 - 461 and presented in Figure 7.

Line 219 – Perhaps grain size and tephra samples would be better than glass-shard size and glass samples? Unless density separates are performed to concentrate glass or the original deposits are almost pure glass already, these samples technically aren't composed of just glass.
Edited to remove the definition "glass" line 221.

Line 438 – Do you mean two different magmatic liquids (and therefore glasses) formed due to fractionation of different amounts of biotite?
Yes, text added lines 453-454 for clarity.

Lines 906-908 – Perhaps "most responsible for the separation in PC1 and PC2 space" ?
Changed

Line 943 – At the time of this re-review, www.tephranz.com resolves to a page with an "Oops! That page can't be found" message. However, the header on this page does allow navigation to the data.
Thanks for checking this for us – we will look into this issue and hope to have the page up and running again in time for the publication of the manuscript.

Excel reports a formula error upon opening the Excel file which contains the tables and supplemental data. The problem may lie in the plots associated with supplemental Table 5.
This is now resolved – hopefully.

Additional comparative sets of analyses for VG-568 may be found in the following references. The latter uses a high-precision approach with additional EPMA elements (Rb, Sr, Zr).
• Rowe et al., 2008, Catalog of Mount St. Helens 2004–2005 Tephra Samples with Majorand Trace-Element Chemistry, U.S. Geological Survey Open File Report 2008-1131. (see appendix in the pdf)
• Kuehn and Lyon, 2020, June Lake Tephra Dataset

https://zenodo.org/record/4074290#.YNEuSi2cZ38 (specifically see June Lake Glass
DATA.xlsx)
We really appreciate these studies being brought to our attention and note that the values for Na2O
even within these studies is variable. For example, VG-568 values in Rowe et al (2008) = 3.53 wt.%
whereas in Kuehn and Lyon (2020) = 3.96 wt.%. Comparatively Kuehn and Lyon (2020) also present
Na2O data for ATHO-G at 4.58 wt.% - high in comparison to those reported by Portnyagin et al
(2020) discussed below. We feel that this issue with Na2O analysis is a community problem that
cannot be resolved in the scope of our manuscript, but hope that the transparency in the text now
allows that to be understood by future studies using the data. We also note, that very rarely is Na2O
used in isolation for correlation purposes because of the known problems with analysing it.

**Response to - M. Portnyagin - 2nd Review**

Unfortunately, I have not received a detailed response to my previous comments.
These can be found online at: https://gchron.copernicus.org/preprints/gchron-2020-34/#discussion

On this iteration, I have checked only analytical techniques and tables. I see that the authors made
some efforts towards publication. However, more detailed description of the analytical techniques
revealed new problems.
We note the reviewer has only reviewed the analytical techniques and tables, and has not seen the
rebuttals or changes made in the text therefore they are not necessarily aware of the additional details
and transparency added about some of the comments highlighted below. Regardless, we hope that the
comments made below, and the links to where changes have been made in the text, allow the updated
manuscript to be acceptable to the reviewer.

EMPA
The authors provide more detailed description of their EMP analyses now. This description shows
some fundamental problems, which were not obvious before. The major problem is that they used
rhyolite glass VG568 for calibration of ALL elements, including Ti, Mn, Mg and Cl, which all occur
in VG568 in concentrations below 0.1 wt%! Therefore, the authors obtained large variability of data
for secondary standards, where these elements are more abundant. The best example is MgO in
basaltic glass VGA99 (Figure in SM Table 3). The measured concentrations range from 4 to 6 wt%!
Extremely poor accuracy. In essence, this figure illustrates that calibration of MgO on VG568 was not
precise and varied between 7 sessions, as expected for such low concentration in standard and
therefore imprecise peak positioning. Similar problems are evident for Mn, Ca, Ti, all minor elements
poorly characterized in rhyolite.
See comments below where this is discussed in more detail by the reviewer.

The authors decided to use data for VG568 from Streck and Wacaster (2006) as reference values
("Streck" – not "Stracke" in tables)
Changes made to the spelling error throughout the text, tables, and figures.

Why? Simply because Streck and Wacaster reported some data for minor elements. These are not
certified values. Moreover, these authors used 10nA and 5 um beam to analyze rhyolite glass, giving
high electron beam density, and not surprisingly their Na2O is even lower than reported by Jarosevich
et al. (1980).
(1) See comments about our decision to use the Streck and Wacaster 2006 data as our reference in
original response to this reviewer, and the text lines 363-373 (added in first revision).
 (2) See also the additional supplementary data added in first revision (SM Table 6) where we
compare the data from Streck and Wacaster 2006 with the data from Jarosewich et al (1980) and show
there is a negligible difference between their values (apart from Na2O, which is discussed further
below).

(3) We also note additional data supplied with this rebuttal to show the difference between the use of the Jarosewich et al and the Streck and Wacaster reference values when using it to correct the raw EPMA data.
(4) Finally we suggest that because we have been very transparent about our process and the reference data used if future studies feel the need to recalibrate the data this is possible.

Na2O is also lower for secondary standard ATHO-G, and it must be higher as it has been discussed in literature already and mentioned in our previous reviews and by the editor.
See comments in more detail below where this issue is discussed in more detail by the reviewer.

I see that counting times for elements are not reported in tables. This was requested.
Counting times are reported on sheet SM Table 1 Analytical set up > SM Table 1.1(b) "EPMA machine set up" - this change was made during the first revision of the manuscript.

Thus, the data suffer from inadequate calibration and imprecise standard values. Normally, I would recommend rejection of manuscript which reported such inadequate EMPA technique. However, I understand that most glasses studied in this work have very low Ti, Fe, Mn, Mg and Cl, and the accuracy of the analyses reduced to 20 rel.% for some of these elements is comparable with single point precision. These elements seem not very important for the regional systematics. The analyses are not perfect but, I guess, still usable for correlation of tephras in this region.

I suggest the authors to do next iteration of their corrections.

Si, Al, Na, K should be re-calculated using reference data from Jarosevich et al. (1980) or even better ATHO-G, which is dry glass less affected by beam damage. Ti, Fe, Mg, Mn must be corrected using basaltic glass VGA99, where these elements are much more abundant. Cl can be kept as it is or slightly corrected using data from Portnyagin et al (2020) who reported average of c.2700 analyses obtained using Cl-rich scapolite as standard. What are these strange numbers written for "Cl" in Table SM6?

We are unsure what the reviewer is suggesting here for "correction". In the first iteration of this manuscript, the details of the calibration and data reduction process were outlined in detail. For clarity, transparency, and in a bid to reduce misunderstanding we list below our data handling process (which can be found in the text Sect. 2.3):
(1) EPMA is calibrated using a range of standards to determine peak positions the standards, crystals, count times and channels used are specific for each element and these details are list in SM Table 1, Table 1.1 (a) and (b) after "Best practises" published by Abbott et al., 2021. We believe this is possibly a source of confusion in the above comments?
(2) During the analysis VG568 is run as a primary standard, VG-A99 and ATHOG are run as secondary standards. For this data collection all samples are rhyolitic, therefore ATHOG provides the most appropriate secondary standard for this process.
(3) EPMA undertakes an online ZAF correction
(4) A secondary offline data reduction process is performed using the primary standard VG568 for all elements. This process involves calculating a correction factor for each element's variation from the reference value for VG568, then applying this correction factor back to all the data.
(5) Sample data are corrected for difference from 100 wt.% total with difference reported as "$H_2O_D$" to allow back calculation to the original values.

In case we have misunderstood the reviewers request, we have undertaken the secondary offline data reduction process using both the Streck and Wacaster (2006) and Jarosewich et al. (1980) reference values, and as suggested above, with different elements calibrated using different standards. We have kept Cl as it is (as suggested), $SiO_2$, $Al_2O_3$, $Na_2O$, and $K_2O$ are calibrated using VG568, and $TiO_2$, $FeO_t$, MgO, MnO are calibrated using VG-A99. We note that Ca was not discussed in the comments above, but based on the concentration of this element and text in Portnyagin et al., (2020) we have

used VG-A99 for Ca. We also note, that where VG-A99 is used as the primary standard it would be a circular argument to look at the VG-A99 output therefore for the mixed calibrations, we have just presented ATHO-G values as a secondary standard.

For clarity:
Calibration 1 – VG568 for all using Streck and Wacaster 2006 reference values
Calibration 2 – VG568 for all using Jarosewich et al., 1980 reference values
Calibration 3 - The mix outlines above using Streck and Wacaster 2006 reference values
Calibration 4 - The mix outlines above Jarosewich et al., 1980 reference values
We have uploaded this as an additional file, but do not intend on having this as a supplementary addition to the published manuscript.

The results show that when VG568 is used as a primary standard the reference values from Streck and Wacaster (2006) result in values for our secondary standards (VG-A99) and (ATHO-G) that are more aligned to the reference values – the preferred range on GeoREM for VG-A99 and Jochum et al., 2006 values for ATHO-G. When the mixed values are used as detailed above (VG568 for $SiO_2$, $Al_2O_3$, $Na_2O$, and $K_2O$; and VG-A99 for $TiO_2$, $FeO_t$, $MgO$, $MnO$, and $CaO$) there are negligible differences between our original results and the output from the new results. See sheet "2$^{nd}$ std compare" and figures therein. We also note that the impact of these different reference values on the sample values are also negligible – see sheet "sample compare".

For these reasons, and the transparency in our methods, the publication of all the reference values used within our methods, and the details highlighted in the text we feel that it is unnecessary to recalibrate the entire dataset.

Na2O is definitely too low in the entire dataset because of too low accepted Na2O in primary standard after Streck and Wacaster. Correction using values from Jarosevich et al. (1980) for VG568 or GEOREM for ATHO-G data will bring the results to the lower range of accepted values. Lowe et al. (2017, page 8) – some co-authors of this manuscript- mentioned that GEOREM Na2O=3.75% for ATHO is likely too low. Portnyagin et al. (2020) suggested Na2O =4.1-4.2% in ATHO based on large set of EMP and LA-ICP-MS data. The authors may consider these results for further correction.
We recognise what is being pointed out here, and agree that depending on the reference data used $Na_2O$ is one of the most impacted (e.g. for ATHOG; 3.85 wt.% (Streck and Wacaster 2006) vs. 4.1 wt.% (Jarosewich et al., 1979)). We have added a further note in the text (line 389-397) to make this concern more apparent, however, the data for our secondary standards are currently in agreement with the preferred value on GeoREM of 3.75 wt% (after Jochum et al., 2006). We feel that going into detail about this reference data issue is beyond the scope of this manuscript, but hope that the additional text, and the transparency in the methods used are now more acceptable?

LA-ICP-MS.
There is still poor agreement between Ti in EMP and La-ICP-MS data. The authors probably misunderstood that the data falling out from the error envelope must be excluded from tables. This was not done. The data for Ti in standards are missing (SM Table 4), but I assume that 20 rel % is acceptable deviation. Larger deviation is symptomatic of contamination of analysis by mineral phases. The data should be excluded from tables.
We have added Ti values back into the standard SM Table 4 as requested, and removed samples with Ti and TiO2 values which fall out of the error envelope (SM Table 6), the $R^2$ value for this relationship is now 0.71 (Figure SM 6.2.1).

Sc data for secondary standards is strongly affected by interference with SiO, as already said before and unevoidable at such high ThO/Th ratios. This is why they have up to 80 rel% (5 ppm) deviation from reference data for StHs6/80. This data should not be reported. The authors should admit that they likely had strong addition from SiO but because their samples have similar SiO2 content with ATHO, the data can still be usable. In other words, the relative contribution from SiO is similar for ATHO and unknown rhyolite samples. The data is however not usable for samples with lower SiO2

such as BHVO-2G or StHs60/8-G. Perhaps, other authors also reported ThO/Th>1%. It does not mean that this is good practice and that Sc data is not affected by oxides. It means that the authors run instrument with not optimal conditions and could generate poor results for some elements

Text, figures, and additional supplementary data (**SM Figure 6.2.5**) was added to highlight this problem in the first review of this manuscript, within which we comment on the impact SiO can have on Sc, and BaO can have on Eu. We also highlight our method to monitor and check this issue (lines 398 – 408). We have added in text to further highlight this issue, and add transparency to the data quality as suggested above (lines 403 to 406).